# GENERAL RISK MEASURE MEETS OFFLINE RL: PROVABLY EFFICIENT RISK-SENSITIVE OFFLINE RL VIA OPTIMIZED CERTAINTY EQUIVALENT

## ABSTRACT

We study the risk-sensitive reinforcement learning (RL), which is crucial in scenarios involving uncertainty and potential adverse outcomes. However, existing works on risk-sensitive RL either only focus on a specific risk measure or overlook the offline RL setting. In this work, we investigate the provably efficient risk-sensitive RL under the offline setting with a general risk measure, the optimized certainty equivalent (OCE), which captures various risk measures studied in prior risk-sensitive RL works, such as value-at-risk, entropic risk, and mean-variance. To the best of our knowledge, we (i) introduce the first offline OCE-RL frameworks and propose corresponding pessimistic value iteration algorithms (OCE-PVI) for both dynamic and static risk measures; (ii) establish suboptimality bounds for the algorithms, which can reduce to known results for risk-sensitive RL as well as risk-neutral RL with appropriate utility functions; (iii) derive the first information-theoretic lower bound of the sample complexity of offline risk-sensitive RL, matching the upper bounds and certifying optimality of our algorithms; and (iv) propose the first provably efficient risk-sensitive RL with linear function approximation for both dynamic and static risk measures, together with rigorous suboptimality bounds, yielding a scalable and model-free approach.

## 1 INTRODUCTION

Risk-sensitive reinforcement (risk-sensitive RL) is widely used by a variety of risk-sensitive industries, ranging from finance (Hambly et al., 2023), self-driving (Kamran et al., 2020), to wireless networks (Khalifa et al., 2019). In risk-sensitive RL, the agent aims to optimize certain risk-sensitive reward metrics such as mean-variance risk measure (Sood et al., 2023; Huang et al., 2022), entropic risk (Hau et al., 2023), and conditional value-at-risk (CVaR) (Hakobyan et al., 2019). The risk-sensitive nature of these fields makes data collection costly, motivating a line of work on offline risk-sensitive RL (Ma et al., 2021; Zhang et al., 2024), in which the agent only has access to a pre-collected dataset and cannot further interact with the environment.

However, existing offline risk-sensitive RL studies often focus on a single risk measure, and there is no algorithm that is provably efficient for general risk-sensitive measures. Recently, the optimized certainty equivalent (OCE) framework, introduced by Ben-Tal & Teboulle (2007), has emerged as a suitable candidate for risk-sensitive RL research due to its ability to unify commonly used risk measures such as CVaR, entropic risk measure, and mean-variance. Although these works are sufficiently general in terms of risk metrics considered, they only consider the online setting, with little guidance on how to best utilize a pre-collected dataset. The gap in research highlights an intriguing question: **Can we design offline risk-sensitive RL algorithms that are provably efficient for the general OCE risk measure?**

Answering the question posed requires addressing four challenges. First, while pessimism is well understood in the risk-neutral offline RL (Jin et al., 2021; Levine et al., 2020; Nguyen-Tang et al., 2023), it is unclear how pessimistic estimators can be constructed in the offline risk-sensitive RL with general OCE risk measures, as earlier research relied on the mathematical properties of specific risk measures (Zhang et al., 2024). Second, the risk-sensitive RL framework naturally leads to two distinct formulations—dynamic risk and static risk—which introduce additional challenges in algo-

rithm design. A clear discussion and comparison between these formulations is still lacking. Third, as we aim to find provably efficient algorithms for offline risk-sensitive RL, a corresponding lower bound on sample complexity is crucial for validating our results. Finally, while earlier OCE-based RL research focuses on the tabular setting (Xu et al., 2023; Wang et al., 2024), real-world problems often contain large state spaces, and our framework needs to allow for function approximation.

**Contributions.** We make the following four main contributions as we derive a unifying framework for offline risk-sensitive RL with general risk measures. First, we develop a provably efficient offline RL algorithm under both dynamic and static OCE. Second, we provide the suboptimality bounds for the algorithms, which can reduce to risk-neutral RL and various risk-sensitive RL. Third, we obtain the first sample complexity lower bound for offline risk-sensitive RL, which holds for multiple types of offline risk-sensitive RL with the property of OCE. Finally, we generalize our results to the linear function approximation setting, which is the first provably efficient risk-sensitive RL algorithm with linear function approximation for OCE.

**Related Work.** This work builds upon a growing body of research on offline risk-neutral RL, where the central goal is to identify optimal policies using only pre-collected datasets, without additional interaction with the environment (Levine et al., 2020). In such a setting, the agent is required to infer the optimal policy exclusively from the dataset with no direct access to the underlying transition dynamics. A series of recent studies have investigated this challenge from multiple angles, leading to a rich line of results (Chen & Jiang, 2019; Jin et al., 2021; Rashidinejad et al., 2021; Xie et al., 2021; Cheng et al., 2022; Nguyen-Tang et al., 2023).

Our work is closely related to a long line of research on risk-sensitive RL. For the entropic risk measure, Fei et al. (2020) proposed an online algorithm in the tabular MDP setting, which was later extended to the function approximation regime in Fei et al. (2021). For iterated CVaR, Du et al. (2022) introduced a tabular algorithm, while Chen et al. (2023) extended this framework to incorporate function approximation. Xu et al. (2023) developed a dynamic-OCE-based algorithm for online tabular MDPs. In the offline setting, Zhang et al. (2024) proposed a linear function approximation method with entropic risk. In terms of static-OCE risk formulations, Wang et al. (2023) studied the online tabular CVaR-RL problem and further extended their framework to the more general OCE measure in Wang et al. (2024). Beyond these, a number of earlier works have laid theoretical foundations for risk-sensitive RL (Osogami, 2012; Shen et al., 2013; Bäuerle & Rieder, 2014; Prashanth, 2014; Shen et al., 2014; Ma et al., 2025).

There are also a number of works that focus on linear function approximation, which are closely related to our work. Zhang et al. (2024) introduced a linear function approximation method for offline RL under the entropic risk metrics. Our algorithmic design is further motivated by a broader set of advances in function approximation and offline RL methods (Cai et al., 2020; Jin et al., 2020; 2021; Wang et al., 2020; Agarwal et al., 2020; Zanette et al., 2021; Qiu et al., 2022; Zhong & Zhang, 2023; Liu et al., 2023; Modi et al., 2024).

## 2 PROBLEM SETTING

**Offline RL.** We define an episodic Markov decision process (MDP) $\mathcal{M}$ using the tuple $(\mathcal{S}, \mathcal{A}, \mathbb{P}, r, H)$, where $\mathcal{S}$ denotes a (possibly infinite) state space, $\mathcal{A}$ a finite action space, and $H$ the horizon. We let $\mathbb{P} = \{\mathbb{P}_h\}_{h=1}^{H}$ denote the transition kernel, where $\mathbb{P}_h(s'|s,a)$ is the probability of transitioning to state $s' \in \mathcal{S}$ from state $s \in \mathcal{S}$ upon taking action $a \in \mathcal{A}$ at step $h$. We assume a deterministic reward function $r = \{r_h\}_{h=1}^{H}$, where $r_h : \mathcal{S} \times \mathcal{A} \to [0, 1]$. We assume both $\mathbb{P}, r$ are unknown beforehand, and wlog assume that the initial state is fixed at some $s_1$.

We assume a pre-collected dataset is generated by some behavioral policy, formalized as follows.

**Assumption 2.1 (Offline Dataset)** *Let* $\mathcal{D} = \left\{ \left( s_h^k, a_h^k, r_h(s_h^k, a_h^k) \right) \right\}_{h=1,k=1}^{H,K}$ *be a pre-collected dataset consisting of $K$ trajectories. Assume that the dataset is generated by an unknown behavioral policy $\mu$ via interacting with the environment.*

For any policy $\pi$, define its state-action distribution as $d_h^\pi(s,a) = \Pr(s_h = s, a_h = a | \pi, s_1)$, where $d_h^\pi(s) = \Pr(s_h = s | \pi, s_1)$. In line with existing offline RL research, we define the single concentrability coefficient as follows.

**Definition 2.1 (Single Concentrability)** *For an optimal policy $\pi^*$, we define $C^*$ to be the smallest value such that $\max_{h \in [H], (s,a) \in \mathcal{S} \times \mathcal{A}} \frac{d_h^{\pi^*}(s,a)}{d_h^\mu(s,a)} \leq C^*$.*

**Optimized Certainty Equivalent.** This work focuses on risk-sensitive offline RL incorporating a general risk measure named optimized certainty equivalent (OCE) (Ben-Tal & Teboulle, 2007).

**Definition 2.2 (OCE)** *Let $u : \mathbb{R} \to [-\infty, +\infty)$ be a closed, non-decreasing, and concave utility function with a non-empty effective domain. The OCE of a random variable $X$ is defined as*

$$\mathrm{OCE}^u(X) = \sup_{b \in \mathbb{R}} \left\{ b + \mathbb{E}\big[ u(X - b) \big] \right\}. \tag{1}$$

We note that OCE is a sufficiently general risk measure. Depending on the choice of the utility function $u$, which typically satisfies $u(0) = 0$ and $1 \in \partial u(0)$, OCE recovers commonly used risk metrics such as conditional value-at-risk (CVaR), entropic risk, and mean variance. We refer interested readers to Table 1 for a list of specific instantiations of OCE risk. In addition to its generality, OCE has several key properties, including monotonicity, translation invariance, and positive homogeneity. We defer detailed discussions to Appendix A.1. This paper investigates the OCE risk measure in both of its two formulations, dynamic-OCE RL and static-OCE RL.

**Dynamic-OCE RL.** Under this setting, we have a Markovian policy $\pi = \{\pi_h\}_{h=1}^H$, where $\pi_h(a|s)$ is the probability of taking action $a$ at state $s$ at step $h$ and $\Pi$ is the associated policy class. To incorporate risk measures into sequential decision-making, the dynamic-OCE RL formulation has been proposed in prior works (Ruszczyński, 2010; Bäuerle & Glauner, 2022; Xu et al., 2023), leading to the following Bellman equation that applies the OCE risk measure *iteratively* from step $H$ to 1:

$$Q_h^\pi(s_h, a_h) = r_h(s_h, a_h) + \mathrm{OCE}^u_{s_{h+1} \sim \mathbb{P}_h(\cdot|s_h, a_h)}\big( V_{h+1}^\pi(s_{h+1}) \big), \quad V_h^\pi(s_h) = \big\langle Q_h^\pi(s_h, \cdot), \pi_h(\cdot|s_h) \big\rangle_\mathcal{A},$$

where $V_h^\pi$ and $Q_h^\pi$ are the dynamic-OCE value function and dynamic-OCE Q-function at step $h$ under policy $\pi$. With a slight abuse of notations, we let $\mathrm{OCE}^u_{s' \sim \mathbb{P}_h(\cdot|s, a)}\big( V_{h+1}^\pi(s') \big) := \sup_{b \in \mathbb{R}} \{ b + \mathbb{E}_{s' \sim \mathbb{P}_h(\cdot|s, a)}[u(V_{h+1}^\pi(s') - b)] \}$. According to the definition of OCE in Equation 1, there exists an optimal policy $\pi^* = \{\pi_h^*\}_{h=1}^H$ such that $\pi^* = \arg\max_\pi V_1^\pi(s_1)$ (Bäuerle & Glauner, 2022). We evaluate the performance of a policy $\pi$ under dynamic-OCE RL by its suboptimality, defined as

$$\mathrm{SubOpt}_\mathrm{D}(\pi) = V_1^{\pi^*}(s_1) - V_1^\pi(s_1),$$

which quantifies the gap between the value of the optimal policy $\pi^*$ and that of the policy $\pi$ at the initial state $s_1$. A policy $\pi$ is said to be $\varepsilon$-approximate optimal if $\mathrm{SubOpt}_\mathrm{D}(\pi) \leq \varepsilon$.

**Static-OCE RL.** The static-OCE setting considers when a dynamic programming formulation is not possible (e.g. CVaR) for certain choices of $u$. As the optimal policy can be non-Markovian under this setting, we consider the following specialized definition of OCE objective

$$\mathrm{OCE}^u_{\pi, \mathbb{P}}\big( \textstyle\sum_{h=1}^H r_h(s_h, a_h) \big) = \sup_{b \in [0, H]} \left\{ b + \mathbb{E}_{\pi, \mathbb{P}}[u(\textstyle\sum_{h=1}^H r_h(s_h, a_h) - b)] \right\}, \tag{2}$$

where $\mathbb{E}_{\pi, \mathbb{P}}$ represents taking expectation following $a_h \sim \pi_h, s_{h+1} \sim \mathbb{P}_h$ for all $h \in [H]$. Note that by Lemma A.1, the value of $b$ in Equation 2 can be restricted to $[0, H]$. The key challenge is that the optimal policies for the above problem are *history-dependent* (Wang et al., 2024). To tackle this challenge, we employ the augmented MDP (Bäuerle & Ott, 2011; Wang et al., 2024; Bäuerle & Glauner, 2021) with an expanded state space $(s_h, b_h) \in \mathcal{S}_\mathrm{aug} := \mathcal{S} \times [0, H]$ for each step $h$, comprising the state $s_h$ and a budget variable $b_h$ that transitions via $b_{h+1} = b_h - r_h$ with $b_1 \in [0, H]$ chosen by the learning algorithm. The budget variable tracks the cumulative rewards. Under such construction, we define a *Markovian* policy in the form of $\pi_h(a_h|s_h, b_h)$ (with a slight abuse of notation). We define the augmented value functions as $V_h^\pi(s_h, b_h) := \mathbb{E}_{\pi, \mathbb{P}}[u(\sum_{h'=h}^H r_{h'}(s_{h'}, a_{h'}) - b_h)|s_h, b_h]$. Then, a Bellman-like equation is given by

$$Q_h^\pi(s_h, b_h, a_h) = \mathbb{E}_{s_{h+1} \sim \mathbb{P}_h(\cdot|s_h, a_h)}\big[ V_{h+1}^\pi(s_{h+1}, b_{h+1}) \big], \quad V_h^\pi(s_h, b_h) = \langle Q_h^\pi(s_h, b_h, \cdot), \pi_h(\cdot|s_h, b_h) \rangle_\mathcal{A}.$$

where we use $b_{h+1} = b_h - r_h$. By the definition of $V_h^\pi(s_h, b_h)$, we have $V_{H+1}^\pi(s, b) = u(-b), \forall(s, b)$. Further by Equation 2, static-OCE RL equivalently solves $\max_\pi \sup_{b_1 \in [0, H]} \{ b_1 + V_1^\pi(s_1, b_1) \}$, where $\pi$ is the Markovian policy defined on $\mathcal{S}_\mathrm{aug}$. There always exist an initial budget $b_1^*$ and an optimal policy $\pi^* := \{\pi_h^*\}$ such that $\pi^*$ with $b_1^*$ can maximize $\sup_{b_1 \in [0, H]} \{ b_1 + V_1^\pi(s_1, b_1) \}$ (Wang et al., 2024). Ideally, $b_h$ ought to be a variable in continuous interval $[0, H]$. However, for practical and computationally efficient implementation, we discretize $b_h$ with a $\varepsilon$-net of [0,H], defined as $\mathcal{N}_b := \{ n\varepsilon : n \in \lfloor H/\varepsilon \rfloor \}$. The approximation error introduced by this discretization is negligible as long as $\varepsilon$ is set to be small enough. Accordingly, the suboptimality under any policy $\pi$ in static-OCE RL can be defined as

$$\mathrm{SubOpt}_\mathrm{S}(\pi) := \sup_{b_1 \in [0, H]} \left\{ b_1 + V_1^{\pi^*}(s_1, b_1) \right\} - \sup_{b_1 \in [0, H]} \left\{ b_1 + V_1^\pi(s_1, b_1) \right\}.$$

For dynamic-OCE and static-OCE RL under the offline setting, our goal is to find policies $\widehat{\pi}$ in their corresponding policy classes such that $\mathrm{SubOpt}_\mathrm{D}(\pi)$ or $\mathrm{SubOpt}_\mathrm{S}(\pi)$ is sufficiently small.

| **Algorithm 1** DOCE-PVI | **Algorithm 2** SOCE-PVI |
|---|---|
| 1: **Input:** Offline data $\mathcal{D} = \{(s_h^k, a_h^k, r_h(s_h^k, a_h^k))\}_{h=1,k=1}^{H,K}$ | 1: **Input:** Offline data $\mathcal{D} = \{(s_h^k, a_h^k, r_h(s_h^k, a_h^k))\}_{h=1,k=1}^{H,K}$ |
| 2: **Initialize:** $\widehat{V}_{H+1}(s) = 0$ for all $s$ | 2: **Initialize:** $\widehat{V}_{H+1}(s,b) = u(-b)$ for all $(s,b)$ |
| 3: **for** $h = H, H-1\ldots, 1$ **and** all $(s,a,b) \in \mathcal{S} \times \mathcal{A}$ **do** | 3: **for** $h = H, H-1\ldots, 1$ **and** all $(s,a,b) \in \mathcal{S} \times \mathcal{A} \times \mathcal{N}_b$ **do** |
| 4: Estimate $\widehat{\mathbb{P}}_h(\cdot\|s,a)$ and $\widehat{r}_h(s,a)$ using $\mathcal{D}$ via Equation 3 | 4: Estimate $\widehat{\mathbb{P}}_h(\cdot\|s,a)$ and $\widehat{r}_h(s,a)$ using $\mathcal{D}$ via Equation 3 |
| 5: $\Gamma_h(s,a) = \sqrt{\frac{1}{\max\{1, N_h(s,a)\}}} + [u(H-h) - u(h-H)]\sqrt{\frac{2\log(\|\mathcal{S}\|\|\mathcal{A}\|HK/\delta)}{\max\{1, N_h(s,a)\}}}$ | 5: $\Gamma_h(s,a) = u(H-h)\sqrt{\frac{2\log(\|\mathcal{S}\|\|\mathcal{A}\|HK/\delta)}{\max\{1, N_h(s,a)\}}}$ |
| 6: $\overline{Q}_h(s,a) = \widehat{r}_h(s,a) + \mathrm{OCE}_{s'\sim\widehat{\mathbb{P}}_h(\cdot\|s,a)}^u\left[\widehat{V}_{h+1}(s')\right] - \Gamma_h(s,a)$ | 6: Let $b' := b - \widehat{r}_h(s,a)$ |
| 7: $\widehat{Q}_h(s,a) = \mathrm{clip}\{\overline{Q}_h(s,a), [0, H-h+1]\}$ | 7: $\overline{Q}_h(s,a,b) = \mathbb{E}_{s'\sim\widehat{\mathbb{P}}_h(\cdot\|s,a)}[\widehat{V}_{h+1}(s',b')] - \Gamma_h(s,a)$ |
| 8: $\widehat{\pi}_h(\cdot\|\cdot) = \arg\max_{\pi_h} \langle \widehat{Q}_h(\cdot,\cdot), \pi_h(\cdot\|\cdot)\rangle_{\mathcal{A}}$ | 8: $\widehat{Q}_h(s,a,b) = \mathrm{clip}\{\overline{Q}_h(s,a,b), [u(-b), u(H-h+1-b)]\}$ |
| 9: $\widehat{V}_h(\cdot) = \langle \widehat{Q}_h(\cdot,\cdot), \widehat{\pi}_h(\cdot\|\cdot)\rangle_{\mathcal{A}}$ | 9: $\widehat{\pi}_h(\cdot\|b) = \arg\max_{\pi_h} \langle \widehat{Q}_h(s,\cdot,b), \pi_h(\cdot\|s,b)\rangle_{\mathcal{A}}$ |
| 10: **end for** | 10: $\widehat{V}_h(s,b) = \langle \widehat{Q}_h(s,\cdot,b), \widehat{\pi}_h(\cdot\|s,b)\rangle_{\mathcal{A}}$ |
| 11: **Return:** $\widehat{\pi} = \{\widehat{\pi}_h\}_{h=1}^H$. | 11: **end for** |
| | 12: $\widehat{b}_1 = \arg\max_{b\in\mathcal{N}_b}\{b + \widehat{V}_1(s_1, b)\}$ |
| | 13: **Return:** $\widehat{\pi} = \{\widehat{\pi}_h\}_{h=1}^H, \widehat{b}_1$. |

# 3 RISK-SENSITIVE OFFLINE RL WITH OCE

In this section, we study the learning algorithms for risk-sensitive offline RL with both dynamic-OCE RL and static-OCE RL formulations in the tabular setting.

## 3.1 DYNAMIC-OCE PESSIMISTIC VALUE ITERATION

**Algorithm.** We first propose a pessimistic value iteration algorithm for the dynamic-OCE RL setting named **D**ynamic-**OCE P**essimistic **V**alue **I**teration (DOCE-PVI), summarized in Algorithm 1. The algorithm first estimates the transition and the reward via

$$\widehat{\mathbb{P}}_h(s'|s,a) = \frac{N_h(s,a,s')}{\max\{1, N_h(s,a)\}}, \quad \widehat{r}_h(s,a) = \frac{\sum_{k=1}^K \mathbb{I}\{(s_h^k, a_h^k) = (s,a)\}r_h(s_h^k, a_h^k)}{\max\{1, N_h(s,a)\}}, \quad (3)$$

where $\mathbb{I}\{\cdot\}$ is an indicator function and $N_h(s,a,s')$ and $N_h(s,a)$ are the state-action visitation counters for the pre-collected data $\mathcal{D}$, defined as $N_h(s,a,s') = \sum_{k=1}^K \mathbb{I}\{(s_h^k, a_h^k, s_{h+1}^k) = (s,a,s')\}$ and $N_h(s,a) = \sum_{k=1}^K \mathbb{I}\{(s_h^k, a_h^k) = (s,a)\}$. The bonus, $\Gamma_h$, is constructed on Line 5, which measures the uncertainty related to model estimation. The term explicitly incorporates the OCE risk measure through the factor $u(H-h) - u(h-H)$, a term that depends on the choice of the utility function $u$. Lines 6 and 7 pessimistically estimate the Q-function, denoted by $\widehat{Q}_h$, via the Bellman equation formulation of the OCE risk in the dynamic-OCE setting. The $\mathrm{clip}\{x, [a,b]\}$ operator in Line 7 projects $x$ into the interval $[a,b]$ to ensure boundedness. The estimated optimal policy at step $h$, denoted by $\widehat{\pi}_h$, is a greedy deterministic policy based on the Q-function estimate $\widehat{Q}_h$, and the value function estimate $\widehat{V}_h$ is then constructed using the learned policy. We note that the algorithm degenerates to the risk-neutral pessimistic value iteration when $u(t) = t$. The algorithm involves an optimization problem of the form $\mathrm{OCE}_{s'\sim\widehat{\mathbb{P}}_h(\cdot|s,a)}^u[\widehat{V}_{h+1}(s')] = \sup_{b\in[0,H-h]} \sum_{s'\in\mathcal{S}} \widehat{\mathbb{P}}_h(s'|s,a)[b + u(\widehat{V}_{h+1}(s') - b)]$ in Line 6, which depends on the choice of $u$. Since $u$ is concave, this becomes a one-dimensional concave maximization problem with an efficient solution.

**Theoretical Result.** The following theorem establishes the suboptimality bound for Algorithm 1.

**Theorem 3.1** *For offline dynamic-OCE RL under the tabular setting, with probability at least $1 - \delta$ for $\delta \in (0, 1)$, the learned policy $\widehat{\pi}$ via Algorithm 1 admits the following suboptimality bound*

$$\mathrm{SubOpt}_{\mathrm{D}}(\widehat{\pi}) \leq \widetilde{\mathcal{O}}\Big(\sum_{h=1}^H [u(H-h) - u(h-H)]\sqrt{C^*|\mathcal{S}|/K}\Big),$$

*where $\widetilde{\mathcal{O}}$ hides logarithmic dependence on $H, |\mathcal{S}|, K,$ and $1/\delta$.*

In Theorem 3.1, the result depends on the utility function $u$ in the OCE, reflecting the influence of risk consideration. As this is the first result of the upper bound on offline risk-sensitive RL, we compare our approach with non-risk-sensitive offline value iteration algorithms to examine their similarities and differences, and to verify the effectiveness of our method. Compared with the result of Xie et al. (2021), our algorithm achieves the same suboptimality upper bound of $\widetilde{\mathcal{O}}(\sqrt{C^*S})$. With respect to the horizon $H$, we have $\mathrm{SubOpt}_{\mathrm{D}}(\widehat{\pi}) \leq 2\sum_{h=1}^H [u(H) - u(-H)]\sqrt{2C^*SK^{-1}\log(SAHK\delta^{-1})}$. Then our result includes a multiplicative factor $[u(H) - u(-H)]$, which represents the risk-sensitive term in the OCE formulation. This reveals that the suboptimality is affected by the risk preferences encoded in the utility function $u$. Moreover, when $u(t) = t$, the overall error scales as $\widetilde{\mathcal{O}}(H^2)$, matching the standard result for vanilla offline RL with

a Hoeffding-style bonus (Levine et al., 2020). That is to say, our algorithm attains the same maximal sample complexity as the standard offline RL algorithms but with an additional risk-sensitive term that captures the influence of risk preferences in the OCE.

To proof Theorem 3.1, we first show that the key point is to bound the error brought by the estimation of Bellman operator, spacificed as $\{r_h(s, a) + \text{OCE}^u_{s' \sim \mathbb{P}_h(\cdot|s,a)}\{\widehat{V}_{h+1}(s')\}\} - \{\widehat{r}_h(s, a) + \text{OCE}^u_{s' \sim \widehat{\mathbb{P}}_h(\cdot|s,a)}\{\widehat{V}_{h+1}(s')\}\}$. The biggest gap here is the nonlinear property of OCE. To facilitate the proof, we design a novel probability measure based on $\mathbb{P}$, so as to transfer the problem to a linear domain. The complete proof is presented in Appendix B.2.

## 3.2 STATIC-OCE RL PESSIMISTIC VALUE ITERATION

**Algorithm.** It is worth noting that the static-OCE RL formulation is distinct from that of the dynamic-OCE RL, and the static-OCE RL requires a history-dependent policy. A detailed discussion of this is provided in Appendix A.2. Based on the definition of static-OCE RL and the corresponding history-dependent policy class, we introduce the **S**tatic-**OCE** **P**essimistic **V**alue **I**teration (SOCE-PVI) algorithm in Algorithm 2 based on the augmented MDP (AugMDP), thereby enabling history-dependent policies via an iterative update on the augmented state space $\mathcal{S}_{\text{aug}}$ as shown in Section 2 .

Algorithm 2 first estimates the transition and reward models via Equation 3 as well. The bonus term $\Gamma_h$ is then computed in Line 5, which measures the model estimation uncertainty for each state-action pair $(s, a)$. The bonus term captures the OCE risk via the factor $u(H - h)$. Importantly, the bonus in Algorithm 2 is not the same as in Algorithm 1, which emphasizes that different problem structures lead to distinct bonus designs. Line 6 presents the transition of the state $b$ to $b'$ based on the estimated reward $\widehat{r}_h$. Lines 7 and 8 construct the pessimistic estimate of the Q-function as $\widehat{Q}_h$ through the static-OCE RL Bellman equation and truncation operator clip. Line 9 gives the estimated optimal policy $\widehat{\pi}_h$ via a greedy optimization of $Q-$function. Line 10 presents the estimated value function $\widehat{V}_h$. The estimated optimal budget $\widehat{b}_1$ is computed via Line 12.

Algorithm 2 outputs a history-dependent policy involving $\widehat{b}_h$ with a recursive update rule starting from $\widehat{b}_1$, i.e., $\widehat{b}_{h+1} = \widehat{b}_h - r_h(s, a)$ where $r_h$ is the observed reward during policy deployment. Due to the special structure of static-OCE RL, we note that Algorithm 2 applies the OCE only once at the end of the algorithm rather than at every step as in Algorithm 1, thereby substantially lowering the overall computational burden. On the other hand, because of this setup, an extra update for the auxiliary state $b$ is required and is performed iteratively during the algorithm. With different choices of $u$, our algorithm can reduce to the risk-neutral offline RL algorithm and to other risk-sensitive offline RL methods with different risk measures.

**Theoretical Result.** The following theorem establishes the suboptimality bound for Algorithm 2.

**Theorem 3.2** *For the offline static-OCE RL under the tabular setting, with probability at least $1-\delta$, for $\delta \in (0, 1)$, the learned policy $\widehat{\pi}$ via Algorithm 2 admits the following suboptimality bound*

$$\text{SubOpt}_{\text{S}}(\widehat{\pi}) \leq \widetilde{\mathcal{O}}\Big( \textstyle\sum_{h=1}^{H} u(H - h)\sqrt{C^*|\mathcal{S}|/K}\Big),$$

*where $\widetilde{\mathcal{O}}$ hides logarithmic dependence on $H, |\mathcal{S}|, K$, and $1/\delta$.*

This result demonstrates that the suboptimality is influenced by the utility function $u$ in the OCE, thereby capturing the effect of risk. Similar to Theorem 3.1, the result achieves a suboptimality upper bound of $\widetilde{\mathcal{O}}(\sqrt{C^*|\mathcal{S}|})$, which is consistent with the standard offline RL algorithms (Xie et al., 2021). For the horizon $H$, we have $\text{SubOpt}_{\text{S}}(\widehat{\pi}) \leq 2\sum_{h=1}^{H} u(H)\sqrt{2C^*|\mathcal{S}|K^{-1}\log(|\mathcal{S}||\mathcal{A}|HK\delta^{-1})}$. When $u(t) = t$, the overall error scales as $\widetilde{\mathcal{O}}(H^2)$, matching the result for vanilla risk-neutral offline RL. However, there remains a difference in the multiplicative factor, namely $u(H)$.

The potential of static-OCE lies not only in extending the problem to history-dependent policy, but also in its its ability to handle stochastic rewards. Therefore, we undertake the more challenging task of proving the suboptimality bound under the stochastic reward setting, which generalizes the deterministic case. In this case, through wisely choice of $b$ and reasonable bounding techniques,

we have $\text{SubOpt}_{\text{S}}(\pi) \leq V_1^*(s_1, b_1^*) - \widehat{V}_1(s_1, b_1^*) + \widehat{V}_1(s_1, b_1^*) - V_1^{\widehat{\pi}}(s_1, b_1^*)$, which serves as the foundation for the subsequent analysis. The detailed proof is provided in Appendix B.4.

For completeness, we conduct a numerical simulation on a well-designed MDP to verify our algorithms, as well as making a comparision between the dynamic and static OCE. Experiments are performed with the CVaR risk measure for different $H$ and $K$. The simulation results demonstrate that the suboptimality decreases with the increase of $K$, and that static-OCE converges faster than dynamic-OCE. These observations are consistent with the theorical results above. The detailed discussion is presented in Appendix F.

## 4 INFORMATION-THEORETIC LOWER BOUNDS

Then we provide the minimax lower bound of the suboptimality in Theorem 4.1.

**Theorem 4.1 (Minimax Lower Bound)** *Consider an MDP* $\mathcal{M} = (\mathcal{S}, \mathcal{A}, H, \mathbb{P}, r)$*, where* $|\mathcal{S}| \geq 3$*,* $H \geq 2$*,* $|\mathcal{A}| \geq 2$*,* $C^* \geq 2$*, and* $K > \frac{1}{4}C^*SH$*. Let* $\mathcal{D}$ *denote a dataset collected from the underlying MDP* $\mathcal{M}$*. Then the following minimax lower bound holds:*

$$\inf_{Alg} \max_{\mathcal{M}} \text{SubOpt}_{\text{D}}(\mathcal{M}, Alg(\mathcal{D}), s_1) \geq \Omega\big(\big[u(\rho H - b_1^*) - u(-b_1^*)\big]\sqrt{C^*|\mathcal{S}|H/K}\big)$$

$$\inf_{Alg} d \max_{\mathcal{M}} \text{SubOpt}_{\text{S}}(\mathcal{M}, Alg(\mathcal{D}), s_1) \geq \Omega\big(\big[u(\rho H - b_1^*) - u(-b_1^*)\big]\sqrt{C^*|\mathcal{S}|H/K}\big),$$

*where* $\rho \in (0,1)$ *is a constant and* $b_1^* = \arg\max_{b \in (0, \rho H)}\{b + \frac{1}{2H}u(\rho H - b) + (1 - \frac{1}{2H})u(-b)\}$*.*

For the first time, we incorporate risk into offline RL and establish the corresponding lower bounds. In particular, we present a general formulation of the lower bound for both the dynamic-OCE and static-OCE, accounting for dataset coverage, through a carefully designed hard-case MDP that incorporates the factor $\rho$. Leveraging the properties of OCE, our results can be specialized to various offline risk-sensitive RL by appropriately choosing the utility function $u$ and the parameter $\rho$. Thus, we provide a general lower bound for offline risk-sensitive RL under broad classes of risk measures.

Letting a constant $c = \frac{b_1^*}{\rho H}, c \in (0,1)$, the lower bound simplifies to $\Omega\big(u(c\rho H)\sqrt{C^*|\mathcal{S}|HK^{-1}}\big)$. Hence, the lower bound in Theorem 4.1 aligns with the upper bounds in Theorems 3.1 and 3.2 in terms of the factor $\Omega(\sqrt{C^*|\mathcal{S}|K^{-1}})$. Nevertheless, a gap remains: the upper bounds scale as $\widetilde{\mathcal{O}}([u(H) - u(-H)] \cdot H)$ and $\widetilde{\mathcal{O}}(u(H) \cdot H)$, whereas the lower bound only grows as $\Omega(u(c\rho H) \cdot \sqrt{H})$.

Moreover, Theorem 4.1 shows that under specially constructed hard instance settings, we observe that both the dynamic-OCE and static-OCE algorithms have the same form of lower bound. The underlying mechanism is that, for hard-case MDPs with a single step of OCE computation and absorbing states, the two OCE settings can achieve the same lower bound. In Appendix C, we show that it is reasonable to construct such hard instances.

To the best of our knowledge, this is the first information-theoretic lower bound for offline RL with OCE. Therefore, in order to verify our results, we first compare against the lower bounds of risk-neutral offline RL algorithms. Our algorithms attain the minimax lower bound $\Omega(\sqrt{C^*|\mathcal{S}|})$, matching the results of Xie et al. (2021); Rashidinejad et al. (2021). However, our lower bound explicitly incorporates the risk-sensitive component through its dependence on the utility function $u$, highlighting the additional complexity introduced by risk considerations in our framework.

Then, we compare our results with the prior lower bounds for online risk-sensitive RL. Xu et al. (2023) proved a lower bound of $\Omega([u((1 - 2/c_2)H - b_1^*) - u(-b_1^*)]\sqrt{C^*|\mathcal{S}|HK}), c_2 > 2$, for online dynamic-OCE RL. Our bound is consistent with theirs, in terms of risk-factor, setting $\rho = 1 - 2/c_2$. Moreover, under specific choices of utility functions, our framework recovers several known online risk-sensitive RL lower bounds: By choosing $u(t) = -\frac{1}{\alpha}[-t]_+$ with $\alpha \in (0,1]$ and $\rho = \sqrt{\alpha^{2-n}}$, our result aligns with the iterated CVaR-based lower bound in Chen et al. (2023). With the same utility function but $\rho = \sqrt{\alpha}$, the risk factor of our bound matches the result of Wang et al. (2023) with CVaR. Setting $u(t) = \frac{1}{|\alpha|}(e^{|\alpha|t} - 1)$ reduces our result to align with the entropic risk-sensitive lower bound established by Fei et al. (2020). For CVaR and mean-variance risk measures, existing lower bounds are restricted to the online setting. Nevertheless, the risk-sensitive terms identified in those works offer valuable guidance for understanding the offline scenario. In addition, there are

related results on offline risk-sensitive RL via entropic risk measure (Zhang et al., 2024). For the risk factor, our lower bound simplifies to $\Omega(\frac{e^{|\alpha|H}-1}{|\alpha|})$, which is consistent with their upper bound, choosing $u(t) = \frac{1}{|\alpha|}(e^{|\alpha|t} - 1)$ and $\rho = \alpha$. The detailed proof of Theorem 4.1 is in Appendix C.2.

# 5 LINEAR FUNCTION APPROXIMATION FOR OFFLINE RL WITH OCE

When facing the large state space, the proposed algorithms under the tabular setting would suffer from high suboptimality bounds according to Theorems 3.1 and 3.2. A key technique for addressing such a challenge lies in employing function approximation. While function approximation has been widely applied in RL, how to design a provable algorithm for RL with the OCE risk measure remains unexplored. This section studies linear function approximation, a practical implementation of function approximation, for offline RL with the OCE measure, and proposes learning algorithms for both dynamic-OCE RL and static-OCE RL.

**Linear MDP.** Considering a commonly adopted linear MDP model, in which both the reward function and the transition kernel admit linear structure, we have

$$r_h(s, a) = \langle \theta_h, \phi(s, a) \rangle, \quad \mathbb{P}_h(\cdot | s, a) = \langle \mu_h(\cdot), \phi(s, a) \rangle, \tag{4}$$

where $\int_{\mathcal{S}} \|\mu_h(s)\| ds \leq \sqrt{d}$ and $\|\theta_h\| \leq \sqrt{d}$. We define $\phi : \mathcal{S} \times \mathcal{A} \to \mathbb{R}^d$ to be a feature map satisfying $\|\phi(s, a)\| \leq 1$ for all $(s, a) \in \mathcal{S} \times \mathcal{A}$. It is also flexible enough to include the tabular MDP setting as a special case by choosing $d = |\mathcal{S}| \cdot |\mathcal{A}|$ and setting the feature map to the canonical basis vector: $\phi(s, a) = \mathbf{e}_{(s,a)}$, assuming discrete state and action spaces.

## 5.1 DYNAMIC-OCE PESSIMISTIC LEAST-SQUARES VALUE ITERATION

**Algorithm.** In this section, we propose the pessimistic value iteration with linear function approximation for the dynamic-OCE RL, termed **D**ynamic-**OCE P**essimistic **L**east-**S**quares **V**alue **I**teration (DOCE-PLSVI), as summarized in Algorithm 3. Due to the special structure in the Bellman equation for dynamic-OCE RL, we consider linear function approximation from two separate aspects. We directly perform the function approximation for the reward function $r_h$ by solving the following ridge regression

$$\min_{\theta \in \mathbb{R}^d} \sum_{k=1}^K \left[ r_h(s_h^k, a_h^k) - \phi(s_h^k, a_h^k)^\top \theta \right]^2 + \lambda \|\theta\|_2^2, \tag{5}$$

such that the estimated reward function is constructed as $\widehat{r}_h(\cdot, \cdot) = \phi(\cdot, \cdot)^\top \widehat{\theta}_h$ with $\widehat{\theta}_h$ being the solution. On the other hand, by exploiting the linear structure of the transition model, we have $\mathbb{E}_{s' \sim \mathbb{P}_h(\cdot|s,a)}[u(\widehat{V}_{h+1}(s') - b)] = \int_{\mathcal{S}}[u(\widehat{V}_{h+1}(s') - b)]\langle \mu_h(s'), \phi(s, a) \rangle \, ds' = \langle w(b), \phi(s, a) \rangle$ where $w(b) := \int_{\mathcal{S}}[u(\widehat{V}_{h+1}(s') - b)]\mu_h(s') \, ds'$. Therefore, the algorithm performs a ridge regression via finding $\widehat{w}_h(b)$ to solve

$$\min_{w(b) \in \mathbb{R}^d} \sum_{k=1}^K \left[ u(\widehat{V}_{h+1}(s_{h+1}^k) - b) - \phi(s_h^k, a_h^k)^\top w(b) \right]^2 + \lambda \|w(b)\|_2^2. \tag{6}$$

Then, we have $\phi(s, a)^\top \widehat{w}_h(b) \approx \mathbb{E}_{s' \sim \mathbb{P}_h(\cdot|s,a)}[u(\widehat{V}_{h+1}(s') - b)]$ without explicitly estimate $\mathbb{P}_h$, which is thus a model-free method. Thus, $\text{OCE}_{s' \sim \mathbb{P}_h(\cdot|s,a)}^u \{\widehat{V}_{h+1}(s')\}$ can be estimated by $\sup_{b \in [0, H-h]}\{b + \phi(s, a)^\top \widehat{w}_h(b)\}$, where the budget $b$ is restricted to $[0, H - h]$ by Lemma A.1. Lines 4, 5, and 6 in Algorithm 3 estimate the parameters for the above least-squares problem. Line 7 constructs the bonus term $\Gamma_h(\cdot, \cdot)$ that measures the uncertainties in estimating the reward $r_h$ and the term $\mathbb{E}_{s' \sim \mathbb{P}_h(\cdot|s,a)}[u(\widehat{V}_{h+1}(s') - b)]$ with $\beta$ being set to $\mathcal{O}(d[1 + u(H - h)]\sqrt{\log(2dHK\delta^{-1})})$ that depends on the utility function $u$. Line 8 and Line 9 construct the pessimistic Q-function $\widehat{Q}_h$ via the Bellman equation and the estimates of the reward function as well as $\text{OCE}_{s' \sim \mathbb{P}_h(\cdot|s,a)}^u \{\widehat{V}_{h+1}(s')\}$ as discussed above. Line 10 offers an estimated greedy optimal policy $\widehat{\pi}_h$. The associated value function $\widehat{V}_h$ is obtained in Line 11.

**Theoretical Result.** We establish the suboptimality bound for Algorithm 3.

| **Algorithm 3** DOCE-PLSVI | **Algorithm 4** SOCE-PLSVI |
|---|---|
| 1: **Input:** Offline data $\mathcal{D} = \{(s_h^k, a_h^k, r_h(s_h^k, a_h^k))\}_{h=1,k=1}^{H,K}$ | 1: **Input:** Offline data $\mathcal{D} = \{(s_h^k, a_h^k, r_h(s_h^k, a_h^k))\}_{h=1,k=1}^{H,K}$ |
| 2: **Initialize:** $\widehat{V}_{H+1}(s) = 0$ for all $s$ | 2: **Initialize:** $\widehat{V}_{H+1}(s,b) = u(-b)$ for all $(s,b)$ |
| 3: **for** $h = H, H-1 \dots, 1$ **do** | 3: **for** $h = H, H-1 \dots, 1$ **and** all $b \in \mathcal{N}_b$ **do** |
| 4: $\quad \Lambda_h = \sum_{k=1}^K \phi(s_h^k, a_h^k)\phi(s_h^k, a_h^k)^\top + \lambda \mathbf{I}$ | 4: $\quad \Lambda_h = \sum_{k=1}^K \phi(s_h^k, a_h^k)\phi(s_h^k, a_h^k)^\top + \lambda \mathbf{I}$ |
| 5: $\quad \widehat{w}_h(b) = \Lambda_h^{-1} \sum_{k=1}^K \phi(s_h^k, a_h^k)u(\widehat{V}_{h+1}(s_{h+1}^k)) - b)$ | 5: $\quad \widehat{w}_h(b) = \Lambda_h^{-1} \sum_{k=1}^K \phi(s_h^k, a_h^k)\widehat{V}_{h+1}(s_{h+1}^k, b - r_h(s_h^k, a_h^k))$ |
| 6: $\quad \widehat{\theta}_h = \Lambda_h^{-1} \sum_{k=1}^K \phi(s_h^k, a_h^k)r_h(s_h^k, a_h^k)$ | 6: $\quad \Gamma_h(\cdot, \cdot) = \beta\sqrt{\phi(\cdot, \cdot)^\top \Lambda_h^{-1}\phi(\cdot, \cdot)}$ |
| 7: $\quad \Gamma_h(\cdot, \cdot) = \beta\sqrt{\phi(\cdot, \cdot)^\top \Lambda_h^{-1}\phi(\cdot, \cdot)}$ | 7: $\quad \overline{Q}_h(\cdot, \cdot, b) = \phi(\cdot, \cdot)^\top \widehat{w}_h(b) - \Gamma_h(\cdot, \cdot)$ |
| 8: $\quad \overline{Q}_h(\cdot, \cdot) = \phi(\cdot, \cdot)^\top \widehat{\theta}_h + \sup_{b\in[0,H-h]}\{b + \phi(\cdot, \cdot)^\top \widehat{w}_h(b)\} - \Gamma_h(\cdot, \cdot)$ | 8: $\quad \widehat{Q}_h(\cdot, \cdot, b) = \mathrm{clip}\{\overline{Q}_h(\cdot, \cdot, b), [u(-b), u(H-h+1-b)]\}$ |
| 9: $\quad \widehat{Q}_h(\cdot, \cdot) = \mathrm{clip}\{\overline{Q}_h(\cdot, \cdot), [0, H-h+1]\}$ | 9: $\quad \widehat{\pi}_h(\cdot|\cdot, b) = \arg\max_{\pi_h} \langle \widehat{Q}_h(\cdot, \cdot, b), \pi_h(\cdot|\cdot, b)\rangle_{\mathcal{A}}$ |
| 10: $\quad \widehat{\pi}_h(\cdot|\cdot) = \arg\max_{\pi_h} \langle \widehat{Q}_h(\cdot, \cdot), \pi_h(\cdot|\cdot)\rangle_{\mathcal{A}}$ | 10: $\quad \widehat{V}_h(\cdot, b) = \langle \widehat{Q}_h(\cdot, \cdot, b), \widehat{\pi}_h(\cdot|\cdot, b)\rangle_{\mathcal{A}}$ |
| 11: $\quad \widehat{V}_h(\cdot) = \langle \widehat{Q}_h(\cdot, \cdot), \widehat{\pi}_h(\cdot|\cdot)\rangle_{\mathcal{A}}$ | 11: **end for** |
| 12: **end for** | 12: $\widehat{b}_1 = \arg\max_{b\in\mathcal{N}_b}\{b + \widehat{V}_1(s_1, b)\}$ |
| 13: **Return:** $\widehat{\pi} = \{\widehat{\pi}_h\}_{h=1}^H$. | 13: **Return:** $\widehat{\pi} = \{\widehat{\pi}_h\}_{h=1}^H, \widehat{b}_1$. |

**Theorem 5.1** *For the offline static-OCE RL with linear function approximation, with probability at least $1 - \delta$, for $\delta \in (0, 1)$, the learned policy $\widehat{\pi}$ via Algorithm 3 admits the suboptimality bound*

$$\mathrm{SubOpt}_\mathrm{D}(\widehat{\pi}) \leq \widetilde{\mathcal{O}}\Big(d\sum_{h=1}^H [1 + u(H-h)]\mathbb{E}_{\pi^*}\Big[\sqrt{\phi(s_h, a_h)^\top \Lambda_h^{-1}\phi(s_h, a_h)}\Big|s_1\Big]\Big),$$

*where $\Lambda_h \leftarrow \sum_{k=1}^K \phi(s_h^k, a_h^k)\phi(s_h^k, a_h^k)^\top + \lambda\mathbf{I}$, $\lambda = 1$. And let $\beta = cd[1 + u(H-h)]\sqrt{\log(2dHK\delta^{-1})}$, where $c$ is a constant satisfying $c > 0$ and $12\log(64c^2) + 46 \leq \frac{c^2}{4}$. $\widetilde{\mathcal{O}}$ hides logarithmic dependence on $H, d, K$, and $1/\delta$.*

Like the tabular dynamic-OCE RL, the result in Theorem 5.1 explicitly depends on the utility function $u$ used in the OCE, thereby capturing the effect of risk. By appropriately selecting parameters, we can achieve a suboptimality bound $\widetilde{\mathcal{O}}(d)u(H)\sum_{h=1}^H \mathbb{E}_{\pi^*}[\sqrt{\phi(s_h, a_h)^\top \Lambda_h^{-1}\phi(s_h, a_h)}|s_1]$, matching prior risk-sensitive offline RL algorithms (Zhang et al., 2024) by taking $u(t) = \frac{1}{\alpha}(e^{\alpha t} - 1)$. When different utility functions $u$ are chosen, the bound naturally adapts to the corresponding risk measure. In particular, setting $u(t) = t$ reduces the result to the offline risk-neutral RL, $\widetilde{\mathcal{O}}(dH)\sum_{h=1}^H \mathbb{E}_{\pi^*}[\sqrt{\phi(s_h, a_h)^\top \Lambda_h^{-1}\phi(s_h, a_h)}|s_1]$ (Jin et al., 2021). Therefore, our algorithm attains the same maximal sample complexity as standard offline RL algorithms, augmented by an additional risk-sensitive term reflecting the influence of risk preferences in OCE.

Compared with Algorithm 1, Algorithm 3 provides a more general framework capable of handling complex high-dimensional state and action spaces. When $\phi(s, a) = \mathbf{e}_{(s,a)}$ and $d = |\mathcal{S}| \cdot |\mathcal{A}|$, we have $\Lambda_h = \mathrm{diag}(\{N_h(s, a) + \lambda\}_{(s,a)\in\mathcal{S}\times\mathcal{A}})$, by the definition of $\Lambda_h$. Consequently, we have $\mathbb{E}_{\pi^*}[\sqrt{\phi(s_h, a_h)^\top \Lambda_h^{-1}\phi(s_h, a_h)}|s_1] = (N_h(s_h, a_h) + \lambda)^{-1/2}$. Substituting this expression into our suboptimality bound in Theorem 5.1 and following the proof in Appendix B.2 yields an upper bound on suboptimality equivalent to $\widetilde{\mathcal{O}}(SA)u(H)\sum_{h=1}^H \sqrt{2C^*SK^{-1}\log(SAHK\delta^{-1})}$. In practical scenarios, the feature dimension $d$ is not necessarily large, and thus the result in Theorem 3.1 can match the result in Theorem 5.1.

To the best of our knowledge, this is the first effective OCE-RL algorithm with linear function approximation, either for online or offline settings. Since we proposed a completely new method of function approximation, it requires totally new function class, which is significant in the theoretic analysis, leading to novel methods of bounding the $\varepsilon-$covering number. The detailed proof is provided in Appendix D.2.

## 5.2 STATIC-OCE RL LEAST-SQUARES VALUE ITERATION

**Algorithm.** To derive a gengeral and practiacl risk-sensitive RL algorithm, we propose the pessimistic value iteration with linear function approximation for the static-OCE RL, termed **S**tatic-**OCE** **P**essimistic **L**east-**S**quares **V**alue **I**teration (SOCE-PLSVI). Based on the linear structure of the transition model, letting $b' = b - r_h(s, a)$, there is $\mathbb{E}_{s'\sim\mathbb{P}_h(\cdot|s,a)}[\widehat{V}_{h+1}(s', b')] = \int_\mathcal{S}[\widehat{V}_{h+1}(s', b')]\langle\mu_h(s'), \phi(s, a)\rangle ds' = \langle w(b), \phi(s, a)\rangle$, where $w(b) := \int_\mathcal{S}[\widehat{V}_{h+1}(s', b')]\mu_h(s')ds'$. Then, we can perform a ridge regression via finding the estimated $\widehat{w}_h(b)$ to solve

$$\min_{w(b)\in\mathbb{R}^d}\left\{\sum_{k=1}^K\left[\widehat{V}_{h+1}(s_{h+1}^k, b - r_h(s_h^k, a_h^k)) - \phi(s_h^k, a_h^k)^\top w(b)\right]^2 + \lambda\|w(b)\|_2^2\right\}.$$

Therefore, we have $\phi(s_h^k, a_h^k)^\top \widehat{w}(b) \approx \mathbb{E}_{s' \sim \mathbb{P}_h(\cdot|s,a)}[\widehat{V}_{h+1}(s', b - r_h(s,a))]$. Unlike the dynamic-OCE RL algorithm in Algorithm 3, with the ridge regressions above, we do not need to estimate $\mathbb{P}_h$ and $r_h$ separately. In Algorithm 4, Lines 4 and 5 implements the estimation for the least-square problem. Line 6 builds the bonus term, denoted by $\Gamma_h(\cdot, \cdot)$ that captures the estimation uncertainty with the station-action pair $(s, a)$. In the bouns term, $\beta$ is set to be $\mathcal{O}(d \cdot u(H - h)\sqrt{\log(2dHK\delta^{-1})})$ based on the utility function $u$. Line 7 and Line 8 implement the estimated Q-function via Bellman equation, incorporating the influence of both risk and pessimism. Lines 9 and 10 respectively derive the estimated optimal policy $\widehat{\pi}_h$ and the value function $\widehat{V}_h$ at step $h$. The optimal initial budget $b_1$ in Algorithm 4 is estimated in Line 12. With the budget $b$, which is updated starting from the initial value $\widehat{b}_1$ by $\widehat{b}_{h+1} = \widehat{b}_h - r_h$, we obtain the history-dependent policy $\widehat{\pi}(\cdot|\cdot, b)$. Notably, Algorithm 4 leverages the parameter $b$ to avoid computing the OCE at every step. Similar to the tabular setting, this simplification comes at the expense of enlarging the state space.

**Theoretical Result.** Next, we present the suboptimality bound for Algorithm 4.

**Theorem 5.2** *For the offline static-OCE RL with linear function approximation, with probability at least $1 - \delta$, for $\delta \in (0, 1)$, the learned policy $\widehat{\pi}$ via Algorithm 4 admits the suboptimality bound*

$$\text{SubOpt}_\text{S}(\widehat{\pi}) \leq \widetilde{\mathcal{O}}\Big(d \sum_{h=1}^H u(H - h)\mathbb{E}_{\pi^*}\Big[\sqrt{\phi(s_h, a_h)^\top \Lambda_h^{-1} \phi(s_h, a_h)}\Big|s_1, b_1^*\Big]\Big),$$

*where $b_1^* = \arg\max_{b_1 \in [0, H]} \{b_1 + V_1^*(s_1, b_1)\}$. And let $\Lambda_h = \sum_{k=1}^K \phi(s_h^k, a_h^k)\phi(s_h^k, a_h^k)^\top + \lambda \mathbf{I}$, $\lambda = 1$, and $\beta = cd \cdot u(H - h)\sqrt{\log(2dHK\delta^{-1})}$, where $c$ is a constant satisfying $c > 0$ and $8\log(64c^2) + 34 \leq \frac{c^2}{4}$. $\widetilde{\mathcal{O}}$ hides logarithmic dependence on $H, d, K,$ and $1/\delta$.*

The result explicitly depends on the utility function $u$ and the optimal initial budget $b_1^*$, reflecting the global consideration of risk in the problem. Theorem 5.2 indicates that Algorithm 4 achieves a sub-optimality upper bound of $\widetilde{\mathcal{O}}(d)u(H) \sum_{h=1}^H \mathbb{E}_{\pi^*}[\sqrt{\phi(s_h, a_h)^\top \Lambda_h^{-1} \phi(s_h, a_h)}|s_1, b_1^*]$ with appropriately chosen parameters. This result properly aligns with the findings of prior offline risk-sensitive RL work (Zhang et al., 2024) when the OCE reduces to the entropic risk measure. Furthermore, by setting $u(t) = t$, the bound simplifies to $\widetilde{\mathcal{O}}(dH) \sum_{h=1}^H \mathbb{E}_{\pi^*}[\sqrt{\phi(s_h, a_h)^\top \Lambda_h^{-1} \phi(s_h, a_h)}|s_1, b_1^*]$, which matches the result of offline risk-neutral RL (Jin et al., 2021).

Moreover, compared with Algorithm 2, Algorithm 4 provides a more general version that can handle complex high-dimensional state and action spaces. When choosing $\phi(s, a) = \mathbf{e}_{(s,a)}$ and $d = |\mathcal{S}| \cdot |\mathcal{A}|$, the matrix $\Lambda_h$ takes the form $\Lambda_h = \text{diag}(\{N_h(s, a) + \lambda\}_{(s,a) \in \mathcal{S} \times \mathcal{A}})$. Accordingly, we obtain $\mathbb{E}_{\pi^*}[\sqrt{\phi(s_h, a_h)^\top \Lambda_h^{-1} \phi(s_h, a_h)} | s_1, b_1^*] = (N_h(s_h, a_h) + \lambda)^{-1/2}$. If we insert this result into the suboptimality bound of Theorem 5.2, and then follow the proof in Appendix B.4, we would obtain the upper bound $\widetilde{\mathcal{O}}(SA) u(H) \sum_{h=1}^H \sqrt{2C^* S K^{-1} \log(SAHK\delta^{-1})}$. Thus, in applied settings where the feature dimension $d$ is not excessively large, the result in Theorem 3.2 can be viewed as a specific instance of the more general bound in Theorem 5.2.

Following the idea of Theorem 2, we try to prove the result extended to stochastic reward cases. However, it becomes even more challenging since the joint distribution of the transition and stochastic reward is required. Therefore, we propose an innovative way of function approximation, which simplifies the problem so that we still have $\mathbb{E}[V_{h+1}(s', b - r)] := \phi(s, a)^\top w(b)$. Additionally, due to our novel construction, we analyse the new covering number in-depth in the proof. The detailed proof of Theorem 5.2 is provided in Appendix D.4.

## 6 CONCLUSION

Since the majority of existing research on risk-sensitive RL primarily focuses on online settings or specific risk measures, we address the offline risk-sensitive RL based on OCE. We develop provably efficient offline RL algorithms for both dynamic-OCE and static-OCE, supported by rigorous theoretical analysis of suboptimality bounds. Additionally, we obtain the first minimax lower bound on the sample complexity of offline risk-sensitive RL. Finally, we propose the first provably efficient risk-sensitive RL with linear function approximation for both dynamic and static OCE and provide rigorous suboptimality bounds.

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

# Appendix

## CONTENTS

# A DISCUSSIONS OF OCE RL

## A.1 PROPERTIES OF OCE

In this section, we demonstrate that the OCE can be reduced to various other risk measures, and summarize some of its key properties. This highlights the flexibility, tractability, and expressive power of the OCE framework.

| Name | $\mathrm{OCE}^u(X)$ | Utility function $u$ |
|------|---------------------|----------------------|
| —rule Mean | $\mathbb{E}[X]$ | $u(t) = t$ |
| Entropic risk | $\frac{1}{\alpha} \log \mathbb{E}[e^{\alpha X}]$ | $u(t) = \frac{1}{\alpha} e^{\alpha t} - \frac{1}{\alpha}$ |
| Mean-Variance | $\mathbb{E}[X] - cVar(X)$ | $u(t) = (t - ct^2)\mathbb{I}\{t \le \frac{1}{2c}\} + \frac{1}{4c}\mathbb{I}\{t > \frac{1}{2c}\}$ |
| CVaR | $\mathbb{E}[x|X \le \min\{x|F_X(x) \ge \alpha\}]$ | $u(t) = -\frac{1}{\alpha}[-t]_+$ |

Table 1: Special cases of OCE risk measure with corresponding $u$.

Furthermore, for any utility function $u$ satisfying the above properties, a constant $c \in \mathbb{R}$, and a bounded random variable $X$, the OCE satisfies the following desirable properties:

1. $\mathrm{OCE}^u(X + c) = \mathrm{OCE}^u(X) + c$;

2. $\mathrm{OCE}^u(c) = c$;

3. If $X_1(\omega) \le X_2(\omega)$ $(\omega \in \Omega)$, $\mathrm{OCE}^u(X_1) \le \mathrm{OCE}^u(X_2)$;

4. For any $\mu \in (0,1)$, $\mathrm{OCE}^u(\mu X_1 + (1 - \mu)X_2) \ge \mu \mathrm{OCE}^u(X_1) + (1 - \mu)\mathrm{OCE}^u(X_2)$.

Moreover, for the optimization step in the OCE, when $X$ is positive and bounded, it is sufficient to optimize over a finite set of $b$ values rather than the entire space of $b$, as shown in Lemma A.1.

**Lemma A.1** *For any bounded positive random variable $X$, where $X \in [0, M]$ for some $M > 0$, we have,*

$$\mathrm{OCE}^u(X) = \sup_{b \in \mathbb{R}} \left\{ b + \mathbb{E}\big[u(X - b)\big] \right\} = \sup_{b \in [0,M]} \left\{ b + \mathbb{E}\big[u(X - b)\big] \right\}.$$

**Proof** *First, we define a function $F(b)$ as follows:*

$$F(b) = b + \mathbb{E}\big[u(X - b)\big].$$

*Then, we have*

$$\frac{\partial}{\partial b} F(b) = 1 - \mathbb{E}\big[u'(X - b)\big].$$

*Since $1 \in \partial u(0)$ and $u(\cdot)$ is concave, for any $t < 0$ we have $u'(t) > 1$. Therefore, if $b > M$, it follows that $u'(X - b) > 1$, which implies $\frac{\partial}{\partial b} F(b) < 0$. This shows that $F(b)$ is decreasing for $b > M$, and hence its supremum is attained at $b \le M$. Similarly, if $b < 0$, we have $u'(X - b) < 1$, which implies $\frac{\partial}{\partial b} F(b) > 0$. This means that $F(b)$ is increasing for $b < 0$, and thus its supremum is attained at $b \ge 0$. Then, we conclude that*

$$\mathrm{OCE}^u(X) = \sup_{b \in \mathbb{R}} \big\{ F(b) \big\} = \sup_{b \in [0,M]} \left\{ b + \mathbb{E}\big[u(X - b)\big] \right\}.$$

*Then we finish the proof.*

## A.2 DISCUSSION OF STATIC-OCE AND AUGMDP

For the static-OCE setting, our objective is to maximize $\mathrm{OCE}\{\sum_{i=1}^{H} r_i\}$. With the definition of OCE in Equation 1, we have

$$\mathrm{OCE}^*\Big\{ \sum_{i=1}^{H} r_i \Big\} = \max_{\pi \in \Pi_s} \max_{b \in [0,H]} \Big\{ b + \mathbb{E}\Big[ u\Big( \sum_{i=1}^{H} r_i - b \Big) \Big] \Big\}$$

$$= \max_{b \in [0,H]} \Big\{ b + \max_{\pi \in \Pi_s} \mathbb{E}\Big[ u\Big( \sum_{i=1}^{H} r_i - b \Big) \Big] \Big\}.$$

There have been lots of methods proposed to solve the optimization problem of $b$. Thus, the rest of our task is to solve $\max_{\pi \in \Pi_s} \mathbb{E}[u(\sum_{i=1}^{H} r_i - b)]$. Following the idea of RL, we define $V_1^\pi(s_1, b) = \mathbb{E}[u(\sum_{i=1}^{H} r_i - b)]$. Then we can use dynamic programming to obtain $V_1^*(s_1, b)$. Following the Augmented-MDP proposed by Bäuerle & Ott (2011); Bäuerle & Glauner (2021), we have

$$\text{OCE}^*\Big\{ \sum_{i=1}^{H} r_i \Big\} = \max_{b \in [0,H]} \Big\{ b + V_1^{\pi^*}(s_1, b) \Big\}$$

$$= \max_{b \in [0,H]} \Big\{ b + V_1^*(s_1, b) \Big\}$$

Under the setting of AugMDP, we have a history-independent policy $\pi(\cdot|s, b)$, regarding $b$ as an augmented state. However, from the aspect of the original MDP, $\pi(\cdot|s, b)$ is actually a history-dependent policy. Since we have shown that the optimal policy $\pi^*(\cdot|s, b)$ is history-dependent, we can conclude that no history-independent policy could exceed the history-dependent policy on the original MDP. To explain this, considering the CVaR risk measure, we use the following MDP as an example:

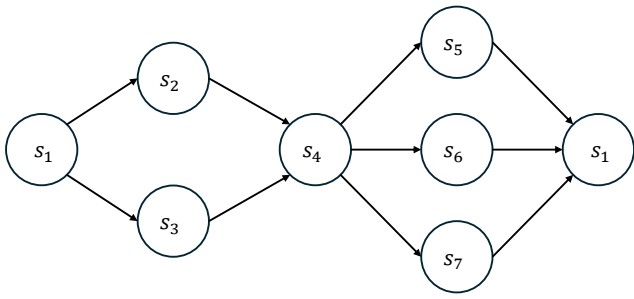

Figure 1: An MDP with the history-dependent optimal CVaR policy.

In this MDP, there are seven states $(s_1, s_2, \ldots, s_7)$ in the state space $\mathcal{S}$ and two actions $(a_1, a_2)$ in the action space $\mathcal{A}$, we have

$$\mathbb{P}(s_1|s_5, a) = \mathbb{P}(s_1|s_6, a) = \mathbb{P}(s_1|s_7, a) = 1, \quad \forall a \in \mathcal{A}$$
$$\mathbb{P}(s_2|s_1, a) = \mathbb{P}(s_3|s_1, a) = 0.5, \quad \forall a \in \mathcal{A}$$
$$\mathbb{P}(s_4|s_2, a) = \mathbb{P}(s_4|s_3, a) = 1, \quad \forall a \in \mathcal{A}$$
$$\mathbb{P}(s_5|s_4, a_1) = 0.75, \quad \mathbb{P}(s_7|s_4, a_1) = 0.25, \quad \mathbb{P}(s_6|s_4, a_2) = 1$$

and

$$r(s_1, a) = r(s_3, a) = r(s_4, a) = r(s_7, a) = 0, \quad \forall a \in \mathcal{A}$$
$$r(s_2, a) = 1, \quad r(s_5, a) = 1.5, \quad r(s_6, a) = 0.5, \quad \forall a \in \mathcal{A}.$$

And we set $H = 4$. Based on this MDP, we can find that only the action at step $h = 3$ will influence $\text{CVaR}(\sum_{h=1}^{H} r_h)$. Therefore, through computation, we can easily find that $\text{CVaR}(\sum_{h=1}^{H} r_h) = 0.5$ for all possible history-independent policies $\pi_3(\cdot|s_{h=3})$ at step $h = 3$. Then we study the history-dependent policy defined as $\pi_3(\cdot|s_{h=2}, s_{h=3})$. We define

$$\pi_3(a_1|s_{h=2} = s_3, s_{h=3} = s_4) = 1$$
$$\pi_3(a_2|s_{h=2} = s_2, s_{h=3} = s_4) = 1.$$

By taking this history-dependent policy, we have the accumulated reward of the total 4 steps:

$$\sum_{h=1}^{H} r_h = \left\{ \begin{array}{ll} 0, & w.r.p. \frac{1}{8} \\ 1.5, & w.r.p. \frac{7}{8}. \end{array} \right.$$

Then we have $\text{CVaR}(\sum_{h=1}^H r_h) = 0.75$. Therefore, we successfully constructed an MDP, where there is at least one history-dependent policy that surpasses all the history-independent policies. This shows that the optimal policy of static-OCE is history-dependent.

### A.3 COMPARISON BETWEEN DYNAMIC-OCE RL AND STATIC-OCE RL

The dynamic-OCE formulation corresponds to the risk-sensitive RL objective commonly referred to as "dynamic risk" (also known as iterated risk). In this setting, the optimization objective is

$$
J_R = \text{OCE}^u \Bigg\{ r_1(s_1, a_1) + \text{OCE}^u_{s_2 \sim \mathbb{P}_1(\cdot|s_1, a_1)} \Big\{ r_2(s_2, a_2) + \text{OCE}^u_{s_3 \sim \mathbb{P}_2(\cdot|s_2, a_2)} \Big\{ r_3(s_3, a_3) +
$$

$$
\Big\{ \cdots \text{OCE}^u_{s_H \sim \mathbb{P}_{H-1}(\cdot|s_{H-1}, a_{H-1})} \{ r_H(s_H, a_H) \} \Big\} \Big\} \Bigg\}.
$$

A key advantage of the dynamic (iterated) risk is the existence of Bellman equations and optimal Markovian policies, which allow direct adaptations of standard RL algorithms. To illustrate, consider the widely used mean-variance risk measure. By choosing $u(t) = (t - ct^2)\,\mathbb{I}\{t \leq \frac{1}{2c}\} + \frac{1}{4c}\,\mathbb{I}\{t > \frac{1}{2c}\}$, the dynamic-OCE objective reduces to

$$
J_R = \sum_{i=1}^H \mathbb{E}[r_i] - \sum_{i=1}^H \text{Var}[r_i].
$$

In contrast, under the static-OCE setting, when reduced to the mean-variance measure, the objective becomes

$$
J_G = \mathbb{E}\Big[ \sum_{i=1}^H r_i \Big] - \text{Var}\Big[ \sum_{i=1}^H r_i \Big].
$$

In practice, decision-making often involves balancing the expected cumulative reward against its overall variance. Due to the properties of variance, the dynamic-OCE formulation effectively behaves as a step-wise greedy strategy: it separately accounts for the variance of each stage reward. Therefore, unlike the static-OCE formulation, the dynamic-OCE formulation also implicitly incorporates covariance terms across different time steps. This makes dynamic-OCE potentially less stable, being overly aggressive in some cases and overly conservative in others, compared to the static-OCE formulation. Moreover, when OCE reduces to CVaR, the dynamic-OCE formulation becomes particularly difficult to interpret, as CVaR lacks favorable linearity properties. This further highlights that dynamic-OCE risk, while algorithmically convenient, is generally less interpretable than its static-OCE counterpart.

## B PROOFS FOR SECTION 3

### B.1 LEMMAS FOR THEOREM 3.1

Typically, the suboptimal relates to the model evaluation error. Here, we define $\iota_h$ as the error raised by the estimated Bellman equation at step $h$ as

$$
\iota_h(s, a) = r_h(s, a) + \text{OCE}^u_{s' \sim \mathbb{P}(\cdot|s, a)}\{\widehat{V}^\pi_{h+1}(s')\} - \widehat{Q}_h(s, a), \tag{7}
$$

Based on the dynamic-OCE RL setting, we first define the Bellman operator,

$$
\mathbb{B}_h f(s, a) = r_h(s, a) + \text{OCE}^u_{s' \sim \mathbb{P}_h(\cdot|s, a)}\{f(s')\}
$$

$$
\widehat{\mathbb{B}}_h f(s, a) = \widehat{r}_h(s, a) + \text{OCE}^u_{s' \sim \widehat{\mathbb{P}}_h(\cdot|s, a)}\{f(s')\}.
$$

Then, we define the event $\mathcal{E}_h$.

**Definition B.1** *Under the dynamic-OCE setting, define the event $\mathcal{E}_h$,*

$$
\mathcal{E}_h = \Big\{ \Big| \mathbb{B}_h \widehat{V}_{h+1}(s, a) - \widehat{\mathbb{B}}_h \widehat{V}_{h+1}(s, a) \Big| \leq \Gamma_h(s, a) \Big\},
$$

*where $\{\Gamma_h\}_{h=1}^H$ is the bonus, satisfies $\mathbb{P}(\bigcap_{h=1}^H \mathcal{E}_h) \geq 1 - \delta$.*

With event $\mathcal{E}_h$, we can find that the upper bound of suboptimality is related to the Bellman estimation error.

**Lemma B.1** *Under the dynamic-OCE setting, we have*

$$\mathbb{B}_h V_{h+1}^{\pi^*}(s_h, a_h) - \mathbb{B}_h \widehat{V}_{h+1}(s_h, a_h) \leq \mathbb{E}_{s' \sim \mathbb{P}_h(\cdot|s,a)}\Big[V_{h+1}^{\pi^*}(s_{h+1}) - \widehat{V}_{h+1}(s_{h+1})\Big].$$

**Proof** *For the left side of the inequality, we have*

$$\mathbb{B}_h V_{h+1}^{\pi^*}(s_h, a_h) - \mathbb{B}_h \widehat{V}_{h+1}(s_h, a_h)$$

$$= \Big(r_h(s_h, a_h) + \mathrm{OCE}_{s' \sim \mathbb{P}_h(\cdot|s,a)}^u\big\{V_{h+1}^{\pi^*}(s_{h+1})\big\}\Big)$$

$$- \Big(r_h(s_h, a_h) + \mathrm{OCE}_{s' \sim \mathbb{P}_h(\cdot|s,a)}^u\big\{\widehat{V}_{h+1}(s_{h+1})\big\}\Big)$$

$$= \sup_{b \in [0,H]} \Big\{b + \mathbb{E}_{s' \sim \mathbb{P}_h(\cdot|s,a)}\big[u(V_{h+1}^{\pi^*}(s_{h+1}) - b)\big]\Big\}$$

$$- \sup_{b \in [0,H]} \Big\{b + \mathbb{E}_{s' \sim \mathbb{P}_h(\cdot|s,a)}\big[u(\widehat{V}_{h+1}(s_{h+1}) - b)\big]\Big\}.$$

*Then by setting $b^\dagger = \arg\max_{b \in [0,H]}\{b + \mathbb{E}_{s' \sim \mathbb{P}_h(\cdot|s,a)}[u(V_{h+1}^{\pi^*}(s_{h+1}) - b)]\}$, we have*

$$\mathbb{B}_h V_{h+1}^{\pi^*}(s_h, a_h) - \mathbb{B}_h \widehat{V}_{h+1}(s_h, a_h)$$

$$\leq \mathbb{E}_{s' \sim \mathbb{P}_h(\cdot|s,a)}\Big[u\big(V_{h+1}^{\pi^*}(s_{h+1}) - b^\dagger\big) - u\big(\widehat{V}_{h+1}(s_{h+1}) - b^\dagger\big)\Big]$$

$$\leq \mathbb{E}_{s' \sim \mathbb{P}_h(\cdot|s,a)}\Big[V_{h+1}^{\pi^*}(s_{h+1}) - \widehat{V}_{h+1}(s_{h+1})\Big].$$

*The last inequality holds due to $1 \in \partial u(0)$, $0 \leq \widehat{V}_h(s_h) \leq V_h^{\pi^*}(s_h) \leq H$, as well as the concavity and non-decreasing property of the utility function $u$. Here we finish the proof.*

**Lemma B.2** *Under the dynamic-OCE setting, there is*

$$V_h^{\pi^*}(s_h) - \widehat{V}_h(s_h)$$

$$\leq \big\langle \mathbb{E}_{s' \sim \mathbb{P}_h(\cdot|s,a)}\big[V_{h+1}^{\pi^*}(s_{h+1}) - \widehat{V}_{h+1}(s_{h+1})\big] + \iota_h(s_h, \cdot), \pi^*(\cdot|s_h)\big\rangle_{\mathcal{A}}$$

$$- \big\langle \widehat{Q}_h(s_h, \cdot), \pi^*(\cdot|s_h) - \widehat{\pi}(\cdot|s_h)\big\rangle_{\mathcal{A}}.$$

**Proof** *By the Bellman equation, there is*

$$V_h^{\pi^*}(s_h) - \widehat{V}_h(s_h)$$

$$= \big\langle Q_h^{\pi^*}(s_h, \cdot), \pi^*(\cdot|s_h)\big\rangle_{\mathcal{A}} - \big\langle \widehat{Q}_h(s_h, \cdot), \widehat{\pi}(\cdot|s_h)\big\rangle_{\mathcal{A}}$$

$$= \big\langle Q_h^{\pi^*}(s_h, \cdot), \pi^*(\cdot|s_h)\big\rangle_{\mathcal{A}} - \big\langle \widehat{Q}_h(s_h, \cdot), \pi^*(\cdot|s_h)\big\rangle_{\mathcal{A}}$$

$$+ \big\langle \widehat{Q}_h(s_h, \cdot), \pi^*(\cdot|s_h)\big\rangle_{\mathcal{A}} - \big\langle \widehat{Q}_h(s_h, \cdot), \widehat{\pi}(\cdot|s_h)\big\rangle_{\mathcal{A}}.$$

*Then by rearranging the terms, we have*

$$V_h^{\pi^*}(s_h) - \widehat{V}_h(s_h)$$

$$= \big\langle Q_h^{\pi^*}(s_h, \cdot) - \widehat{Q}_h(s_h, \cdot), \pi^*(\cdot|s_h)\big\rangle_{\mathcal{A}} - \big\langle \widehat{Q}_h(s_h, \cdot), \pi^*(\cdot|s_h) - \widehat{\pi}(\cdot|s_h)\big\rangle_{\mathcal{A}}$$

$$= \big\langle \mathbb{B}_h V_h^{\pi^*}(s_h, \cdot) - \mathbb{B}_h \widehat{V}_h(s_h, \cdot) + \iota_h(s_h, \cdot), \pi^*(\cdot|s_h)\big\rangle_{\mathcal{A}} - \big\langle \widehat{Q}_h(s_h, \cdot), \pi^*(\cdot|s_h) - \widehat{\pi}(\cdot|s_h)\big\rangle_{\mathcal{A}}$$

$$\leq \big\langle \mathbb{E}_{s' \sim \mathbb{P}_h(\cdot|s,a)}\big[V_{h+1}^{\pi^*}(s_{h+1}) - \widehat{V}_{h+1}(s_{h+1})\big] + \iota_h(s_h, \cdot), \pi^*(\cdot|s_h)\big\rangle_{\mathcal{A}}$$

$$- \big\langle \widehat{Q}_h(s_h, \cdot), \pi^*(\cdot|s_h) - \widehat{\pi}(\cdot|s_h)\big\rangle_{\mathcal{A}},$$

*where the last inequality holds due to Lemma B.1. This completes the proof.*

**Lemma B.3** *Under the dynamic-OCE setting, we have*

$$V_1^*(s_1) - \widehat{V}_1(s_1) \leq \sum_{h=1}^{H} \mathbb{E}_{\pi*}\big[\iota(a_h, a_h)|s_1\big].$$

**Proof** *In order to prove this lemma, we first define*

$$\big(\mathbb{J}_h f\big)(s) = \big\langle f(s,\cdot), \pi^*(\cdot|s)\big\rangle_{\mathcal{A}}$$
$$\big(\mathbb{P}_h f\big)(s,a) = \mathbb{E}_{s'\sim\mathbb{P}_h(\cdot|s,a)}\big[f(s)\big].$$

*By recursively using Lemma B.2 and the previous definitions, there is*

$$V_1^*(s_1) - \widehat{V}_1(s_1)$$

$$\leq \Big(\prod_{h=1}^{H} \mathbb{J}_h\mathbb{P}_h\Big)\big(V_{H+1}^*(s_{H+1}) - \widehat{V}_{H+1}(s_{H+1})\big) + \sum_{h=1}^{H}\Big(\prod_{i=1}^{h-1}\mathbb{J}_h\mathbb{P}_h\Big)\big(\mathbb{J}_h\iota_h(s_h,a_h)\big)$$

$$+ \sum_{h=1}^{H}\Big(\prod_{i=1}^{h-1}\mathbb{J}_h\mathbb{P}_h\Big)\big\langle \widehat{Q}_h(s_h,\cdot), \pi^*(\cdot|s_h) - \widehat{\pi}(\cdot|s_h)\big\rangle_{\mathcal{A}}$$

$$= \sum_{h=1}^{H}\Big(\prod_{i=1}^{h-1}\mathbb{J}_h\mathbb{P}_h\Big)\big(\mathbb{J}_h\iota_h(s_h,a_h)\big) + \sum_{h=1}^{H}\Big(\prod_{i=1}^{h-1}\mathbb{J}_h\mathbb{P}_h\Big)\big\langle \widehat{Q}_h(s_h,\cdot), \pi^*(\cdot|s_h) - \widehat{\pi}(\cdot|s_h)\big\rangle_{\mathcal{A}}$$

$$= \sum_{h=1}^{H}\mathbb{E}_{\pi*}\big[\iota(a_h,a_h)\big|s_1\big] + \sum_{h=1}^{H}\mathbb{E}_{\pi*}\Big[\big\langle \widehat{Q}_h(s_h,\cdot), \pi^*(\cdot|s_h) - \widehat{\pi}(\cdot|s_h)\big\rangle_{\mathcal{A}}\Big|s_1\Big]$$

$$\leq \sum_{h=1}^{H}\mathbb{E}_{\pi*}\big[\iota(a_h,a_h)\big|s_1\big],$$

*where the first equation holds since $V_{H+1}^*(s_{H+1}) = \widehat{V}_{H+1}(s_{H+1}) = 0$ for any $s_{H+1} \in \mathcal{S}$; and the last inequality holds since $\widehat{\pi}(\cdot|s_h) = \arg\max_{\widehat{\pi}}\langle \widehat{Q}_h(s_h,\cdot), \widehat{\pi}(\cdot|s_h)\rangle_{\mathcal{A}}$ implies $\langle \widehat{Q}_h(s_h,\cdot), \pi^*(\cdot|s_h) - \widehat{\pi}(\cdot|s_h)\rangle_{\mathcal{A}} \leq 0$. This completes the proof.*

**Definition B.2** *With the dynamic-OCE setting, we define a new probability measure,*

$$\mathbb{C}_h(s'|s,a) = \mathbb{P}_h(s'|s,a)B_{h+1}(s'),$$

*where $B_{h+1}(s') \in \partial u(V_{h+1}^{\widehat{\pi}}(s') - b_{h+1})$, such that $\mathbb{E}_{s'\sim\mathbb{P}_h(s'|s,a)}[B_{h+1}(s')] = 1$. Due to the nondecreasing property of the utility function $u$, for any $s' \in \mathcal{S}$, $B_{h+1}(s') \geq 0$. This implies $\sum_{s'\in\mathcal{S}}\mathbb{C}_h(s'|s,a) = 1$.*

**Lemma B.4** *Under the dynamic-OCE setting, it always holds that*

$$\text{OCE}_{s'\sim\mathbb{P}_h(\cdot|s,a)}^{u}\Big\{\widehat{V}_{h+1}(s_{h+1})\Big\} - \text{OCE}_{s'\sim\mathbb{P}_h(\cdot|s,a)}^{u}\Big\{V_{h+1}^{\widehat{\pi}}(s_{h+1})\Big\}$$

$$\leq \mathbb{E}_{s'\sim\mathbb{C}_h(\cdot|s,a)}\Big[\widehat{V}_{h+1}(s_{h+1}) - V_{h+1}^{\widehat{\pi}}(s_{h+1})\Big],$$

*where $\mathbb{C}_h(\cdot|s,a)$ is a probability measure defined in Definition B.2.*

**Proof** *Setting $\widehat{b}_{h+1} = \arg\max_{b\in[0,H-h]}\{b + \mathbb{E}_{s'\sim\mathbb{P}_h(\cdot|s,a)}[\widehat{V}_{h+1}(s_{h+1}) - b]\}$ and $b_{h+1}^{\widehat{\pi}} = \arg\max_{b\in[0,H-h]}\{b + \mathbb{E}_{s'\sim\mathbb{P}_h(\cdot|s,a)}[V_{h+1}^{\widehat{\pi}}(s_{h+1}) - b]\}$, we have*

$$\text{OCE}_{s'\sim\mathbb{P}_h(\cdot|s,a)}^{u}\Big\{\widehat{V}_{h+1}(s_{h+1})\Big\} - \text{OCE}_{s'\sim\mathbb{P}_h(\cdot|s,a)}^{u}\Big\{V_{h+1}^{\widehat{\pi}}(s_{h+1})\Big\}$$

$$= \max_{b\in[0,H-h]}\Big\{b + \mathbb{E}_{s'\sim\mathbb{P}_h(\cdot|s,a)}\Big[u\Big(\widehat{V}_{h+1}(s_{h+1}) - b\Big)\Big]\Big\}$$

$$- \max_{b\in[0,H-h]}\Big\{b + \mathbb{E}_{s'\sim\mathbb{P}_h(\cdot|s,a)}\Big[u\Big(V_{h+1}^{\widehat{\pi}}(s_{h+1}) - b\Big)\Big]\Big\}$$

$$= \Big\{\widehat{b}_{h+1} + \mathbb{E}_{s'\sim\mathbb{P}_h(\cdot|s,a)}\Big[u\Big(\widehat{V}_{h+1}(s_{h+1}) - \widehat{b}_{h+1}\Big)\Big]\Big\}$$

$$- \Big\{b_{h+1}^{\widehat{\pi}} + \mathbb{E}_{s'\sim\mathbb{P}_h(\cdot|s,a)}\Big[u\Big(V_{h+1}^{\widehat{\pi}}(s_{h+1}) - b_{h+1}^{\widehat{\pi}}\Big)\Big]\Big\}$$

$$\leq \Big(\widehat{b}_{h+1} - b_{h+1}^{\widehat{\pi}}\Big)$$

$$+ \mathbb{E}_{s'\sim\mathbb{P}_h(\cdot|s,a)}\Big[B_{h+1}(s_{h+1})\Big(\widehat{V}_{h+1}(s_{h+1}) - V_{h+1}^{\widehat{\pi}}(s_{h+1}) - \Big(\widehat{b}_{h+1} - b_{h+1}^{\widehat{\pi}}\Big)\Big)\Big].$$

*Then, since the last inequality holds due to the concavity of $u(\cdot)$, which leads to the inequality $u(y) \leq u(x) + z(y - x)$, $z \in \partial u(x)$, we have*

$$\mathrm{OCE}^u_{s' \sim \mathbb{P}_h(\cdot|s,a)}\left\{\widehat{V}_{h+1}(s_{h+1})\right\} - \mathrm{OCE}^u_{s' \sim \mathbb{P}_h(\cdot|s,a)}\left\{V^{\widehat{\pi}}_{h+1}(s_{h+1})\right\}$$

$$= \left(1 - \mathbb{E}_{s' \sim \mathbb{P}_h(\cdot|s,a)}\left[B_{h+1}(s_{h+1})\right]\right)\left(\widehat{b}_{h+1} - b^{\widehat{\pi}}_{h+1}\right)$$

$$+ \mathbb{E}_{s' \sim \mathbb{P}_h(\cdot|s,a)}\left[B_{h+1}(s_{h+1})\left(\widehat{V}_{h+1}(s_{h+1}) - V^{\widehat{\pi}}_{h+1}(s_{h+1})\right)\right]$$

$$= \mathbb{E}_{s' \sim \mathbb{C}_h(\cdot|s,a)}\left[\widehat{V}_{h+1}(s_{h+1}) - V^{\widehat{\pi}}_{h+1}(s_{h+1})\right].$$

*The last equation holds because of Definition B.2 and the fact that $1 - \mathbb{E}_{s' \sim \mathbb{P}_h(\cdot|s,a)}\left[B_{h+1}(s_{h+1})\right] = 0$. This completes the proof.*

**Definition B.3** *Under the definition of dynamic-OCE and Definition B.2. We define a new state-action distribution,*

$$\omega_h(s_h, a_h) = \begin{cases} 1, & h = 1 \\ \mathbb{C}_1(s_2|s_1, a_1), & h = 2 \\ \displaystyle\sum_{s_2 \in \mathcal{S}} \sum_{s_3 \in \mathcal{S}} \cdots \sum_{s_{h-1} \in \mathcal{S}} \mathbb{C}_1(s_2|s_1, a_1)\mathbb{C}_2(s_3|s_2, a_2)\ldots\mathbb{C}_{h-1}(s_h|s_{h-1}, a_{h-1}), & h \geq 3, \end{cases}$$

*where $\mathbb{C}_h(\cdot|s, a)$ is a probability measure defined in Definition B.2*

**Lemma B.5** *Under the dynamic-OCE setting, we have*

$$\widehat{V}_1(s_1) - V^{\widehat{\pi}}_1(s_1) \leq \sum_{h=1}^{H} \mathbb{E}_{\omega_h}\left[-\iota_h(s_h, a_h)\big|s_1\right],$$

*where we slightly abuse the notation $\mathbb{E}_{(s_h, a_h) \sim \omega_h(\cdot, \cdot)}$ by $\mathbb{E}_{\omega_h}$.*

**Proof** *By the definition of $\widehat{V}_1(s_1)$ and $\widehat{Q}_1(s_1, a_1)$, we have*

$$\widehat{V}_1(s_1) - V^{\widehat{\pi}}_1(s_1)$$

$$\leq \widehat{Q}_1(s_1, a_1) - Q^{\widehat{\pi}}_1(s_1, a_1)$$

$$= \left(\widehat{\mathbb{B}}_1 \widehat{V}_2\right)(s_1, a_1) - \Gamma_1(s_1, a_1) - \left(\mathbb{B}_1 V^{\widehat{\pi}}_2\right)(s_1, a_1)$$

$$= \left(\widehat{\mathbb{B}}_1 \widehat{V}_2\right)(s_1, a_1) - \left(\mathbb{B}_1 \widehat{V}_2\right)(s_1, a_1) + \left(\mathbb{B}_1 \widehat{V}_2\right)(s_1, a_1) - \left(\mathbb{B}_1 V^{\widehat{\pi}}_2\right)(s_1, a_1) - \Gamma_1(s_1, a_1)$$

$$= \left(\widehat{\mathbb{B}}_1 \widehat{V}_2\right)(s_1, a_1) - \left(\mathbb{B}_1 \widehat{V}_2\right)(s_1, a_1) + \mathbb{B}_1\left(\widehat{V}_2 - V^{\widehat{\pi}}_2\right)(s_1, a_1) - \Gamma_1(s_1, a_1),$$

*where the first inequality holds because of $a_1 = \arg\max_{a \in \mathcal{A}} \widehat{Q}_1(s_1, a)$ such that $V^{\widehat{\pi}}_1(s_1) = \max_{a \in \mathcal{A}} Q^{\widehat{\pi}}_1(s_1, a) \geq Q^{\widehat{\pi}}_1(s_1, a_1)$. Then by plugging in the definition of $\mathbb{B}_h$,*

$$\widehat{V}_1(s_1) - V^{\widehat{\pi}}_1(s_1)$$

$$\leq \left(\widehat{\mathbb{B}}_1 \widehat{V}_2\right)(s_1, a_1) - \left(\mathbb{B}_1 \widehat{V}_2\right)(s_1, a_1)$$

$$+ \mathrm{OCE}^u_{s' \sim \mathbb{P}_1(\cdot|s,a)}\left\{\widehat{V}_2(s_2)\right\} - \mathrm{OCE}^u_{s' \sim \mathbb{P}_1(\cdot|s,a)}\left\{V^{\widehat{\pi}}_2(s_2)\right\} - \Gamma_1(s_1, a_1)$$

$$\leq \left(\widehat{\mathbb{B}}_1 \widehat{V}_2\right)(s_1, a_1) - \left(\mathbb{B}_1 \widehat{V}_2\right)(s_1, a_1) + \mathbb{E}_{s' \sim \mathbb{C}_h(\cdot|s,a)}\left[\widehat{V}_2(s_2) - V^{\widehat{\pi}}_2(s_2)\right] - \Gamma_1(s_1, a_1).$$

*The last inequality holds based on Lemma B.4. By recursively using Equation B.5, based on Definition B.3, Equation 7 and the fact that $\widehat{V}_{H+1}(s) = V^{\widehat{\pi}}_{H+1}(s) = 0$ for any $s \in \mathcal{S}$, we finish the proof of Lemma B.5. This completes the proof.*

**Lemma B.6** *Under event $\mathcal{E}_h$, for all $s \in \mathcal{S}$, $a \in \mathcal{A}$, and $h \in [H]$, we have*

$$\iota_h(s, b, a) \geq 0.$$

**Proof** *If $\overline{Q}_h(s,a) < 0$, by the definition of $\widehat{Q}_h(s,a)$ in Algorithm 3, we have*

$$\widehat{Q}(s,a) = \max\Big\{\min\big\{\overline{Q}_h(\cdot,\cdot), H-h+1\big\}, 0\Big\} = 0.$$

*This leads to*

$$\iota_h(s,a) = \mathbb{B}_h\widehat{V}_{h+1}(s) \geq 0.$$

*If $\overline{Q}_h(s,a) \geq 0$, we have*

$$\widehat{Q}(s,a) = \max\Big\{\min\big\{\overline{Q}_h(\cdot,b,\cdot), H-h+1\big\}, 0\Big\} \leq \overline{Q}_h(s,a).$$

*Then, we have*

$$\iota_h(s,b,a) \geq \mathbb{B}_h\widehat{V}_{h+1}(s) - \Big(\widehat{\mathbb{B}}_h\widehat{V}_{h+1}(s) - \Gamma_h(s,a)\Big) \geq 0,$$

*where the second inequality holds following the definition of $\mathcal{E}_h$. Therefore, we complete the proof of Lemma B.6.*

By Lemma B.7, the upper bound of suboptimality depends on bonus $\Gamma_h$ and $\mathbb{B}_h\widehat{V}_{h+1}(s,a) - \widehat{\mathbb{B}}_h\widehat{V}_{h+1}(s,a)$.

**Lemma B.7** *With probability at least $1 - \delta$ and the dynamic-OCE setting, there is*

$$\Big|\mathbb{B}_h\widehat{V}_{h+1}(s,a) - \widehat{\mathbb{B}}_h\widehat{V}_{h+1}(s,a)\Big| \leq \Gamma_h(s,a), \forall h \in [H],$$

*where $\{\Gamma_h\}_{h=1}^H$ is the bonus. Then we have the suboptimal of Algorithm 1 and Algorithm 3 bounded by*

$$\mathrm{SubOpt}_{\mathrm{D}}(\widehat{\pi}) \leq \sum_{h=1}^H \mathbb{E}_{\pi^*}\big[\iota_h(s_h, a_h)\big|s_1\big],$$

*where $\mathbb{E}_{\pi^*}$ is based on trajectory generated by $\pi^*$.*

Notice that Lemma B.7 holds for both tabular and linear function approximation settings.

**Proof** *Based on the definition of suboptimality, we can prove this lemma by*

$$\begin{aligned}
\mathrm{SubOpt}_{\mathrm{D}}(\widehat{\pi}) =& V_1^*(s_1) - V_1^{\widehat{\pi}}(s_1) \\
=& V_1^*(s_1) - \widehat{V}_1(s_1) + \widehat{V}_1(s_1) - V_1^{\widehat{\pi}}(s_1) \\
\leq& \sum_{h=1}^H \mathbb{E}_{\pi^*}\big[\iota(a_h, a_h)\big|s_1\big] + \sum_{h=1}^H \mathbb{E}_{\omega_h}\big[-\iota_h(s_h, a_h)\big|s_1\big] \\
\leq& \sum_{h=1}^H \mathbb{E}_{\pi^*}\big[\iota(a_h, a_h)\big|s_1\big],
\end{aligned}$$

*where the first inequality holds due to Lemma B.3 and Lemma B.5, and the last inequality holds due to Lemma B.6 guarantees $\sum_{h=1}^H \mathbb{E}_{\omega_h}\big[-\iota_h(s_h, a_h)\big|s_1\big] \leq 0$. Here we finish the proof of Lemma B.7.*

**Lemma B.8** *For any $\delta \in (0,1)$, any $(s,a) \in \mathcal{S} \times \mathcal{A}$, $h \in [H]$, and $b^* \in [0, H-h]$, with probability at least $1 - \delta$, the following inequality holds that*

$$\mathbb{E}_{s' \sim \mathbb{P}_h(\cdot|s,a)}\Big[u(\widehat{V}_{h+1}(s_{h+1}) - b^*)\Big] - \mathbb{E}_{s' \sim \widehat{\mathbb{P}}_h(\cdot|s,a)}\Big[u(\widehat{V}_{h+1}(s_{h+1}) - b^*)\Big]$$

$$\leq u(H-h)\sqrt{\frac{2\log\frac{|\mathcal{S}||\mathcal{A}|HK}{\delta}}{\max\{1, N_h(s,a)\}}}.$$

**Proof** *When $N_h(s,a) = 0$, we have*

$$\mathbb{E}_{s' \sim \mathbb{P}_h(\cdot|s,a)}\Big[u\big(\widehat{V}_{h+1}(s_{h+1}) - b^*\big)\Big] - \mathbb{E}_{s' \sim \widehat{\mathbb{P}}_h(\cdot|s,a)}\Big[u\big(\widehat{V}_{h+1}(s_{h+1}) - b^*\big)\Big]$$

$$\leq u(H-h-b^*) - u(-b^*)$$

$$\leq \big[u(H-h-b^*) - u(-b^*)\big]\sqrt{\frac{2\log\frac{|\mathcal{S}||\mathcal{A}|HK}{\delta}}{\max\{1, N_h(s,a)\}}},$$

*where the first inequality holds since the utility function $u$ is nondecreasing and $\widehat{V}_{h+1}(s) \in [0, H - h]$. The last inequality holds due to the fact that $\log \frac{|\mathcal{S}||\mathcal{A}|HK}{\delta} > 1$. When $N_h(s,a) \geq 1$, we have*

$$\mathbb{E}_{s' \sim \widehat{\mathbb{P}}_h(\cdot|s,a)}\left[u\left(\widehat{V}_{h+1}(s_{h+1}) - b^*\right)\right]$$

$$= \frac{1}{N_h(s,a)} \sum_{k=1}^{K} \mathbb{I}\left((s_h^k, a_h^k) = (s,a)\right) u\left(\widehat{V}_{h+1}(s_{h+1}^k) - b^*\right).$$

*Then by setting*

$$X_i = \mathbb{E}\left[\mathbb{I}\left((s_h^k, a_h^k) = (s,a)\right) u\left(\widehat{V}_{h+1}(s_{h+1}^k) - b^*\right)\right]$$

$$- \mathbb{I}\left((s_h^k, a_h^k) = (s,a)\right) u\left(\widehat{V}_{h+1}(s_{h+1}^k) - b^*\right),$$

*we have*

$$|X_i| \leq u(H - h - b^*) - u(-b^*),$$

*since $\widehat{V}_{h+1}(s) \in [0, H - h]$. And it is evident that for any $i \neq j$, $X_i$ and $X_j$ are independent. Therefore, with Hoeffding's inequality, with probability at least $1 - \frac{\delta}{|\mathcal{S}||\mathcal{A}|HK}$, we have*

$$\sum_{i=1}^{K} X_i$$

$$= N_h(s,a)\mathbb{E}_{s' \sim \mathbb{P}_h(\cdot|s,a)}\left[u\left(\widehat{V}_{h+1}(s_{h+1}) - b^*\right)\right] - \sum_{k=1}^{K} \mathbb{I}\left((s_h^k, a_h^k) = (s,a)\right) u\left(\widehat{V}_{h+1}(s_{h+1}^k) - b^*\right)$$

$$= N_h(s,a)\mathbb{E}_{s' \sim \mathbb{P}_h(\cdot|s,a)}\left[u\left(\widehat{V}_{h+1}(s_{h+1}) - b^*\right)\right] - N_h(s,a)\mathbb{E}_{s' \sim \widehat{\mathbb{P}}_h(\cdot|s,a)}\left[u\left(\widehat{V}_{h+1}(s_{h+1}) - b^*\right)\right]$$

$$\leq \left[u(H - h - b^*) - u(-b^*)\right]\sqrt{2N_h(s,a)\log \frac{|\mathcal{S}||\mathcal{A}|HK}{\delta}}.$$

*Therefore, we can conclude the following result with probability at least $1 - \frac{\delta}{|\mathcal{S}||\mathcal{A}|HK}$,*

$$\mathbb{E}_{s' \sim \mathbb{P}_h(\cdot|s,a)}\left[u\left(\widehat{V}_{h+1}(s_{h+1}) - b^*\right)\right] - \mathbb{E}_{s' \sim \widehat{\mathbb{P}}_h(\cdot|s,a)}\left[u\left(\widehat{V}_{h+1}(s_{h+1}) - b^*\right)\right]$$

$$\leq \left[u(H - h - b^*) - u(-b^*)\right]\sqrt{\frac{2\log \frac{|\mathcal{S}||\mathcal{A}|HK}{\delta}}{\max\{1, N_h(s,a)\}}}.$$

*This completes the proof of Lemma B.8.*

**Lemma B.9** *For any $\delta \in (0,1)$, any $(s,a) \in \mathcal{S} \times \mathcal{A}$, and $h \in [H]$, with probability at least $1 - \delta$, the following inequality holds that*

$$r_h(s,a) - \widehat{r}_h(s,a) \leq \sqrt{\frac{1}{\max\{1, N_h(s,a)\}}}.$$

**Proof** *When $N_h(s,a) = 0$, $\widehat{r}_h(s,a) = 0$, we have*

$$r_h(s,a) - \widehat{r}_h(s,a)$$

$$= r_h(s,a)$$

$$\leq 1$$

$$= \sqrt{\frac{1}{\max\{1, N_h(s,a)\}}},$$

*where the first inequality holds since $r_h(s,a) \in [0,1]$. The last inequality holds due to the fact that $\log \frac{|\mathcal{S}||\mathcal{A}|HK}{\delta} > 1$. When $N_h(s,a) \geq 1$, we have*

$$r_h(s,a) - \widehat{r}_h(s,a) = 0 \leq \sqrt{\frac{1}{\max\{1, N_h(s,a)\}}}.$$

*Then we complete the proof of Lemma B.9.*

## B.2 PROOF OF THEOREM 3.1

With Lemma B.7, we need to bound $\mathbb{B}_h \widehat{V}_{h+1}(s,a) - \widehat{\mathbb{B}}_h \widehat{V}_{h+1}(s,a)$, considering the definition of $\iota_h$. Based on the setting of dynamic-OCE, we have,

$$\mathbb{B}_h \widehat{V}_{h+1}(s,a) - \widehat{\mathbb{B}}_h \widehat{V}_{h+1}(s,a)$$

$$= r_h(s,a) - \widehat{r}_h(s,a) + \text{OCE}^u_{s' \sim \mathbb{P}_h(\cdot|s,a)}\left\{\widehat{V}_{h+1}(s_{h+1})\right\} - \text{OCE}^u_{s' \sim \widehat{\mathbb{P}}_h(\cdot|s,a)}\left\{\widehat{V}_{h+1}(s_{h+1})\right\}$$

$$= r_h(s,a) - \widehat{r}_h(s,a) + \max_{b \in [0, H-h]}\left\{b + \mathbb{E}_{s' \sim \mathbb{P}_h(\cdot|s,a)}\left[u\left(\widehat{V}_{h+1}(s_{h+1}) - b\right)\right]\right\}$$

$$\quad - \max_{b \in [0, H-h]}\left\{b + \mathbb{E}_{s' \sim \widehat{\mathbb{P}}_h(\cdot|s,a)}\left[u\left(\widehat{V}_{h+1}(s_{h+1}) - b\right)\right]\right\}$$

$$\leq r_h(s,a) - \widehat{r}_h(s,a) + \mathbb{E}_{s' \sim \mathbb{P}_h(\cdot|s,a)}\left[u\left(\widehat{V}_{h+1}(s_{h+1}) - b^*\right)\right]$$

$$\quad - \mathbb{E}_{s' \sim \widehat{\mathbb{P}}_h(\cdot|s,a)}\left[u\left(\widehat{V}_{h+1}(s_{h+1}) - b^*\right)\right],$$

where the first inequality holds when $b^* = \arg\max_{b \in [0, H-h]}\{b + \mathbb{E}_{s' \sim \mathbb{P}_h(\cdot|s,a)}[u(\widehat{V}_{h+1}(s_{h+1}) - b)]\}$. Then based on Lemma B.8 and Lemma B.9, with probability at least $1 - \delta$, we have

$$\mathbb{B}_h \widehat{V}_{h+1}(s,a) - \widehat{\mathbb{B}}_h \widehat{V}_{h+1}(s,a)$$

$$\leq \sqrt{\frac{1}{\max\{1, N_h(s,a)\}}} + \left[u(H - h - b^*) - u(-b^*)\right]\sqrt{\frac{2\log\frac{|\mathcal{S}||\mathcal{A}|HK}{\delta}}{\max\{1, N_h(s,a)\}}}$$

$$\leq \sqrt{\frac{1}{\max\{1, N_h(s,a)\}}} + \left[u(H - h) - u(h - H)\right]\sqrt{\frac{2\log\frac{|\mathcal{S}||\mathcal{A}|HK}{\delta}}{\max\{1, N_h(s,a)\}}}.$$

Therefore we succeed to upper bound $\mathbb{B}_h \widehat{V}_{h+1}(s,a) - \widehat{\mathbb{B}}_h \widehat{V}_{h+1}(s,a)$. Then we can obtain

$$\text{SubOpt}_{\text{D}}(\widehat{\pi})$$

$$\leq 2\sum_{h=1}^{H}\left[u(H-h) - u(h-H)\right]\mathbb{E}_{\pi^*}\left[\sqrt{\frac{1}{\max\{1, N_h(s,a)\}}} + \sqrt{\frac{2\log\frac{|\mathcal{S}||\mathcal{A}|HK}{\delta}}{\max\{1, N_h(s,a)\}}}\middle| s_1, b_1^*\right]$$

$$\leq 2\sum_{h=1}^{H}\left[u(H-h) - u(h-H)\right]\sum_{s,a} d_h^{\pi^*}(s,a)\left(\sqrt{\frac{1}{\max\{1, N_h(s,a)\}}} + \sqrt{\frac{2\log\frac{|\mathcal{S}||\mathcal{A}|HK}{\delta}}{\max\{1, N_h(s,a)\}}}\right)$$

$$= 2\sum_{h=1}^{H}\left[u(H-h) - u(h-H)\right]\sum_{s,a}\sqrt{d_h^{\pi^*}(s,a)}\left(\sqrt{\frac{d_h^{\pi^*}(s,a)}{K d_h^{\mu}(s,a)}} + \sqrt{\frac{2 d_h^{\pi^*}(s,a)\log\frac{|\mathcal{S}||\mathcal{A}|HK}{\delta}}{K d_h^{\mu}(s,a)}}\right).$$

Due to the fact that $\frac{d_h^{\pi^*}(s,a)}{d_h^{\mu}(s,a)} \leq C^*$, we have

$$\text{SubOpt}_{\text{D}}(\widehat{\pi})$$

$$\leq 2\sum_{h=1}^{H}\sum_{s,a}\sqrt{d_h^{\pi^*}(s,a)}\left(\sqrt{\frac{C^*}{K}} + \left[u(H-h) - u(h-H)\right]\sqrt{\frac{2C^*\log\frac{|\mathcal{S}||\mathcal{A}|HK}{\delta}}{K}}\right)$$

$$= 2\sum_{h=1}^{H}\sum_{s,a}\sqrt{d_h^{\pi^*}(s,a)\cdot\mathbb{I}(a = a_s^*)}\left(\sqrt{\frac{C^*}{K}} + \left[u(H-h) - u(h-H)\right]\sqrt{\frac{2C^*\log\frac{|\mathcal{S}||\mathcal{A}|HK}{\delta}}{K}}\right)$$

$$\leq 2\sum_{h=1}^{H}\sqrt{\sum_{s,a} d_h^{\pi^*}(s,a)\cdot\sum_{s,a}\mathbb{I}(a = a_s^*)}\left(\sqrt{\frac{C^*}{K}} + \left[u(H-h) - u(h-H)\right]\sqrt{\frac{2C^*\log\frac{|\mathcal{S}||\mathcal{A}|HK}{\delta}}{K}}\right)$$

$$= 2\sum_{h=1}^{H}\left(\sqrt{\frac{C^* S}{K}} + \left[u(H-h) - u(h-H)\right]\sqrt{\frac{2C^* S\log\frac{|\mathcal{S}||\mathcal{A}|HK}{\delta}}{K}}\right),$$

where $a_s^*$ is sampled by $a_s^* \sim \pi^*(\cdot|s)$. Here we finish the proof.

## B.3 Lemmas for Theorem 3.2

For the dynamic-OCE formulation, an additional advantage is its natural compatibility with stochastic rewards in risk-sensitive RL, which makes it both more practical and more general. Motivated by this, we extend the setting to stochastic reward functions where $r_h \sim \mathcal{R}(\cdot|s,a)$ in the proof. When $\mathcal{R}(r_h|s,a) = 1$, the problem degenerates to the deterministic reward case introduced in the paper. Therefore, in this section we provide a more general proof, which is an extension of Theorem 3.2. We first define the estimated error of the Bellman equation with stochastic reward at step $h$ for any $s$, $a$, and $b$,

$$\iota_h(s,b,a) = \mathbb{E}_{s'\sim\mathbb{P}_h(\cdot|s,a),r\sim\mathcal{R}_h(\cdot|s,a)}\Big[\widehat{V}_{h+1}(s',b-r)\Big] - \widehat{Q}_h(s,b,a). \tag{8}$$

In order to simplify the notations, we slightly abuse $\widehat{\mathbb{E}}_{s',r} := \mathbb{E}_{s'\sim\widehat{\mathbb{P}}_h(\cdot|s,a),r\sim\widehat{\mathcal{R}}_h(\cdot|s,a)}[\widehat{V}_{h+1}(s',b-r)]$. Then, we define an event in order to upper-bound the suboptimality.

**Definition B.4** *Define an event $\mathcal{E}'_h$,*

$$\mathcal{E}'_h = \Big\{ \Big|\mathbb{E}_{s_{h+1},r_h}\widehat{V}_{h+1}(s,b,a) - \widehat{\mathbb{E}}_{s_{h+1},r_h}\widehat{V}_{h+1}(s,b,a)\Big| \le \Gamma_h(s,b,a) \Big\},$$

*where $\{\Gamma_h\}_{h=1}^H$ is the bonus, satisfies $\mathbb{P}(\bigcap_{h=1}^H \mathcal{E}'_h) \ge 1 - \delta$.*

Then we can start the proof.

**Lemma B.10** *By the definition of $\widehat{V}_h(s,b)$ and the static-OCE setting, we have*

$$V_1^\pi(s_1,b_1^*) - \widehat{V}_1(s_1,b_1^*)$$

$$= \sum_{h=1}^H \mathbb{E}_\pi\Big[\iota_h(s_h,b_h,a_h)\Big|s_1,b_1^*\Big] + \sum_{h=1}^H \mathbb{E}_\pi\Big[\Big\langle\widehat{Q}_h(s_h,b_h^*,\cdot),\pi(\cdot|s_h,b_h^*) - \widehat{\pi}(\cdot|s_h,b_h^*)\Big\rangle\Big|s_1,b_1^*\Big].$$

**Proof** *Letting $\Delta_h(s,b) = \langle\widehat{Q}_h(s_h,b_h^*,\cdot),\pi(\cdot|s_h,b_h^*) - \widehat{\pi}(\cdot|s_h,b_h^*)\rangle$, we have*

$$V_h^\pi(s_h,b_h^*) - \widehat{V}_h(s_h,b_h^*)$$

$$= \Big\langle Q_h^\pi(s_h,b_h^*,\cdot),\pi(\cdot|s_h,b_h^*)\Big\rangle - \Big\langle\widehat{Q}_h(s_h,b_h^*,\cdot),\widehat{\pi}(\cdot|s_h,b_h^*)\Big\rangle$$

$$= \Big\langle Q_h^\pi(s_h,b_h^*,\cdot),\pi(\cdot|s_h,b_h^*)\Big\rangle - \Big\langle\widehat{Q}_h(s_h,b_h^*,\cdot),\pi(\cdot|s_h,b_h^*)\Big\rangle$$

$$\quad + \Big\langle\widehat{Q}_h(s_h,b_h^*,\cdot),\pi(\cdot|s_h,b_h^*)\Big\rangle - \Big\langle\widehat{Q}_h(s_h,b_h^*,\cdot),\widehat{\pi}(\cdot|s_h,b_h^*)\Big\rangle$$

$$= \Big\langle Q_h^\pi(s_h,b_h^*,\cdot) - \widehat{Q}_h(s_h,b_h^*,\cdot),\pi(\cdot|s_h,b_h^*)\Big\rangle + \Big\langle\widehat{Q}_h(s_h,b_h^*,\cdot),\pi(\cdot|s_h,b_h^*) - \widehat{\pi}(\cdot|s_h,b_h^*)\Big\rangle$$

$$= \Big\langle\mathbb{E}_{s',r}\Big[V_h^\pi(s_h,b_h^*) - \widehat{V}_h(s_h,b_h^*)\Big] + \iota_h(s,b,\cdot),\pi(\cdot|s_h,b_h^*)\Big\rangle + \Delta_h(s,b).$$

*Therefore, we have*

$$V_h^\pi(s_h,b_h^*) - \widehat{V}_h(s_h,b_h^*)$$

$$= \Big\langle\mathbb{E}_{s',r}\Big[V_h^\pi(s_h,b_h^*) - \widehat{V}_h(s_h,b_h^*)\Big] + \iota_h(s,b,\cdot),\pi(\cdot|s_h,b_h^*)\Big\rangle + \Delta_h(s,b).$$

*Since $V_{H+1}^\pi(s_h,b_h^*) - \widehat{V}_{H+1}(s_h,b_h^*) = u(-b_{H+1}^*) - u(-b_{H+1}^*) = 0$, by recursively applying Equation B.10, we can get*

$$V_1^\pi(s_1,b_1^*) - \widehat{V}_1(s_1,b_1^*)$$

$$= \sum_{h=1}^H \mathbb{E}_\pi\Big[\iota_h(s_h,b_h,a_h)\Big|s_1,b_1^*\Big] + \sum_{h=1}^H \mathbb{E}_\pi\Big[\Big\langle\widehat{Q}_h(s_h,b_h^*,\cdot),\pi(\cdot|s_h,b_h^*) - \widehat{\pi}(\cdot|s_h,b_h^*)\Big\rangle\Big|s_1,b_1^*\Big].$$

*This completes the proof of Lemma B.10.*

**Lemma B.11** *Under the definitions of $V_1^*(s_1,b_1^*)$ and $\widehat{V}_1(s_1,b_1^*)$, it is always true that*

$$V_1^*(s_1,b_1^*) - \widehat{V}_1(s_1,b_1^*) \le \sum_{h=1}^H \mathbb{E}_{\pi^*}\Big[\iota_h(s_h,b_h,a_h)\Big|s_1,b_1^*\Big].$$

**Proof** *Using Lemma B.10 and the static-OCE setting, letting $\pi = \pi^*$, we have*

$$V_1^*(s_1, b_1^*) - \widehat{V}_1(s_1, b_1^*)$$

$$= \sum_{h=1}^{H} \mathbb{E}_{\pi^\cdot}\left[\iota_h(s_h, b_h, a_h)\big|s_1, b_1^*\right]$$

$$+ \sum_{h=1}^{H} \mathbb{E}_{\pi^\cdot}\left[\left\langle \widehat{Q}_h(s_h, b_h^*, \cdot), \pi^*(\cdot|s_h, b_h^*) - \widehat{\pi}(\cdot|s_h, b_h^*)\right\rangle\Big|s_1, b_1^*\right]$$

$$\leq \sum_{h=1}^{H} \mathbb{E}_{\pi^\cdot}\left[\iota_h(s_h, b_h, a_h)\big|s_1, b_1^*\right].$$

*The last inequality holds because of the definition of $\widehat{\pi} = \arg\max_{\pi}\{\langle \widehat{Q}_h(s_h, b_h^*, \cdot), \pi(\cdot|s_h, b_h^*)\rangle\}$ results in $\langle \widehat{Q}_h(s_h, b_h^*, \cdot), \pi^*(\cdot|s_h, b_h^*) - \widehat{\pi}(\cdot|s_h, b_h^*)\rangle \leq 0$. This completes the proof.*

**Lemma B.12** *Under event $\mathcal{E}_h'$, for all $s \in \mathcal{S}$, $a \in \mathcal{A}$, $b \in [0, 1]$, and $h \in [H]$, we have*

$$0 \leq \iota_h(s, b, a) \leq 2\Gamma_h(s, b, a).$$

**Proof** *If $\overline{Q}_h(s, b, a) < 0$, by the definition of $\widehat{Q}_h(s, b, a)$ in Algorithm 4, we obtain*

$$\widehat{Q}(s, b, a) = \max\left\{ \min\left\{\overline{Q}_h(\cdot, b, \cdot), u(H - h - b)\right\}, u(-b)\right\} = u(-b).$$

*Furthermore, this leads to*

$$\iota_h(s, b, a) = \mathbb{E}_{s_{h+1}, r_h}\left[\widehat{V}_{h+1}(s_{h+1}^k, b - r_h)\right] - u(-b) \geq 0,$$

*where the last inequality holds due to the fact that $\widehat{V}_{h+1}(s_{h+1}^k, b - r_h) \geq u(-b)$. If $\overline{Q}_h(s, b, a) \geq 0$, we have*

$$\widehat{Q}(s, b, a) = \max\left\{ \min\left\{\overline{Q}_h(\cdot, b, \cdot), u(H - h - b)\right\}, u(-b)\right\} \leq \overline{Q}_h(s, b, a).$$

*Then, we have*

$$\iota_h(s, b, a) \geq \mathbb{E}_{s_{h+1}, r_h}\left[\widehat{V}_{h+1}(s_{h+1}^k, b - r_h)\right] - \left(\widehat{\mathbb{E}}_{s_{h+1}, r_h}\left[\widehat{V}_{h+1}(s_{h+1}^k, b - r_h)\right] - \Gamma_h(s, b, a)\right)$$

$$\geq 0,$$

*where the second inequality holds, following the definition of $\mathcal{E}_h'$. Therefore, we complete the proof of $\iota_h(s, b, a) \geq 0$. Then we will prove the other half of the inequality. On event $\mathcal{E}'$, by triangle inequality we have*

$$\widehat{\mathbb{E}}_{s_{h+1}, r_h}\widehat{V}_{h+1}(s, b, a) - \Gamma_h(s, b, a) \leq \mathbb{E}_{s_{h+1}, r_h}\widehat{V}_{h+1}(s, b, a).$$

*This leads to*

$$\overline{Q}_h(s, b, a) = \widehat{\mathbb{E}}_{s_{h+1}, r_h}\widehat{V}_{h+1}(s, b, a) - \Gamma_h(s, b, a)$$

$$\leq \mathbb{E}_{s_{h+1}, r_h}\widehat{V}_{h+1}(s, b, a)$$

$$\leq u(H - h - b),$$

*where the last inequality holds because of the definition of $\widehat{V}_h$. Therefore, we have*

$$\widehat{Q} = \max\left\{ \min\left\{\overline{Q}_h(\cdot, b, \cdot), u(H - h - b)\right\}, u(-b)\right\}$$

$$= \max\left\{\overline{Q}_h(\cdot, b, \cdot), u(-b)\right\}$$

$$\geq \overline{Q}_h(s, b, a).$$

*Applying the definition of $\iota_h(s, b, a)$ in Equation 8, we have*

$$\iota_h(s, b, a) = \mathbb{E}_{s_{h+1}, r_h}\widehat{V}_{h+1}(s_{h+1}^k, b - r_h^k) - \widehat{Q}_h(s, b, a)$$

$$\leq \mathbb{E}_{s_{h+1}, r_h}\widehat{V}_{h+1}(s_{h+1}^k, b - r_h^k) - \overline{Q}_h(s, b, a)$$

$$= \mathbb{E}_{s_{h+1}, r_h}\widehat{V}_{h+1}(s_{h+1}^k, b - r_h^k) - \widehat{\mathbb{E}}_{s_{h+1}, r_h}\widehat{V}_{h+1}(s_{h+1}^k, b - r_h^k) + \Gamma(s, b, a)$$

$$\leq 2\Gamma_h(s, b, a).$$

*Therefore, on the event $\mathcal{E}_h'$, we finish to prove $0 \leq \iota_h(s, b, a) \leq 2\Gamma(s, b, a)$.*

By the definition of suboptimality for static-OCE RL and event $\mathcal{E}'_h$, we conclude the following lemma.

**Lemma B.13** *Under the static-OCE setting, with probability at least $1 - \delta$, there is*

$$\left| \mathbb{E}_{s_{h+1}, r_h} \widehat{V}_{h+1}(s, b, a) - \widehat{\mathbb{E}}_{s_{h+1}, r_h} \widehat{V}_{h+1}(s, b, a) \right| \leq \Gamma_h(s, b, a), \forall h \in [H],$$

*where $\{\Gamma_h\}_{h=1}^H$ is the bonus, Then the suboptimality of Algorithm 2 and Algorithm 4 can be bounded by*

$$\mathrm{SubOpt}_{\mathrm{S}}(\widehat{\pi}) \leq \sum_{h=1}^H \mathbb{E}_{\pi^*} \left[ \iota_h(s_h, b_h, a_h) \Big| s_1, b_1^* \right],$$

*where $\mathbb{E}_{\pi^*}$ is with respect to the trajectory generated by $\pi^*$.*

Lemma B.13 shows that the suboptimality is highly related to the estimated error of the Bellman equation, which includes $\mathbb{E}_{s_{h+1}, r_h} \widehat{V}_{h+1}(s, b, a) - \widehat{\mathbb{E}}_{s_{h+1}, r_h} \widehat{V}_{h+1}(s, b, a)$ and the bonus $\Gamma_h$. Again, the Lemma B.13 holds for both tabular and linear function approximation settings.

**Proof** $\mathrm{SubOpt}_{\mathrm{S}}(\widehat{\pi})$ *can be split into two terms. By setting $b_1^* = \arg\max\{b + V_1^*(s_1, b)\}$, we have*

$$\begin{aligned}
\mathrm{SubOpt}_{\mathrm{S}}(\widehat{\pi}) &= \mathrm{OCE}^u \big\{ R(\pi^*) \big\} - \mathrm{OCE}^u \big\{ R(\widehat{\pi}) \big\} \\
&= \sup_{b \in [0, H]} \Big\{ b + V_1^*(s_1, b) \Big\} - \sup_{b \in [0, H]} \Big\{ b + V_1^{\widehat{\pi}}(s_1, b) \Big\} \\
&\leq \Big\{ b_1^* + V_1^*(s_1, b_1^*) \Big\} - \Big\{ b_1^* + V_1^{\widehat{\pi}}(s_1, b_1^*) \Big\} \\
&= V_1^*(s_1, b_1^*) - V_1^{\widehat{\pi}}(s_1, b_1^*) \\
&= V_1^*(s_1, b_1^*) - \widehat{V}_1(s_1, b_1^*) + \widehat{V}_1(s_1, b_1^*) - V_1^{\widehat{\pi}}(s_1, b_1^*),
\end{aligned}$$

*where $R(\pi) = \sum_{h=1}^H r_h$ with policy $\pi$. By applying Lemma B.10 with $\pi = \widehat{\pi}$, we have*

$$\widehat{V}_1(s_1, b_1^*) - V_1^{\widehat{\pi}}(s_1, b_1^*) = - \sum_{h=1}^H \mathbb{E}_{\widehat{\pi}} \left[ \iota_h(s_h, b_h, a_h) \Big| s_1, b_1^* \right].$$

*Then by using Lemma B.11 and Lemma B.12, on $\mathcal{E}'_h$ at every step,*

$$\begin{aligned}
\mathrm{SubOpt}_{\mathrm{S}}(\widehat{\pi}) &\leq V_1^*(s_1) - V_1^{\widehat{\pi}}(s_1) \\
&= V_1^*(s_1) - \widehat{V}_1(s_1) + \widehat{V}_1(s_1) - V_1^{\widehat{\pi}}(s_1) \\
&\leq \sum_{h=1}^H \mathbb{E}_{\pi^*} \left[ \iota_h(s_h, b_h, a_h) \Big| s_1, b_1^* \right] - \sum_{h=1}^H \mathbb{E}_{\widehat{\pi}} \left[ \iota_h(s_h, b_h, a_h) \Big| s_1, b_1^* \right] \\
&\leq \sum_{h=1}^H \mathbb{E}_{\pi^*} \left[ \iota_h(s_h, b_h, a_h) \Big| s_1, b_1^* \right].
\end{aligned}$$

*Here we finish the proof of Lemma B.13.*

**Lemma B.14** *For any $\delta \in (0, 1)$, any $(s, b, a) \in \mathcal{S} \times \mathcal{A} \times \mathcal{N}_b$, and $h \in [H]$, with the probability at least $1 - \delta$, we have*

$$\mathbb{E}_h \widehat{V}_{h+1}(s, b, a) - \widehat{\mathbb{E}}_h \widehat{V}_{h+1}(s, b, a) \leq u(H - h - b) \sqrt{\frac{2 \log \frac{|\mathcal{S}||\mathcal{A}|HK}{\delta}}{\max\{1, N_h(s, a)\}}}.$$

**Proof** *When $N_h(s, a) = 0$, $\widehat{\mathbb{P}}_h(\cdot | s, a) = 0$, we have*

$$\begin{aligned}
&\mathbb{E}_h \widehat{V}_{h+1}(s, b, a) - \widehat{\mathbb{E}}_h \widehat{V}_{h+1}(s, b, a) \\
&= \mathbb{E}_h \widehat{V}_{h+1}(s, b, a) \\
&\leq u(H - h - b) \\
&\leq u(H - h - b) \sqrt{\frac{2 \log \frac{|\mathcal{S}||\mathcal{A}|HK}{\delta}}{\max\{1, N_h(s, a)\}}},
\end{aligned}$$

*where the first inequality holds since at each step $\widehat{V}_h(s,b)$ is upper bounded. And the last inequality above holds due to the fact that $\log\frac{|\mathcal{S}||\mathcal{A}|HK}{\delta} > 1$. When $N_h(s,a) \geq 1$, we have*

$$\widehat{\mathbb{E}}_h\widehat{V}_{h+1}(s,b,a) = \frac{1}{N_h(s,a)}\sum_{k=1}^{K}\mathbb{I}\Big((s_h^k,a_h^k) = (s,a)\Big)\widehat{V}_{h+1}(s_{h+1}^k,b).$$

*Setting $Y_i = \mathbb{E}[\mathbb{I}\big((s_h^k,a_h^k) = (s,a)\big)\widehat{V}_{h+1}(s_{h+1}^k,b)] - \mathbb{I}\big((s_h^k,a_h^k) = (s,a)\big)\widehat{V}_{h+1}(s_{h+1}^k,b)$, we have*

$$|Y_i| \leq u(H-h-b),$$

*which is due to the fact that $\widehat{V}_{h+1}(s) \in [0, H-h-b]$. And it is obvious that for any $i \neq j$, $Y_i$ and $Y_j$ are independent. Therefore, with Hoeffding's inequality, with probability at least $1 - \frac{\delta}{|\mathcal{S}||\mathcal{A}|HK}$, there is*

$$\begin{aligned}
\sum_{i=1}^{K}Y_i =& N_h(s,a)\mathbb{E}_h\widehat{V}_{h+1}(s,b,a) - \sum_{k=1}^{K}\mathbb{I}\Big((s_h^k,a_h^k) = (s,a)\Big)u\Big(\widehat{V}_{h+1}(s_{h+1}^k) - b^*\Big) \\
=& N_h(s,a)\mathbb{E}_h\widehat{V}_{h+1}(s,b,a) - N_h(s,a)\widehat{\mathbb{E}}_h\widehat{V}_{h+1}(s,b,a) \\
\leq& u(H-h-b)\sqrt{2N_h(s,a)\log\frac{|\mathcal{S}||\mathcal{A}|HK}{\delta}}.
\end{aligned}$$

*Therefore, we have*

$$\mathbb{E}_h\widehat{V}_{h+1}(s,b,a) - \widehat{\mathbb{E}}_h\widehat{V}_{h+1}(s,b,a) \leq u(H-h-b)\sqrt{\frac{2\log\frac{|\mathcal{S}||\mathcal{A}|HK}{\delta}}{\max\{1, N_h(s,a)\}}}.$$

*This completes the proof of Lemma B.14.*

## B.4  PROOF OF THEOREM 3.2

We extend the the setting from deterministic reward to stochastic reward, in order to give a more general result. Therefore the expectation of value function at step $h$ is not only related to $\mathbb{P}_h$ but also related to $\mathcal{R}_h$. Note that when $\mathcal{R}_h(r_h|s,a) = 1$ for all $h$, the proof reduce to the stochastic reward setting, $r = r_h(s,a)$, as we introduce in the paper. Based on Lemma B.13, we can bound the suboptimality gap of the policy $\widehat{\pi}$ by bounding $\mathbb{E}_{s'\sim\mathbb{P}_h,r\sim\mathcal{R}_h}[\widehat{V}_{h+1}(s,b)] - \mathbb{E}_{s'\sim\widehat{\mathbb{P}}_h,r\sim\widehat{\mathcal{R}}_h}[\widehat{V}_{h+1}(s,b)]$, which is equal to $\iota_h(s_h,b_h,a_h) - \Gamma_h(s_h,b_h,a_h)$. In the following proof, we will slightly abuse the notation by using $\mathbb{E}_h\widehat{V}_{h+1}(s,b,a)$ to denote $\mathbb{E}_{s'\sim\mathbb{P}_h,r\sim\mathcal{R}_h}[\widehat{V}_{h+1}(s,b)]$, and $\widehat{\mathbb{E}}_h\widehat{V}_{h+1}(s,b,a)$ to denote $\mathbb{E}_{s'\sim\widehat{\mathbb{P}}_h,r\sim\widehat{\mathcal{R}}_h}[\widehat{V}_{h+1}(s,b)]$. By Lemma B.14, we can conclude that with probability at least $1 - \delta$, we have

$$\mathbb{E}_h\widehat{V}_{h+1}(s,b,a) - \widehat{\mathbb{E}}_h\widehat{V}_{h+1}(s,b,a) \leq u(H-h-b)\sqrt{\frac{2\log\frac{|\mathcal{S}||\mathcal{A}|HK}{\delta}}{\max\{1, N_h(s,a)\}}}.$$

Here, we find the upper bound of $\mathbb{E}_h\widehat{V}_{h+1}(s,b,a) - \widehat{\mathbb{E}}_h\widehat{V}_{h+1}(s,b,a)$ successfully. Then, with Lemma B.13, we have

$$\begin{aligned}
\mathrm{SubOpt}_\mathrm{S}(\widehat{\pi}) \leq& 2\sum_{h=1}^{H}u(H-h-b_h)\mathbb{E}_{\pi^*}\Big[\sqrt{\frac{2\log\frac{|\mathcal{S}||\mathcal{A}|HK}{\delta}}{\max\{1, N_h(s,a)\}}}\Big|s_1, b_1^*\Big] \\
\leq& 2\sum_{h=1}^{H}u(H-h-b_h)\sum_{s,a}d_h^{\pi^*}(s,a)\sqrt{\frac{2\log\frac{|\mathcal{S}||\mathcal{A}|HK}{\delta}}{\max\{1, N_h(s,a)\}}} \\
=& 2\sum_{h=1}^{H}u(H-h-b_h)\sum_{s,a}\sqrt{d_h^{\pi^*}(s,a)}\sqrt{\frac{2d_h^{\pi^*}(s,a)\log\frac{|\mathcal{S}||\mathcal{A}|HK}{\delta}}{Kd_h^{\mu}(s,a)}}.
\end{aligned}$$

Because of the fact that $\frac{d_h^{\pi^*}(s,a)}{d_h^\mu(s,a)} \leq C^*$, we have

$$
\begin{aligned}
\mathrm{SubOpt}_\mathrm{S}(\widehat{\pi}) \leq & 2\sum_{h=1}^{H} u(H-h-b_h)\sum_{s,a}\sqrt{d_h^{\pi^*}(s,a)}\sqrt{\frac{2C^*\log\frac{|\mathcal{S}||\mathcal{A}|HK}{\delta}}{K}} \\
= & 2\sum_{h=1}^{H} u(H-h-b_h)\sum_{s,a}\sqrt{d_h^{\pi^*}(s,a)\cdot\mathbb{I}(a=a_s^*)}\sqrt{\frac{2C^*\log\frac{|\mathcal{S}||\mathcal{A}|HK}{\delta}}{K}} \\
\leq & 2\sum_{h=1}^{H} u(H-h-b_h)\sqrt{\sum_{s,a}d_h^{\pi^*}(s,a)\cdot\sum_{s,a}\mathbb{I}(a=a_s^*)}\sqrt{\frac{2C^*\log\frac{|\mathcal{S}||\mathcal{A}|HK}{\delta}}{K}} \\
\leq & 2\sum_{h=1}^{H} u(H-h)\sqrt{\frac{2C^*S\log\frac{|\mathcal{S}||\mathcal{A}|HK}{\delta}}{K}},
\end{aligned}
$$

where $a_s^*$ is sampled by $a_s^* \sim \pi^*(\cdot|s)$. This concludes the proof of Theorem 3.2.

## C  PROOFS FOR SECTION 4

In this section, we give the proof structure of Theorem 4.1. We first need to construct a hard case linear MDP $\mathcal{M}^\dagger$. Define an integer $C = \min\{\lfloor C^*\rfloor, |\mathcal{A}|\}$. Therefore, under this assumption, we have $C^* \geq 2$ and $2 < C < |\mathcal{A}|$, and $C, H, K, |\mathcal{S}|$ satisfies $K > \frac{1}{4}CH|\mathcal{S}|$. Then we can construct the MDP $\mathcal{M}^\dagger$ with $|\mathcal{S}| + 2$ possible states, $A$ possible actions, and $H$ steps. We define the MDP $\mathcal{M}_{a^{\cdot}}$ to be the MDP $\mathcal{M}^\dagger$ with a certian existing optimal action $a_{h,i}^* \in \mathcal{A}$ at step $h$ and state $s_i \in S$. Set there are $S$ so-called "bandit states" $s_1, s_2, \ldots, s_S$ and two absorbing states "good state" $s_g$ and "bad state" $s_b$. Then the state space is $\mathcal{S} = \{s_1, s_2, \ldots, s_S, s_g, s_b\}$ and we define the action space $\mathcal{A} = \{a_1, a_2, \ldots, a_{|\mathcal{A}|}\}$. Moreover, we sample the dataset uniformly, which indicates that $\mu_h(a_{h,i}^*|s_i) = \frac{1}{C}$. We set the $i$-th bandit state $s_i$ to have the following transition dynamics. The transition of the MDP $\mathcal{M}_{a^{\cdot}}$ is defined as follows,

$$
\begin{cases}
\mathbb{P}_h(s_i|s_i,a) = 1-2p, & \text{for all } a \in \mathcal{A} \\
\mathbb{P}_h(s_g|s_i,a) = \mathbb{P}_h(s_b|s_i,a) = p, & \text{for all } a \neq a_{h,i}^* \\
\mathbb{P}_h(s_g|s_i,a_{h,i}^*) = p+\tau & \text{for for all } h \in [1,H] \\
\mathbb{P}_h(s_b|s_i,a_{h,i}^*) = p-\tau & \text{for for all } h \in [1,H],
\end{cases} \tag{9}
$$

where $p \in (0, \frac{1}{2})$ and $\tau \in (0, p)$ are the parameters yet to be determined. $h \leq \overline{H}$, all the states are absorbing states. The transition of the absorbing states is defined as follows,

$$
\begin{cases}
\mathbb{P}_h(s_g|s_g,a) = 1, & \text{for all } a \in \mathcal{A} \\
\mathbb{P}_h(s_b|s_b,a) = 1, & \text{for all } a \in \mathcal{A} \\
\mathbb{P}_h(s_i|s_g,a) = 0, & \text{for all } i \in S, a \in \mathcal{A} \\
\mathbb{P}_h(s_i|s_b,a) = 0, & \text{for all } i \in S, a \in \mathcal{A}.
\end{cases} \tag{10}
$$

For any $\overline{H} \leq h \leq H$, where $\overline{H} \in [1, H]$ is an integer, and $a \in \mathcal{A}$, the reward function is defined as follows,

$$
\begin{cases}
r_h(s_i, a_{h,i}) = 0, & \text{For any } s_i \\
r_h(s_g, a_{h,i}) = 0, & \text{For any } 1 \leq h \leq \overline{H} \\
r_h(s_g, a_{h,i}) = 1, & \text{For any } \overline{H} \leq h \leq H \\
r_h(s_b, a_{h,i}) = 0, & \text{For any } 1 \leq h \leq H,
\end{cases} \tag{11}
$$

Therefore, for any bandit state $s_i$, we can have the illustration of the transition dynamics in Figure 2.

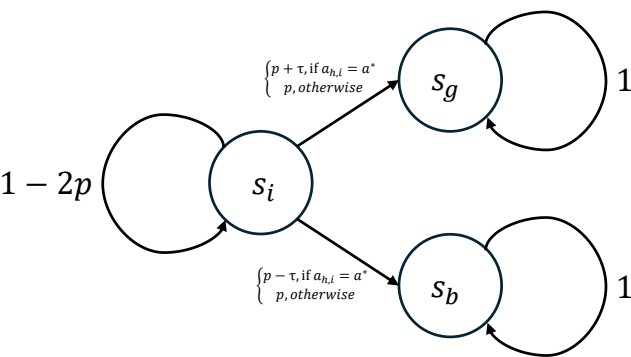

Figure 2: Transition of MDP $\mathcal{M}_{a^*}$.

In order to take as many as hard case into account, we need the MDP $\mathcal{M}^\dagger$ to be more general. Under this requirement, the construction consisting of bandit states and absorbing states is a wise choice. Since it is simple and can be extend to many other constructions. For example, if we take only one bandit state, we get the construction of Jin et al. (2021), and if we take add a tree structure before the bandit states, we can get the construction of Xu et al. (2023); if we set the total steps to be $2H$ and $\overline{H} = H$ we get the hard case of Xie et al. (2021). In another word, our construction of hard case MDP $\mathcal{M}^\dagger$ is probably general enough to cover many hard cases. Furthermore, we find that both kinds of OCE-PVI algorithms, the dynamic-OCE-PVI and the static-OCE-PVI, have the same information-theoretic lower bound under the hard case MDP $\mathcal{M}_{a^*}$. Intuitively, the first property of OCE, shown in Section A.1, makes the dynamic-OCE algorithm perform "true" OCE only once at a deterministic step, as long as we introduce the absorbing state setting. The theoretical proof can be found in the proof of Lemma C.1. Besides, we define the "null" MDP $\mathcal{M}_0$, that have the sane structure as $\mathcal{M}_{a^*}$, but with the transition dynamics defined as follows,

$$\mathbb{P}_h(s_g|s_i, a) = \mathbb{P}_h(s_b|s_i, a) = p, \quad \text{for all } a \in \mathcal{A}.$$

### C.1 LEMMAS FOR THEOREM 4.1

Under the constructed MDP $\mathcal{M}^\dagger$, we conclude Lemma C.1.

**Lemma C.1** *Under a constructed hard case MDP $\mathcal{M}^\dagger = (\mathcal{S}, \mathcal{A}, H, \mathbb{P}, r)$, where $\mathcal{S} = \{s_1, s_g, s_b\}$, $\mathcal{A} = \{x_1, x_2, \ldots, x_{|\mathcal{A}|}\}$, $H \in \mathbb{R}$, $\mathbb{P}$ is defined in Equation 9, Equation 10 and Equation 11, and $r \sim \mathcal{R}(\cdot|s, a)$ where $\mathcal{R}$ is defined in Equation 11. The suboptimality of Algorithm 3 and Algorithm 4 share the same form.*

**Proof** *By the hard case MDP defined in Lemma C.1, we discuss both the dynamic-OCE setting and the static-OCE setting.*

*For the dynamic-OCE setting, notice that for any constant policy $\pi$ and state $s$, we have $V_{H+1}^\pi(s) = 0$. Let the state transfer to $s_g$ or $s_b$ at a deterministic step $\overline{h} \leq H$, $h^* = \max\{\overline{h}, \overline{H}\}$. Then, based on the property of the OCE, we have*

$$V_{h^*+1}^\pi(s) = \begin{cases} H - h^*, \text{ wrp } p + \tau \\ 0, \text{ wrp } 1 - p - \tau. \end{cases}$$

*Then we have*

$$\begin{aligned} V_1^\pi(s_i) &= V_{h^*}^\pi(s) \\ &= OCE_{s \sim \mathbb{P}_{h^*}(\cdot|s_{h^*}, a_{h^*})}^u(V_{h^*+1}^\pi(s)) \\ &= \sup_{b \in [0, H-h^*]} \left\{ b + pu(H - h^* - b) + (1 - p)u(-b) \right\}. \end{aligned}$$

*Therefore, we obtain*

$$\text{SubOpt}_{\text{D}}(\mathcal{M}_{a^\cdot}, Algo(\mathcal{D}))$$

$$= \sup_{b \in [0, H - h^\cdot]} \left\{ b + (p + \tau)u(H - h^* - b) + (1 - p - \tau)u(-b) \right\}$$

$$- \sup_{b \in [0, H - h^\cdot]} \left\{ b + pu(H - h^* - b) + (1 - p)u(-b) \right\}$$

$$\geq \sup_{b \in [0, H - h^\cdot]} \left\{ b + (p + \tau)u(H - h^* - b) + (1 - p - \tau)u(-b) \right\}$$

$$- \sup_{b \in [0, H - h^\cdot]} \left\{ b + pu(H - \overline{H} - b) + (1 - p)u(-b) \right\}$$

$$\geq \sum_{h=1}^{H} \sum_{i=1}^{S} d_h(s_i)\tau \left[ u(H - \overline{H} - b_1^*) - u(-b_1^*) \right] \cdot \mathbb{I}\left\{ \widehat{a}_{h,i} \neq a_{h,i}^* \right\},$$

*where $\widehat{a}_{h,i} \sim \widehat{\pi}_h(\cdot|s_i)$ denotes the action sampled from the stochastic policy obtained by the algorithm. For the static-OCE setting, if $s_{h^\cdot+1} = s_g$, we have*

$$V_{H+1}^{\pi}(s_g, b) = u(-b), \textit{ for any } s, b$$

$$V_H^{\pi}(s_g, b_H) = u(1 - b)$$

$$\cdots$$

$$V_{h^\cdot+1}^{\pi}(s_g, b_{h^\cdot+1}) = u(H - h^* - b).$$

*And if $s_{h^\cdot+1} \neq s_g$, there is,*

$$V_{h^\cdot+1}^{\pi}(s_b, b_{h^\cdot+1}) = u(-b).$$

*Then we have*

$$V_1^{\pi}(s_i, b_1) = V_{h^\cdot}^{\pi}(s, b_{h^\cdot})$$

$$= \mathbb{E}[V_{h^\cdot+1}^{\pi}(s, b_{h^\cdot+1})]$$

$$= pu(H - h^* - b) + (1 - p)u(-b).$$

*Therefore, we can get*

$$OCE_u^{\pi} = \sup_{b \in [0, H]} \left\{ b + V_1^{\pi}(s_i, b) \right\}$$

$$= \sup_{b \in [0, H]} \left\{ b + pu(H - h^* - b) + (1 - p)u(-b) \right\}.$$

*Here we can find that with the hard case MDP we designed, $V_1^{\pi}(s_i)$ under the dynamic-OCE setting and $OCE_u^{\pi}$ under the static-OCE setting have the same form, which will lead to the same form of suboptimality. Similar to $V_1^{\pi}(s_i)$, we can find the form of $V_1^*(s_i)$ for the two settings,*

$$V_1^*(s_i) = \sup_{b \in [0, H - h^\cdot]} \left\{ b + (p + \tau)u(H - h^* - b) + (1 - p - \tau)u(-b) \right\}.$$

*Therefore, with $\widehat{a}_{h,i} \sim \widehat{\pi}_h(\cdot|s_i)$ we can obtain*

$$\text{SubOpt}_{\text{S}}(\mathcal{M}_{a^\cdot}, Algo(\mathcal{D}))$$

$$= \sup_{b \in [0, H - h^\cdot]} \left\{ b + (p + \tau)u(H - h^* - b) + (1 - p - \tau)u(-b) \right\}$$

$$- \sup_{b \in [0, H - h^\cdot]} \left\{ b + pu(H - h^* - b) + (1 - p)u(-b) \right\}$$

$$\geq \sup_{b \in [0, H - h^\cdot]} \left\{ b + (p + \tau)u(H - h^* - b) + (1 - p - \tau)u(-b) \right\}$$

$$- \sup_{b \in [0, H - h^\cdot]} \left\{ b + pu(H - \overline{H} - b) + (1 - p)u(-b) \right\}$$

$$\geq \sum_{h=1}^{H} \sum_{i=1}^{S} d_h(s_i)\tau \left[ u(H - \overline{H} - b_1^*) - u(-b_1^*) \right] \cdot \mathbb{I}\left\{ \widehat{a}_{h,i} \neq a_{h,i}^* \right\},$$

*where the first inequality holds due to the fact that $u(\cdot)$ is non-decreasing and $h^* \geq \overline{H}$, and last inequality holds when setting $b_1^* = \arg\max_{b \in [0, H - \overline{H}]} \{b + pu(H - \overline{H} - b) + (1 - p)u(-b)\}$. Therefore, we define*

$$\text{SubOpt}(\mathcal{M}_{a^{\cdot}}, Algo(\mathcal{D})) = \text{SubOpt}_{\text{D}}(\mathcal{M}_{a^{\cdot}}, Algo(\mathcal{D})) = \text{SubOpt}_{\text{S}}(\mathcal{M}_{a^{\cdot}}, Algo(\mathcal{D}))$$

*to be the shared form of suboptimality of Algorithm 1 and Algorithm 2 under the MDP $\mathcal{M}_{a^{\cdot}}$. This concludes the proof.*

**Lemma C.2** *For any $[a_{i,h}] \in \{1, 2, \ldots, K\}^{H|\mathcal{S}|}$, in MDP $\mathcal{M}_{a^{\cdot}}$, we have*

$$\sup_{h,s,a} \frac{d_h^{\pi^{\cdot}}(s,a)}{d_h^{\mu}(s,a)} < C \leq C^*,$$

*where $\pi^*$ is the optimal policy for the MDP $\mathcal{M}_{a^{\cdot}}$.*

**Proof** *Based on the definition of $d_h^{\pi^{\cdot}}(s_i)$ and $d_h^{\mu}(s_i)$, we have*

$$d_h^{\pi^{\cdot}}(s_i) = d_h^{\mu}(s_i) = (1 - 2p)^{h-1}.$$

*Also, we know that $\pi_h^*(a_{h,i}^*|s_i) = 1$, and we sample the dataset uniformly where $\mu_h(a_{h,i}^*|s_i) = \frac{1}{C}$. Then we have*

$$\frac{d_h^{\pi^{\cdot}}(s_i, a_{h,i}^*)}{d_h^{\mu}(s_i, a_{h,i}^*)} = \frac{d_h^{\pi^{\cdot}}(s_i)\pi_h^*(a_{h,i}^*|s_i)}{d_h^{\mu}(s_i)\mu_h(a_{h,i}^*|s_i)} = C.$$

*Then we consider the good state $s_g$ and the bad state $s_b$. There is*

$$d_h^{\pi^{\cdot}}(s_g) = \sum_S^i \sum_j^{h-1} \frac{1}{S}(1 - 2p)^{j-1}(p + \tau)$$

$$= \sum_j^{h-1}(1 - 2p)^{j-1}(p + \tau).$$

*The underlying policy $\mu$ takes the action $a_{h,i}^*$ with probability $\frac{1}{C}$, then we have*

$$d_h^{\mu}(s_g) = \sum_j^{h-1}(1 - 2p)^{j-1}\left[\frac{1}{C}(p + \tau) + (1 - \frac{1}{C})p\right]$$

$$= \sum_j^{h-1}(1 - 2p)^{j-1}\left[p + \frac{\tau}{C}\right].$$

*Therefore, we can conclude that*

$$\frac{d_h^{\pi^{\cdot}}(s_g)}{d_h^{\mu}(s_g)} = \frac{p + \tau}{p + \frac{\tau}{C}} = C\frac{p + \tau}{Cp + \tau} < C,$$

*where the last inequality holds since $\frac{p+\tau}{Cp+\tau} \leq 1$. Therefore for all the states including $s_i$, $s_g$ and $s_b$, we have*

$$\sup_{h,s,a} \frac{d_h^{\pi^{\cdot}}(s,a)}{d_h^{\mu}(s,a)} < C \leq C^*.$$

*Here we finish the proof of Lemma C.2.*

### C.2 PROOF OF THEOREM 4.1

For the certain MDP $\mathcal{M}$ defined in Theorem 4.1, we have the suboptimality of Algorithm 3 and Algorithm 4 as follows,

$$\text{SubOpt}\big(\mathcal{M}_{a^{\cdot}}, Alog(\mathcal{D})\big)$$

$$\geq \sum_{h=1}^{H}\sum_{i=1}^{|\mathcal{S}|} d_h(s_i)\tau\big[u(H - \overline{H} - b_1^*) - u(-b_1^*)\big] \cdot \mathbb{I}\{\widehat{a}_{h,i} \neq a_{h,i}^*\}$$

$$= \sum_{h=1}^{H}\sum_{i=1}^{|\mathcal{S}|} \frac{1}{|\mathcal{S}|}(1 - 2p)^{h-1}\tau\big[u(H - \overline{H} - b_1^*) - u(-b_1^*)\big] \cdot \mathbb{I}\{\widehat{a}_{h,i} \neq a_{h,i}^*\},$$

where $\widehat{a}_{h,i} \sim \widehat{\pi}_h(\cdot|s_i)$ and the last equation holds based on the definition of $d_h(s_i)$. Then we have

$$\max_{\mathcal{M}} \text{SubOpt}\big(\mathcal{M}, Algo(\mathcal{D})\big)$$

$$\geq \mathbb{E}_{\mathcal{M}} \text{SubOpt}\big(\mathcal{M}, Algo(\mathcal{D})\big)$$

$$\geq \frac{1}{C} \sum_{a^*_{h,i}=1}^{C} \mathbb{E}_{\mathcal{M}_{a^*}} \big[ \text{SubOpt}\big(\mathcal{M}_{a^*}, Algo(\mathcal{D})\big) \big]$$

$$= \frac{1}{C} \sum_{a^*_{h,i}=1}^{C} \mathbb{E}_{\mathcal{M}_{a^*}} \Big[ \sum_{h=1}^{H} \sum_{i=1}^{|\mathcal{S}|} \frac{1}{|\mathcal{S}|} (1-2p)^{h-1} \tau \big[ u(H - \overline{H} - b_1^*) - u(-b_1^*) \big] \cdot \mathbb{I}\big\{ \widehat{a}_{h,i} \neq a^*_{h,i} \big\} \Big]$$

$$= (1-2p)^{H-1} \tau \big[ u(H - \overline{H} - b_1^*) - u(-b_1^*) \big] \cdot \Big( H - \frac{1}{CK|\mathcal{S}|} \sum_{h=1}^{H} \sum_{i=1}^{|\mathcal{S}|} \sum_{a^*_{h,i}=1}^{C} \mathbb{E}_{\mathcal{M}_{a^*}} \big[ N_h(s_i, a^*_{h,i}) \big] \Big).$$

Then we need to bound the term $\sum_{h=1}^{H} \sum_{i=1}^{|\mathcal{S}|} \sum_{a^*_{h,i}=1}^{C} \frac{1}{K} \mathbb{E}_{\mathcal{M}_{a^*}} \big[ N_h(s_i, a^*_{h,i}) \big]$, where $N_h(s_i, a^*_{h,i})$ is the number of times that the action $a^*_{h,i}$ is selected at step $h$ and state $s_i$. Comparing $\mathbb{E}_{\mathcal{M}_{a^*}} \big[ N_h(s_i, a^*_{h,i}) \big]$ and $\mathbb{E}_{\mathcal{M}_0} \big[ N_h(s_i, a^*_{h,i}) \big]$, we have

$$\frac{1}{K} \mathbb{E}_{\mathcal{M}_{a^*}} \big[ N_h(s_i, a^*_{h,i}) \big] - \frac{1}{K} \mathbb{E}_{\mathcal{M}_0} \big[ N_h(s_i, a^*_{h,i}) \big]$$

$$\leq TV\big( \mathbb{P}_{\mathcal{M}_{a^*}}, \mathbb{P}_{\mathcal{M}_0} \big)$$

$$\leq \sqrt{ \frac{1}{2} KL\big( \mathbb{P}_{\mathcal{M}_0}, \mathbb{P}_{\mathcal{M}_{a^*}} \big) }$$

$$= \sqrt{ \frac{1}{2} p \log \frac{p^2}{p^2 - \tau^2} \mathbb{E}_{\mathcal{M}_0} \big[ N_h(s_i, a^*_{h,i}) \big] },$$

where the second inequality follows Pinsker's inequality. Then we have

$$\frac{1}{K} \sum_{h=1}^{H} \sum_{i=1}^{|\mathcal{S}|} \sum_{a^*_{h,i}=1}^{C} \mathbb{E}_{\mathcal{M}_{a^*}} \big[ N_h(s_i, a^*_{h,i}) \big]$$

$$\leq \sum_{h=1}^{H} \sum_{i=1}^{|\mathcal{S}|} \sum_{a^*_{h,i}=1}^{C} \left\{ \frac{1}{K} \mathbb{E}_{\mathcal{M}_0} \big[ N_h(s_i, a^*_{h,i}) \big] + \sqrt{ \frac{1}{2} p \log \frac{p^2}{p^2 - \tau^2} \mathbb{E}_{\mathcal{M}_0} \big[ N_h(s_i, a^*_{h,i}) \big] } \right\}$$

$$= H + \sum_{h=1}^{H} \sum_{i=1}^{|\mathcal{S}|} \sum_{a^*_{h,i}=1}^{C} \sqrt{ \frac{1}{2} p \log \frac{p^2}{p^2 - \tau^2} \mathbb{E}_{\mathcal{M}_0} \big[ N_h(s_i, a^*_{h,i}) \big] }$$

$$\leq H + H \sqrt{ \frac{1}{2} p \log \frac{p^2}{p^2 - \tau^2} CK|\mathcal{S}| },$$

where the first equation holds since $\sum_{h=1}^{H} \sum_{i=1}^{|\mathcal{S}|} \sum_{a^*_{h,i}=1}^{C} \mathbb{E}_{\mathcal{M}_0} \big[ N_h(s_i, a^*_{h,i}) \big] \leq HK$ and the last inequality holds because of Cauchy-Schwarz inequality. Therefore, we have

$$\max_{\mathcal{M}} \text{SubOpt}\big(\mathcal{M}, Algo(\mathcal{D})\big)$$

$$\geq H(1-2p)^{H-1} \tau \big[ u(H - \overline{H} - b_1^*) - u(-b_1^*) \big] \cdot \left( 1 - \frac{1}{C|\mathcal{S}|} - \frac{1}{C|\mathcal{S}|} \sqrt{ \frac{1}{2} p \log \frac{p^2}{p^2 - \tau^2} CK|\mathcal{S}| } \right)$$

$$\geq H(1-2p)^{H-1} \tau \big[ u(H - \overline{H} - b_1^*) - u(-b_1^*) \big] \cdot \left( \frac{1}{2} - \frac{1}{C|\mathcal{S}|} \sqrt{ \frac{\tau^2}{2p} CK|\mathcal{S}| } \right),$$

where the last inequality holds since $\frac{1}{C|\mathcal{S}|} \leq \frac{1}{2}$ and $\log \frac{p^2}{p^2 - \tau^2} \leq \frac{\tau^2}{p^2}$. Then, by setting $p = \frac{1}{2H}$ (It is reasonable to set $p = \frac{1}{2H}$, since $H \geq 1$ guarantees $p \in (0, \frac{1}{2})$), $\overline{H} = \lceil (1-\rho)H \rceil$, where $\rho \in (0,1)$.

Then we have

$$\max_{\mathcal{M}} \mathrm{SubOpt}\big(\mathcal{M}, Algo(\mathcal{D})\big)$$

$$\geq H\Big(1 - \frac{1}{H}\Big)^{H-1} \tau\big[u\big(\rho H - b_1^*\big) - u(-b_1^*)\big] \cdot \Big(\frac{1}{2} - \frac{1}{C|\mathcal{S}|}\sqrt{\tau^2 CHK|\mathcal{S}|}\Big)$$

$$\geq \frac{1}{3} H\tau\Big[u\big(\rho H - b_1^*\big) - u(-b_1^*)\Big] \cdot \Big(\frac{1}{2} - \sqrt{\frac{\tau^2 HK}{C|\mathcal{S}|}}\Big),$$

where the last inequality holds since $(1 - \frac{1}{H})^{H-1} \geq e^{-1} \geq \frac{1}{3}$. Let $\tau = \sqrt{\frac{C|\mathcal{S}|}{16HK}} < p = \frac{1}{2H}$, we have

$$\max_{\mathcal{M}} \mathrm{SubOpt}(\mathcal{M}, Algo(\mathcal{D}))$$

$$\geq \frac{1}{48}\sqrt{\frac{CH|\mathcal{S}|}{K}}\Big[u\big(\rho H - b_1^*\big) - u(-b_1^*)\Big]$$

$$\geq \frac{1}{48}\big[u\big(\rho H - b_1^*\big) - u(-b_1^*)\big]\sqrt{\frac{C^* H|\mathcal{S}|}{K}},$$

where the last inequality holds based on Lemma C.2 and $b_1^* = \arg\max_{b\in(0,\rho H)}\{b + \frac{1}{2H}u(\rho H - b) + (1 - \frac{1}{2H})u(-b)\}$. Let a function $F_b(b) = b + \frac{1}{2H}u(\rho H - b) + (1 - \frac{1}{2H})u(-b)$, we have

$$F_b'(b) = 1 - \frac{1}{2H}u'(\rho H - b) - (1 - \frac{1}{2H})u'(-b).$$

Based on the properties of the utility function $u(\cdot)$, we have $F_b'(0) > 0$ and $F_b'(\rho H) < 0$. Therefore, there exists a $b_1^* \in (0, \rho H)$ such that $F_b'(b_1^*) = 0$. This concludes the proof of Theorem 4.1.

## D   PROOFS FOR SECTION 5

### D.1   LEMMAS FOR THEOREM 5.1

**Lemma D.1** *Based on the dynamic-OCE RL setting, we have*

$$\Big\|\theta_h + w_h(b)\Big\| \leq [1 + u(H - h)]\sqrt{d}.$$

**Proof** *Based on the definition of $w_h(b)$ and the dynamic-OCE RL setting, we have*

$$\Big\|\theta_h + w_h(b)\Big\| \leq \Big\|\theta_h\Big\| + \Big\|\int_{\mathcal{S}} u(V(s') - b)\mu_h(s')ds'\Big\|$$

$$\leq \sqrt{d} + \int_{\mathcal{S}}\Big\|u(V(s') - b)\mu_h(s')\Big\|ds'$$

$$\leq [1 + u(H - h)]\sqrt{d}.$$

*The third inequality holds since $V(s) \in [0, H - h]$ and $b \in [0, H - h]$. This completes the proof.*

**Definition D.1** *Define a function class $\mathcal{V}$ mapping from $\mathcal{S} \times [0, H]$ to $\mathbb{R}$ with parametric form,*

$$V(\cdot) = \max_a \Big\{ \max \big\{ \min \big\{\phi(\cdot, a)^\top \theta + \sup_{b\in[0,H-h]}\{b + \phi(\cdot, a)^\top w(b)\} $$

$$- \beta\sqrt{\phi(\cdot, a)^\top \Lambda^{-1}\phi(\cdot, a)}, H - h + 1\big\}, 0\big\}\Big\},$$

*where $b \in [0, 1]$ is a parameter, $\|\theta\| \leq T$, $\|w(b)\| \leq L$, $\beta \in [0, B]$ and $\Lambda \succeq \lambda I$.*

**Lemma D.2** *Under the dynamic-OCE RL setting, we have*

$$\Big\|\widehat{w}_h(b)\Big\| \leq u(H - h)\sqrt{\frac{dK}{\lambda}}$$

$$\Big\|\widehat{\theta}_h\Big\| \leq \sqrt{\frac{dK}{\lambda}}.$$

**Proof** *Based on the definition of $\widehat{w}_h(b)$, we have*

$$\left\|\widehat{w}_h(b)\right\| = \left\|\Lambda_h^{-1}\Big\{\sum_{k=1}^{K}\phi(s_h^k,a_h^k)u\Big(\widehat{V}_{h+1}(s_{h+1}^k)-b\Big)\Big\}\right\|$$

$$\leq \sum_{k=1}^{K}\left\|\Lambda_h^{-1}\phi(s_h^k,a_h^k)u\Big(\widehat{V}_{h+1}(s_{h+1}^k)-b\Big)\right\|$$

$$\leq u(H-h)\sum_{k=1}^{K}\left\|\Lambda_h^{-1}\phi(s_h^k,a_h^k)\right\|$$

$$= u(H-h)\sum_{k=1}^{K}\sqrt{\phi(s_h^k,a_h^k)^{\top}\Lambda_h^{-\frac{1}{2}}\Lambda_h^{-1}\Lambda_h^{-\frac{1}{2}}\phi(s_h^k,a_h^k)}.$$

*Then, based on the Cauchy-Schwarz inequality and the property of the trajectory, we have*

$$\left\|\widehat{w}_h(b)\right\| \leq u(H-h)\sqrt{\frac{K}{\lambda}}\sqrt{\mathrm{Tr}\Big(\Lambda_h^{-1}\sum_{k=1}^{K}\phi(s_h^k,a_h^k)^{\top}\phi(s_h^k,a_h^k)\Big)}$$

$$= u(H-h)\sqrt{\frac{K}{\lambda}}\sqrt{\mathrm{Tr}\Big(\Lambda_h^{-1}(\Lambda_h-\lambda I)\Big)}$$

$$\leq u(H-h)\sqrt{\frac{K}{\lambda}}\sqrt{\mathrm{Tr}\Big(\Lambda_h^{-1}\Lambda_h\Big)}$$

$$= u(H-h)\sqrt{\frac{dK}{\lambda}}.$$

*Following the same method, we can prove $\left\|\widehat{\theta}_h\right\| \leq \sqrt{\frac{dK}{\lambda}}$ with $|r_h| \leq 1$. Then we complete the proof.*

**Lemma D.3** *Based on the dynamic-OCE setting, for a fixed function $f^h : \mathcal{S} \to [0, 1+u(H-h)]$ at step $h \in [H]$, under the assumption that $\mathcal{D}$ is obtained by an underlying policy $\mu$, for any $\Delta \in (0,1)$, we have*

$$\mathbb{P}_{\mathcal{D}}\left(\Big\|\sum_{K=1}^{K}\phi(s_h^k,a_h^k)\epsilon_h^k(V_h)\Big\|_{\Lambda_h^{-1}}^{2} > [1+u(H-h)]^2\Big(2\log\frac{1}{\Delta}+d\log\big(1+\frac{K}{\lambda}\big)\Big)\right) \leq \Delta.$$

**Proof** *For any fixed $h \in [H]$ and $k \in \{0,1,\dots,K\}$, we have the $\sigma$-algebra*

$$\mathcal{F}_h^k = \sigma\left(\big\{(s_h^j,a_h^j)\big\}_{j=1}^{\min\{k+1,K\}} \cup \big\{(r_h^j,s_{h+1}^j)\big\}_{j=1}^{k}\right).$$

*Then for any $k \in [K]$, we have $\phi(s_h^k,a_h^k) \in \mathcal{F}_h^k$, since $(s_h^k,a_h^k)$ is measurable with respect to $\mathcal{F}_h^{k-1}$. Then, with the fact that $(r_h^j,s_{h+1}^j)$ is measurable with respect to $\mathcal{F}_h^k$, for a fixed function $f^h : \mathcal{S} \to [0, 1+u(H-h)]$ at step $h$, and $k \in [K]$, we have*

$$\epsilon_h^k(f^h)$$
$$= r_h^k + u(f^h(s_{h+1}^k) - b_h) - \mathbb{B}_h f^h(s_h^k,a_h^k)$$
$$\in \mathcal{F}_h^k.$$

*Therefore, $\{\epsilon_h^k(f^h)\}_{k=1}^{K}$ is a stochastic process with respect to the filtration $\{\mathcal{F}_h^k\}_{k=0}^{K}$. Then with Assumption 2.1, we have*

$$\mathbb{E}_{\mathcal{D}}\Big[\epsilon_h^k(f^h)|\mathcal{F}_h^{k-1}\Big]$$
$$= \mathbb{E}_{\mathcal{D}}\Big[r_h^k + u(f^h(s_{h+1}^k)-b_h)|\{(s_h^j,a_h^j)\}_{j=1}^{k},(r_h^j,s_{h+1}^j)_{j=1}^{k}\Big] - \mathbb{B}_h f^h(s_h^k,a_h^k)$$
$$= 0.$$

Based on the definition of $\epsilon_h^k(f^h)$, we have $|\epsilon_h^k(f^h)| \le 1 + u(H-h)$. Thus, for the fixed $h$ and all $k \in [K]$, $\epsilon_h^k(f^h)$ is a zero-mean and $[1 + u(H-h)]$-sub-Gaussian random variable conditioning on $\mathcal{F}_h^{k-1}$. Based on Lemma E.1 with $M_0 = \lambda I$ and $M_k = \lambda I + \frac{1}{K}\sum_{j=1}^{K} \phi(s_h^j, a_h^j)\phi(s_h^j, a_h^j)^\top$, for all $\Delta \in (0,1)$, we have

$$\mathbb{P}_{\mathcal{D}}\left( \Big\| \sum_{k=1}^{K} \phi(s_h^k, a_h^k)\epsilon_h^k(f^h) \Big\|_{\Lambda_h^{-1}}^2 > 2[1+u(H-h)]^2 \log\Big( \frac{\det(\Lambda_h)^{\frac{1}{2}}}{\Delta \cdot \det(\lambda I)^{\frac{1}{2}}} \Big) \right) \le \Delta,$$

where the equation holds based on the fact that $M_K = \Lambda_h$. By applying the definition of $\Lambda_h$, we have $\|\Lambda_h\|_2 \le \lambda + K$ which implies $\det(\Lambda_h) \le (\lambda + K)^d$. Therefore, we can get

$$\mathbb{P}_{\mathcal{D}}\left( \Big\| \sum_{K=1}^{K} \phi(s_h^k, a_h^k)\epsilon_h^k(f^h) \Big\|_{\Lambda_h^{-1}}^2 > [1+u(H-h)]^2\Big( 2\log\frac{1}{\Delta} + d\log\big(1 + \frac{K}{\lambda}\big) \Big) \right)$$

$$\le \mathbb{P}_{\mathcal{D}}\left( \Big\| \sum_{k=1}^{K} \phi(s_h^k, a_h^k)\epsilon_h^k(V_h) \Big\|_{\Lambda_h^{-1}}^2 > 2[1+u(H-h)]^2 \log\Big( \frac{\det(\Lambda_h)^{\frac{1}{2}}}{\Delta \cdot \det(\lambda I)^{\frac{1}{2}}} \Big) \right)$$

$$\le \Delta$$

Here we finish the proof.

**Lemma D.4** *Based on Definition D.1, for all $h \in [H]$ and $\varepsilon > 0$, we have*

$$\log \mathcal{N}_h(\varepsilon) \le d\log\Big(1 + \frac{4T}{\varepsilon}\Big) + d\log\Big(1 + \frac{4L}{\varepsilon}\Big) + d^2 \log\Big(1 + \frac{8\sqrt{d}B^2}{\lambda \varepsilon^2}\Big).$$

**Proof** *For the function class $\mathcal{V}$, we set $A = \beta^2 \Lambda^{-1}$. Therefore, by the definition of function class $\mathcal{V}$, we have $\|\theta\| \le T$, $|w(b)\| \le L$ and $\|A\| \le \frac{B^2}{\lambda}$. Letting any two functions $V_1, V_2 \in \mathcal{V}$, it holds that*

$$\text{dist}(V_1, V_2)$$

$$\le \sup_{s,a} \left| \Big[\phi(\cdot, a)^\top \theta_1 + \sup_{b \in [0, H-h]}\big\{ b + \phi(\cdot, a)^\top w_1(b) \big\} - \sqrt{\phi(\cdot, a)^\top A_1 \phi(\cdot, a)} \Big] \right.$$

$$\left. - \Big[\phi(\cdot, a)^\top \theta_2 + \sup_{b \in [0, H-h]}\big\{ b + \phi(\cdot, a)^\top w_2(b) \big\} - \sqrt{\phi(\cdot, a)^\top A_2 \phi(\cdot, a)} \Big] \right|$$

$$\le \sup_{s,a} \left| \Big[\phi(\cdot, a)^\top \theta_1 + \big\{ b^\dagger + \phi(\cdot, a)^\top w_1(b) \big\} - \sqrt{\phi(\cdot, a)^\top A_1 \phi(\cdot, a)} \Big] \right.$$

$$\left. - \Big[\phi(\cdot, a)^\top \theta_2 + \big\{ b^\dagger + \phi(\cdot, a)^\top w_2(b) \big\} - \sqrt{\phi(\cdot, a)^\top A_2 \phi(\cdot, a)} \Big] \right|$$

$$\le \sup_{\phi:\|\phi\|\le 1, \theta:\|\theta\|\le T, w:\|w\|\le L} \left| \Big[(\theta_1 + w_1)\phi - \sqrt{\phi^\top A_1 \phi}\Big] - \Big[(\theta_2 + w_2)\phi - \sqrt{\phi^\top A_2 \phi}\Big] \right|,$$

*where the second inequality holds by setting $b^\dagger = \arg\max_{b \in [0, H-h]}\{b + \phi(\cdot, a)^\top w_1(b)\}$. Since $|\sqrt{x} - \sqrt{y}| \le \sqrt{|x - y|}$, for $x > 0, y > 0$, we have*

$$\text{dist}(V_1, V_2)$$

$$\le \sup_{\phi:\|\phi\|\le 1, \theta:\|\theta\|\le T, w:\|w\|\le L} \big|(\theta_1 - \theta_2 + w_1 - w_2)\phi\big| - \Big|\sqrt{\phi^\top (A_1 - A_2)\phi}\Big|$$

$$= \Big\|\theta_1 - \theta_2\Big\| + \Big\|w_1 - w_2\Big\| + \sqrt{\Big\|A_1 - A_2\Big\|_2}$$

$$\le \Big\|\theta_1 - \theta_2\Big\| + \Big\|w_1 - w_2\Big\| + \sqrt{\Big\|A_1 - A_2\Big\|_F}.$$

*Let $\mathcal{C}_\theta$ be an $\frac{\varepsilon}{2}$ − cover of $\{\theta \in \mathbb{R}^d \mid \|w\| \le T\}$ with respect to the 2-norm, $\mathcal{C}_w$ be an $\frac{\varepsilon}{2}$ − cover of $\{w \in \mathbb{R}^d \mid \|w\| \le L\}$ with respect to the 2-norm, and $\mathcal{C}_A$ be an $\frac{\varepsilon^2}{4}$ − cover of $\{A \in$*

$\mathbb{R}^{d \times d} \quad | \quad \|A\|_F \le \sqrt{d}B^2\lambda^{-1}\}$ *with respect to the Frobenius norm. By Lemma E.2, we have*

$$|\mathcal{C}_\theta| \le \left(1 + \frac{4T}{\varepsilon}\right)^d,$$

$$|\mathcal{C}_w| \le \left(1 + \frac{4L}{\varepsilon}\right)^d,$$

$$|\mathcal{C}_A| \le \left(1 + \frac{8\sqrt{d}B^2}{\lambda\varepsilon^2}\right)^{d^2}.$$

*By Equation D.4, for any $V_1 \in \mathcal{V}$, there are $\theta_2 \in \mathcal{C}_\theta$, $w_2 \in \mathcal{C}_w$ and $A_2 \in \mathcal{C}_A$ such that $V_2$ parametrized by $(\theta_2, w_2, A_2)$ satisfies $\mathrm{dist}(V_1, V_2) \le \varepsilon$. Therefore, we have $\mathcal{N}(\varepsilon) \le |\mathcal{C}_\theta| \cdot |\mathcal{C}_w| \cdot |\mathcal{C}_A|$. Then, we can conclude that*

$$\log \mathcal{N}_h(\varepsilon) \le \log|\mathcal{C}_\theta| + \log|\mathcal{C}_w| + \log|\mathcal{C}_A|$$

$$\le d\log\left(1 + \frac{4T}{\varepsilon}\right) + d\log\left(1 + \frac{4L}{\varepsilon}\right) + d^2\log\left(1 + \frac{8\sqrt{d}B^2}{\lambda\varepsilon^2}\right).$$

*This completes the proof of Lemma D.4.*

D.2   PROOF OF THEOREM 5.1

Based on Lemma B.7, we begin to bound the difference between $\mathbb{B}_h\widehat{V}_{h+1}(s,a)$ and $\widehat{\mathbb{B}}_h\widehat{V}_{h+1}(s,a)$. We first rewrite the difference as follows,

$$\mathbb{B}_h\widehat{V}_{h+1}(s,a) - \widehat{\mathbb{B}}_h\widehat{V}_{h+1}(s,a)$$

$$= r_h(s,a) - \widehat{r}_h(s,a) + \mathrm{OCE}^u_{s'\sim\mathbb{P}_h(\cdot|s,a)}\left\{\widehat{V}_{h+1}(s_{h+1})\right\} - \mathrm{OCE}^u_{s'\sim\widehat{\mathbb{P}}_h(\cdot|s,a)}\left\{\widehat{V}_{h+1}(s_{h+1})\right\}$$

$$= r_h(s,a) - \widehat{r}_h(s,a) + \max_{b\in[0,H-h]}\left\{b + \mathbb{E}_{s'\sim\mathbb{P}_h(\cdot|s,a)}\left[u\left(\widehat{V}_{h+1}(s_{h+1}) - b\right)\right]\right\}$$

$$- \max_{b\in[0,H-h]}\left\{b + \mathbb{E}_{s'\sim\widehat{\mathbb{P}}_h(\cdot|s,a)}\left[u\left(\widehat{V}_{h+1}(s_{h+1}) - b\right)\right]\right\}$$

$$= \phi(s,a)^\top\theta_h - \phi(\cdot,\cdot)^\top\widehat{\theta}_h$$

$$+ \max_{b\in[0,H-h]}\left\{b + \phi(s,a)^\top w_h(b)\right\} - \max_{b\in[0,H-h]}\left\{b + \phi(s,a)^\top\widehat{w}_h(b)\right\}.$$

Letting $b_h = \arg\max_{b\in[0,H-h]}\{b + \phi(s,a)^\top\widehat{w}_h(b)\}$, there is

$$\mathbb{B}_h\widehat{V}_{h+1}(s,a) - \widehat{\mathbb{B}}_h\widehat{V}_{h+1}(s,a)$$

$$\le \phi(s,a)^\top\theta_h - \phi(s,a)^\top\widehat{\theta}_h + \phi(s,a)^\top w_h(b_h) - \phi(s,a)^\top\widehat{w}_h(b_h)$$

$$= \phi(s,a)^\top\left(\theta_h + w_h(b_h)\right) - \phi(s,a)^\top\left[\Lambda_h^{-1}\sum_{k=1}^K\phi(s_h^k,a_h^k)\left(r_h^k + u\left(\widehat{V}_{h+1}(s_{h+1}^k) - b_h\right)\right)\right]$$

$$= \phi(s,a)^\top\left(\theta_h + w_h(b_h)\right)$$

$$- \phi(s,a)^\top\left[\Lambda_h^{-1}\sum_{k=1}^K\phi(s_h^k,a_h^k)\left(r_h^k(s_h^k,a_h^k) + \mathbb{E}_{s'\sim\mathbb{P}_h(s,a)}\left[u\left(\widehat{V}_{h+1}(s_{h+1}^k) - b_h\right)\right]\right)\right]$$

$$- \phi(s,a)^\top\left[\Lambda_h^{-1}\sum_{k=1}^K\phi(s_h^k,a_h^k)\left(\left\{r_h^k(s_h^k,a_h^k) + u\left(\widehat{V}_{h+1}(s_{h+1}^k) - b_h\right)\right\}\right.\right.$$

$$\left.\left. - \left\{r_h^k(s_h^k,a_h^k) + \mathbb{E}_{s'\sim\mathbb{P}_h(s,a)}\left[u\left(\widehat{V}_{h+1}(s_{h+1}^k) - b_h\right)\right]\right\}\right)\right]$$

$$= (i) + (ii), \tag{12}$$

where we let

$$
(i) := \phi(s,a)^\top \big(\theta_h + w_h(b_h)\big)
$$

$$
- \phi(s,a)^\top \left[ \Lambda_h^{-1} \sum_{k=1}^K \phi(s_h^k, a_h^k) \left( r_h^k(s_h^k, a_h^k) + \mathbb{E}_{s' \sim \mathbb{P}_h(s,a)} \left[ u\big(\widehat{V}_{h+1}(s_{h+1}^k) - b_h\big)\right] \right) \right],
$$

$$
(ii) := - \phi(s,a)^\top \left[ \Lambda_h^{-1} \sum_{k=1}^K \phi(s_h^k, a_h^k) \left( \left\{ r_h^k(s_h^k, a_h^k) + u\big(\widehat{V}_{h+1}(s_{h+1}^k) - b_h\big) \right\} \right. \right.
$$

$$
\left. \left. - \left\{ r_h^k(s_h^k, a_h^k) + \mathbb{E}_{s' \sim \mathbb{P}_h(s,a)} \left[ u\big(\widehat{V}_{h+1}(s_{h+1}^k) - b_h\big) \right] \right\} \right) \right].
$$

For term $(i)$, we have

$$
|(i)| = \left| \phi(s,a)^\top \big(\theta_h + w_h(b_h)\big) - \phi(s,a)^\top \Lambda_h^{-1} \sum_{k=1}^K \Big[ \phi(s_h^k, a_h^k) \Big( r_h(s_h^k, a_h^k) \right.
$$

$$
\left. + \mathbb{E}_{s' \sim \mathbb{P}_h(s,a)} \big[ u\big(\widehat{V}_{h+1}(s_{h+1}^k) - b_h\big)\big]\Big) \Big] \right|
$$

$$
= \left| \phi(s,a)^\top \big(\theta_h + w_h(b_h)\big) - \phi(s,a)^\top \Lambda_h^{-1} \sum_{k=1}^K \phi(s_h^k, a_h^k) \phi(s_h^k, a_h^k)^\top \big(\theta_h + w_h(b_h)\big) \right|
$$

$$
= \left| \phi(s,a)^\top \big(\theta_h + w_h(b_h)\big) - \phi(s,a)^\top \Lambda_h^{-1} \big(\Lambda_h - \lambda I\big) \big(\theta_h + w_h(b_h)\big) \right|
$$

$$
= \lambda \left| \phi(s,a)^\top \Lambda_h^{-1} (\theta_h + w_h(b_h)) \right|.
$$

Due to the Cauchy-Schwarz inequality, we can further bound the term as follows,

$$
|(i)| \leq \lambda \cdot \left\| \Lambda_h^{-1} \right\|_2^{\frac{1}{2}} \cdot \left\| \theta_h + w_h(b_h) \right\| \cdot \left\| \phi(s,a) \right\|_{\Lambda_h^{-1}} \tag{13}
$$

$$
\leq \big(1 + u(H-h)\big) \sqrt{d\lambda} \sqrt{\phi(s,a)^\top \Lambda_h^{-1} \phi(s,a)},
$$

where the last inequality holds due to Lemma D.1. Next, we need to bound the term $(ii)$. We first define

$$
\epsilon_h^k(f) = \big\{ r_h^k + u(f(s_{h+1}^k) - b_h) \big\} - \big\{ r_h^k(s_h^k, a_h^k) + \mathbb{E}_{s' \sim \mathbb{P}_h(\cdot|s,a)} \big[ u(f(s_{h+1}^k) - b_h)\big] \big\},
$$

where $f : \mathcal{S} \to [0, f_{\max}]$ is an arbitrary function. Then we have

$$
|(ii)| = \left| \phi(s,a)^\top \left[ \Lambda_h^{-1} \sum_{k=1}^K \phi(s_h^k, a_h^k) \left( \left\{ r_h^k(s_h^k, a_h^k) + u\big(\widehat{V}_{h+1}(s_{h+1}^k) - b_h\big) \right\} \right. \right. \right.
$$

$$
\left. \left. \left. - \left\{ r_h^k(s_h^k, a_h^k) + \mathbb{E}_{s' \sim \mathbb{P}_h(s,a)} \big[ u\big(\widehat{V}_{h+1}(s_{h+1}^k) - b_h\big)\big] \right\} \right) \right] \right|
$$

$$
= \left| \phi(s,a) \Lambda_h^{-1} \Big( \sum_{k=1}^K \phi(s_h^k, a_h^k) \epsilon_h^k\big(\widehat{V}_{h+1}\big) \Big) \right|
$$

$$
\leq \left\| \sum_{k=1}^K \phi(s_h^k, a_h^k) \epsilon_h^k\big(\widehat{V}_{h+1}\big) \right\|_{\Lambda_h^{-1}} \cdot \sqrt{\phi(s,a)^\top \Lambda_h^{-1} \phi(s,a)}.
$$

Therefore, we need to bound $\big\| \sum_{k=1}^K \phi(s_h^k, a_h^k) \epsilon_h^k(\widehat{V}_{h+1}) \big\|_{\Lambda_h^{-1}}$. Based on Definition D.1, we have $\widehat{V}_{h+1} \in \mathcal{V}$. Let $\mathcal{N}_{h+1}(\varepsilon)$ be the $\varepsilon$-cover of $V(\cdot)$, we can find a function $V'_{h+1} \in \mathcal{N}_{h+1}(\varepsilon)$ such that

$$
\sup_{s \in \mathcal{S}} \left| \widehat{V}_{h+1}(s) - V'_{h+1}(s) \right| \leq \varepsilon.
$$

Therefore, we have

$$\left| u(\widehat{V}_{h+1}(s) - b_h) - u(V'_{h+1}(s) - b_h) \right|$$
$$\leq \left| \widehat{V}_{h+1}(s) - V'_{h+1}(s) \right|$$
$$\leq \varepsilon.$$

The first inequality holds based on the property of the utility function $u$ that $u$ is concave, nondecreasing, and $1 \in \partial u(0)$. Then, we can get

$$\left| \mathbb{E}_{s' \sim \mathbb{P}_h(\cdot|s,a)} \left[ u(\widehat{V}_{h+1}(s) - b_h) \right] - \mathbb{E}_{s' \sim \mathbb{P}_h(\cdot|s,a)} \left[ u(V'_{h+1}(s) - b_h) \right] \right|$$
$$\leq \left| \mathbb{E}_{s' \sim \mathbb{P}_h(\cdot|s,a)} \left[ u(\widehat{V}_{h+1}(s) - b_h) - u(V'_{h+1}(s) - b_h) \right] \right|$$
$$\leq \varepsilon.$$

Therefore, by the triangle inequality, we have

$$2\varepsilon \geq \left| \left( u(\widehat{V}_{h+1}(s) - b_h) - \mathbb{E}_{s' \sim \mathbb{P}_h(\cdot|s,a)} \left[ u(\widehat{V}_{h+1}(s) - b_h) \right] \right) \right.$$
$$\left. - \left( u(V'_{h+1}(s) - b_h) - \mathbb{E}_{s' \sim \mathbb{P}_h(\cdot|s,a)} \left[ u(V'_{h+1}(s) - b_h) \right] \right) \right|.$$

This can further guarantee that

$$\left| \epsilon_h^k(\widehat{V}) - \epsilon_h^k(V') \right| \leq 2\varepsilon.$$

Then based on the fact that $\|a + b\|_{\Lambda_h^{-1}}^2 \leq 2\|a\|_{\Lambda_h^{-1}}^2 + 2\|b\|_{\Lambda_h^{-1}}^2$, we can then get

$$\left\| \sum_{k=1}^{K} \phi(s_h^k, a_h^k) \epsilon_h^k(\widehat{V}_{h+1}) \right\|_{\Lambda_h^{-1}}^2$$
$$\leq 2 \left\| \sum_{k=1}^{K} \phi(s_h^k, a_h^k) \epsilon_h^k(V'_{h+1}) \right\|_{\Lambda_h^{-1}}^2 \tag{14}$$
$$+ 2 \left\| \sum_{k=1}^{K} \phi(s_h^k, a_h^k) \left[ \epsilon_h^k(\widehat{V}_{h+1}) - \epsilon_h^k(V'_{h+1}) \right] \right\|_{\Lambda_h^{-1}}^2$$
$$\leq 2 \sup_{V \in \mathcal{N}_{h+1}(\varepsilon)} \left\| \sum_{k=1}^{K} \phi(s_h^k, a_h^k) \epsilon_h^k(V) \right\|_{\Lambda_h^{-1}}^2 + \frac{8\varepsilon^2 K^2}{\lambda}.$$

Here we can get an upper bound without the influence of the dataset $\mathcal{D}$. Combining Lemma D.3 and the union bound, we have

$$\mathbb{P}_{\mathcal{D}} \left[ \sup_{V \in \mathcal{N}_{h+1}(\varepsilon)} \left\| \sum_{K=1}^{K} \phi(s_h^k, a_h^k) \epsilon_h^k(V) \right\|_{\Lambda_h^{-1}}^2 \right.$$
$$\left. > [1 + u(H - h)]^2 \left( 2\log \frac{1}{\Delta} + d\log \left( 1 + \frac{K}{\lambda} \right) \right) \right]$$
$$\leq \Delta |\mathcal{N}_{h+1}(\varepsilon)|.$$

Letting $\Delta = \frac{\delta}{H|\mathcal{N}_{h+1}(\varepsilon)|}$, for $\delta \in (0, 1)$. Then at any step $h$, with probability $1 - \frac{\delta}{H}$, we have

$$\sup_{V \in \mathcal{N}_{h+1}(\varepsilon)} \left\| \sum_{K=1}^{K} \phi(s_h^k, a_h^k) \epsilon_h^k(V) \right\|_{\Lambda_h^{-1}}^2 \leq [1 + u(H - h)]^2 \left( 2\log \frac{H|\mathcal{N}_{h+1}(\varepsilon)|}{\delta} + d\log \left( 1 + \frac{K}{\lambda} \right) \right).$$

Then, with Equation 14, with probability at $1 - \delta$, for all $\forall h \in [H]$, there is

$$\left\| \sum_{K=1}^{K} \phi(s_h^k, a_h^k) \epsilon_h^k(\widehat{V}_{h+1}) \right\|_{\Lambda_h^{-1}}^2$$
$$\leq 2[1 + u(H - h)]^2 \left( 2\log \frac{H|\mathcal{N}_{h+1}(\varepsilon)|}{\delta} + d\log(1 + \frac{K}{\lambda}) \right) + \frac{8\varepsilon^2 K^2}{\lambda}, \forall h \in [H].$$

Then we need to bound to bound $\log(\mathcal{N}_{h+1}(\varepsilon))$. With Lemma D.2, we set $T = \sqrt{\frac{dK}{\lambda}}$ and $L = u(H-h)\sqrt{\frac{dK}{\lambda}}$. Let $\varepsilon = \frac{dH}{K}$, $B = 2\beta$, $\beta = cd[1+u(H-h)]\sqrt{\zeta}$, and $\zeta = \log(2dHK\delta^{-1})$, where $c > 0$ is a constant. Notice that $u(H-h) \leq u(H) \leq H$, due to the concavity of the utility function $u$ along with $1 \in \partial u(0)$. Then by using Lemma D.4, we have

$$
\begin{aligned}
\log \mathcal{N}_h(\varepsilon) \leq & d\log\left(1+\frac{4T}{\varepsilon}\right) + d\log\left(1+\frac{4L}{\varepsilon}\right) + d^2\log\left(1+\frac{8\sqrt{d}B^2}{\varepsilon^2}\right) \\
\leq & d\log\left(1+4d^{-\frac{1}{2}}K^{\frac{3}{2}}H^{-1}\right) + d\log\left(1+4u(H-h)d^{-\frac{1}{2}}K^{\frac{3}{2}}H^{-1}\right) \\
& + d^2\log\left(1+8B^2d^{-\frac{3}{2}}K^2H^{-2}\right) \\
\leq & 2d\log\left(1+4d^{-\frac{1}{2}}K^{\frac{3}{2}}\right) + d^2\log\left(1+32c^2d^{\frac{1}{2}}K^2\zeta\right) \\
\leq & 3d^2\log\left(1+32c^2d^{\frac{1}{2}}K^2\zeta\right) \\
\leq & 3d^2\log\left(64c^2d^{\frac{1}{2}}K^2\zeta\right).
\end{aligned} \tag{15}
$$

Then with the fact that $\log\zeta \leq \zeta$, $\log(1+K) \leq \log(2K) \leq \zeta$, and Equation 15, we can get

$$
\left\|\sum_{K=1}^K \phi(s_h^k, a_h^k)\epsilon_h^k(\widehat{V}_{h+1})\right\|_{\Lambda_h^{-1}}^2
$$

$$
\leq 2[1+u(H-h)]^2\left(2\log(H\delta^{-1}) + 4d^2\log(64c^2d^{\frac{1}{2}}K^2\zeta) + d\log(1+K) + 4d^2\right)
$$

$$
\leq 2[1+u(H-h)]^2\left(2\log(H\delta^{-1}) + 6d^2\log(64c^2) + 6d^2\zeta + 3d^2\log(dK^4) + d\zeta + 4d^2\right)
$$

$$
\leq 2[1+u(H-h)]^2\left(3d^2\log(dHK^4\delta^{-1}) + 6d^2\log(64c^2) + 11d^2\zeta\right)
$$

$$
= 2[1+u(H-h)]^2\left(3d^2\log(dHK\delta^{-1}) + 9d^2\log K + 6d^2\log(64c^2) + 11d^2\zeta\right)
$$

$$
\leq d^2[1+u(H-h)]^2\zeta\left(12\log(64c^2) + 46\right).
$$

By setting $12\log(64c^2) + 46 \leq \frac{c^2}{4}$, the following inequality holds,

$$
\left\|\sum_{K=1}^K \phi(s_h^k, a_h^k)\epsilon_h^k(\widehat{V}_{h+1})\right\|_{\Lambda_h^{-1}} \leq \frac{1}{2}cd[1+u(H-h)]\sqrt{\zeta} = \frac{\beta}{2}. \tag{16}
$$

Therefore, based on Equation 12, Equation 13, Equation 16, we have

$$
\left|\mathbb{E}_{s',r}\left[\widehat{V}_{h+1}(s, b-r)\right] - \widehat{\mathbb{E}}_{s',r}\left[\widehat{V}_{h+1}(s', b-r)\right]\right|
$$

$$
\leq \left([1+u(H-h)]\sqrt{d} + \frac{1}{2}cd[1+u(H-h)]\sqrt{\zeta}\right)\sqrt{\phi(s,a)^\top\Lambda_h^{-1}\phi(s,a)}
$$

$$
\leq \beta\sqrt{\phi(s,a)^\top\Lambda_h^{-1}\phi(s,a)}.
$$

Then we finish proving Theorem 5.1.

### D.3 LEMMAS FOR THEOREM 5.2

Similar to the dynamic-OCE formulation with tabular setting, we extend the setting to stochastic reward functions where $r_h \sim \mathcal{R}(\cdot|s,a)$ in the proof. When $\mathcal{R}(r_h|s,a) = 1$, it reduce to the deterministic reward case used in the paper. Therefore, in this section we actually provide a more general proof, which extends Theorem 5.2. Under the stochastic reward setting with linear MDP, we slight change the setting to

$$
\mathbb{P}_h(\cdot|s,a) = \langle\mu_h(\cdot), \varphi(s,a)\rangle
$$

$$
\mathcal{R}_h(\cdot|s,a) = \langle\nu_h(\cdot), \psi(s,a)\rangle.
$$

Therefore we set a matrix $\Phi(s,a) \in \mathbb{R}^{d\times d}$, a vector $\xi_h(s',r) \in \mathbb{R}^{d^2\times 1}$ and a a vector $\phi(s',r) \in \mathbb{R}^{d^2\times 1}$ satisfying

$$
\Phi(s,a) = \psi(s,a)\varphi(s,a)^\top
$$

$$
\xi_h(s',r)_{i\times d+j} = \left(\nu_h(r)\mu_h(s')^\top\right)_{i,j}
$$

$$
\phi(s,a)_{i\times d+j} = \Phi(s,a)_{i,j}.
$$

Then we have

$$
\begin{aligned}
\mathbb{E}_{s'\sim\mathbb{P}_h(\cdot|s,a),r\sim\mathcal{R}_h(\cdot|s,a)}\big[V(s',b-r)\big] &= \int_r \mathcal{R}_h(\cdot|s,a)\int_{s'}\mathbb{P}_h(s,a)V(s',b-r)ds'dr \\
&= \nu_h(r)^\top\psi(s,a)\varphi(s,a)^\top\mu_h(s')V(s',b-r) \\
&= \nu_h(r)^\top\Phi(s,a)\mu_h(s')V(s',b-r) \\
&= \phi(s,a)^\top\xi_h(s',r)V(s',b-r) \\
&= \phi(s,a)^\top\widehat{w}_h(b),
\end{aligned}
$$

where the last equality holds when $w_h(b) = \xi_h(s',r)V(s',b-r)$. Here we success-fully extend the setting from $\mathbb{E}_{s'\sim\mathbb{P}_h(\cdot|s,a),r=r_h(s,a)}\big[V(s',b-r)\big]$ with deterministic reward to $\mathbb{E}_{s'\sim\mathbb{P}_h(\cdot|s,a),r\sim\mathcal{R}_h(\cdot|s,a)}\big[V(s',b-r)\big]$ with stochastic reward. Therefore, in the stochastic reward setting, we can still use $\phi(s,a)^\top\widehat{w}_h(b)$ to estimate the transition.

**Lemma D.5** *Based on the definition of $\widehat{w}_h(b)$ and $\widehat{V}_{h+1}(s',b-r)$, we have*

$$
\begin{cases}
\big\|w_h(b)\big\| \le u(H-h-b)\sqrt{d} \\
\big\|\widehat{w}_h(b)\big\| \le u(H-h-b)\sqrt{\frac{dK}{\lambda}}.
\end{cases}
$$

**Proof** *The $w_h(b)$ is defined as follows,*

$$
w_h(b) = \int_r\int_{s'}\xi_h(s',r)\widehat{V}_{h+1}(s',b-r)ds'dr.
$$

*Then we can get*

$$
\begin{aligned}
\big\|w_h(b)\big\| &= \Big\|\int_r\int_{s'}\xi_h(s',r)\widehat{V}_{h+1}(s',b-r)ds'dr\Big\| \\
&\le u(H-h-b)\sqrt{d}.
\end{aligned}
$$

*For $\widehat{w}_h(b)$, we have*

$$
\begin{aligned}
\big\|\widehat{w}_h(b)\big\| &= \Big\|\Lambda_h^{-1}\Big\{\sum_{k=1}^K \phi(s_h^k,a_h^k)\widehat{V}_{h+1}(s_{h+1}^k,b-r_h^k)\Big\}\Big\| \\
&\le \sum_{k=1}^K\Big\|\Lambda_h^{-1}\phi(s_h^k,a_h^k)\widehat{V}_{h+1}(s_{h+1}^k,b-r_h^k)\Big\| \\
&\le u(H-h-b)\sum_{k=1}^K\Big\|\Lambda_h^{-1}\phi(s_h^k,a_h^k)\Big\| \\
&= u(H-h-b)\sum_{k=1}^K\sqrt{\phi(s_h^k,a_h^k)^\top\Lambda_h^{-\frac{1}{2}}\Lambda_h^{-1}\Lambda_h^{-\frac{1}{2}}\phi(s_h^k,a_h^k)}.
\end{aligned}
$$

Based on the Cauchy-Schwarz inequality, we have

$$
\left\| \widehat{w}_h(b) \right\| = \left\| \Lambda_h^{-1} \Big\{ \sum_{k=1}^{K} \phi(s_h^k, a_h^k) \widehat{V}_{h+1}(s_{h+1}^k, b - r_h^k) \Big\} \right\|
$$

$$
\leq u(H - h - b) \sqrt{\frac{K}{\lambda}} \sqrt{\sum_{k=1}^{K} \phi(s_h^k, a_h^k)^\top \Lambda_h^{-1} \phi(s_h^k, a_h^k)}
$$

$$
= u(H - h - b) \sqrt{\frac{K}{\lambda}} \sqrt{\mathrm{Tr}\big(\Lambda_h^{-1} \sum_{k=1}^{K} \phi(s_h^k, a_h^k)^\top \phi(s_h^k, a_h^k)\big)}
$$

$$
= u(H - h - b) \sqrt{\frac{K}{\lambda}} \sqrt{\mathrm{Tr}\big(\Lambda_h^{-1}(\Lambda_h - \lambda I)\big)}
$$

$$
\leq u(H - h - b) \sqrt{\frac{K}{\lambda}} \sqrt{\mathrm{Tr}\big(\Lambda_h^{-1}\Lambda_h\big)}
$$

$$
= u(H - h - b) \sqrt{\frac{dK}{\lambda}}.
$$

*Therefore, we finish the proof.*

**Lemma D.6** *For a fixed function $V_h : \mathcal{S} \to [0, u(H - h - b_h)]$ at step $h \in [H]$, under the assumption that $\mathcal{D}$ is obtained by an underlying policy $\mu$, for any $\Delta \in (0, 1)$, we have*

$$
\mathbb{P}_\mathcal{D}\left( \Big\| \sum_{K=1}^{K} \phi(s_h^k, a_h^k) \epsilon_h^k(V_h) \Big\|_{\Lambda_h^{-1}}^2 > [u(H - h - b_h)]^2 \Big( 2 \log \frac{1}{\Delta} + d \log \big(1 + \frac{K}{\lambda}\big) \Big) \right) \leq \Delta.
$$

**Proof** *For any fixed $h \in [H]$ and $k \in \{0, 1, \dots, K\}$, we have the $\sigma$-algebra*

$$
\mathcal{F}_h^k = \sigma\left( \{(s_h^j, a_h^j)\}_{j=1}^{\min\{k+1, K\}} \right).
$$

*Then for any $k \in [K]$, we have $\phi(s_h^k, a_h^k) \in \mathcal{F}_h^k$, since $(s_h^k, a_h^k)$ is measurable with respect to $\mathcal{F}_h^{k-1}$. Then for a fixed function $V_h : \mathcal{S} \to [0, u(H - h - b_h)]$ at step $h$, and $k \in [K]$, we have*

$$
\begin{aligned}
&\epsilon_h^k(V_h) \\
=& V_h(s_{h+1}^k, b_h - r_h^k) - \mathbb{E}_{s_{h+1}^k \sim \mathbb{P}_h(\cdot|s_h^k, a_h^k), r \sim \mathcal{R}_h(\cdot|s_h^k, a_h^k)} \big[ V_h(s_{h+1}^k, b_h - r_h^k) \big] \\
\in& \mathcal{F}_h^k.
\end{aligned}
$$

*Therefore, $\{\epsilon_h^k(V_h)\}_{k=1}^K$ is a stochastic process with respect to the filtration $\{\mathcal{F}_h^k\}_{k=0}^K$. Then with Assumption 2.1, we have*

$$
\begin{aligned}
&\mathbb{E}_\mathcal{D}\Big[ \epsilon_h^k(V_h) | \mathcal{F}_h^{k-1} \Big] \\
=& \mathbb{E}_\mathcal{D}\Big[ V_h(s_{h+1}^k, b_h - r_h^k) | \{(s_h^j, a_h^j)\}_{j=1}^k \Big] - \mathbb{E}_{s', r}\big[ V_h(s_{h+1}^k, b_h - r_h^k) \big] \\
=& 0.
\end{aligned}
$$

*Based on the definition of $\epsilon_h^k(V_h)$, we have $|\epsilon_h^k(V_h)| \leq u(H - h - b_h)$. Thus, for the fixed $h$ and all $k \in [K]$, $\epsilon_h^k(V_h)$ is a zero-mean and $u(H - h - b_h)$-sub-Gaussian random variable conditioning on $\mathcal{F}_h^{k-1}$. Based on Lemma E.1 with $M_0 = \lambda I$ and $M_k = \lambda I + \frac{1}{K} \sum_{j=1}^{K} \phi(s_h^j, a_h^j) \phi(s_h^j, a_h^j)^\top$, for all $\Delta \in (0, 1)$, we have*

$$
\mathbb{P}_\mathcal{D}\left( \Big\| \sum_{k=1}^{K} \phi(s_h^k, a_h^k) \epsilon_h^k(V_h) \Big\|_{\Lambda_h^{-1}}^2 > 2[u(H - h - b_h)]^2 \log \Big( \frac{\det(\Lambda_h)^{\frac{1}{2}}}{\Delta \cdot \det(\lambda I)^{\frac{1}{2}}} \Big) \right) \leq \Delta,
$$

*where the equation holds based on the fact that $M_K = \Lambda_h$. By applying the definition of $\Lambda_h$, we have $\|\Lambda_h\|_2 \leq \lambda + K$ which implies $\det(\Lambda_h) \leq (\lambda + K)^d$. Therefore, we can get*

$$\mathbb{P}_{\mathcal{D}}\bigg(\bigg\|\sum_{K=1}^{K}\phi(s_h^k, a_h^k)\epsilon_h^k(V_h)\bigg\|_{\Lambda_h^{-1}}^2 > [u(H-h-b_h)]^2\Big(2\log\frac{1}{\Delta} + d\log\big(1+\frac{K}{\lambda}\big)\Big)\bigg)$$

$$\leq\mathbb{P}_{\mathcal{D}}\bigg(\bigg\|\sum_{k=1}^{K}\phi(s_h^k, a_h^k)\epsilon_h^k(V_h)\bigg\|_{\Lambda_h^{-1}}^2 > 2[u(H-h-b_h)]^2\log\Big(\frac{\det(\Lambda_h)^{\frac{1}{2}}}{\Delta\cdot\det(\lambda I)^{\frac{1}{2}}}\Big)\bigg)$$

$$\leq\Delta$$

*Here we finish the proof.*

**Definition D.2** *Define the function class $\mathcal{V}$ mapping from $\mathcal{S} \times [0, H]$ to $\mathbb{R}$ has the following parametric form,*

$$V(\cdot, b) = \max_a\Big\{\max\Big\{\min\big\{\phi(\cdot, a)^\top w(b) - \beta\sqrt{\phi(\cdot, a)^\top\Lambda^{-1}\phi(\cdot, a)}, u(H-h-b)\big\}, 0\Big\}\Big\},$$

*where $b \in [0, 1]$ is a parameter, $\|w(b)\| \leq L$, $\beta \in [0, B]$ and $\Lambda \succeq \lambda I$.*

**Lemma D.7** *Based on Definition D.2, for all $h \in [H]$ and $\varepsilon > 0$, we have*

$$\log\mathcal{N}_h(\varepsilon) \leq d\log\Big(1 + \frac{4L}{\varepsilon}\Big) + d^2\log\Big(1 + \frac{8\sqrt{d}B^2}{\lambda\varepsilon^2}\Big).$$

**Proof** *For the function class $\mathcal{V}$, we set $A = \beta^2\Lambda^{-1}$. Therefore, by the definition of function class $\mathcal{V}$, we have $\|w(b)\| \leq L$ and $\|A\| \leq \frac{B^2}{\lambda}$. Letting any two functions $V_1, V_2 \in \mathcal{V}$, we have*

$$\text{dist}(V_1, V_2)$$

$$\leq\sup_{s,a,b}\Big|\Big[w_1(b)\phi(s, a) - \sqrt{\phi(s, a)^\top A_1\phi(s, a)}\Big] - \Big[w_2(b)\phi(s, a) - \sqrt{\phi(s, a)^\top A_2\phi(s, a)}\Big]\Big|$$

$$\leq\sup_{\phi:\|\phi\|\leq 1, w:\|w\|\leq L}\Big|\Big[w_1\phi - \sqrt{\phi^\top A_1\phi}\Big] - \Big[w_2\phi - \sqrt{\phi^\top A_2\phi}\Big]\Big|$$

$$\leq\sup_{\phi:\|\phi\|\leq 1, w:\|w\|\leq L}\big|(w_1 - w_2)\phi\big| - \Big|\sqrt{\phi^\top(A_1 - A_2)\phi}\Big|$$

$$=\Big\|w_1 - w_2\Big\| + \sqrt{\Big\|A_1 - A_2\Big\|_2} \leq \Big\|w_1 - w_2\Big\| + \sqrt{\Big\|A_1 - A_2\Big\|_F},$$

*where the third inequality holds due to $|\sqrt{x} - \sqrt{y}| \leq \sqrt{|x - y|}$, for $x > 0, y > 0$. Let $\mathcal{C}_w$ be an $\frac{\varepsilon}{2}$ − cover of $\{w \in \mathbb{R}^d| \quad \|w\| \leq L\}$ with respect to the 2-norm, and $\mathcal{C}_A$ be an $\frac{\varepsilon^2}{4}$ − cover of $\{A \in \mathbb{R}^{d\times d}| \quad \|A\|_F \leq \sqrt{d}B^2\lambda^{-1}\}$ with respect to the Frobenius norm. By Lemma E.2, we have*

$$|\mathcal{C}_w| \leq \Big(1 + \frac{4L}{\varepsilon}\Big)^d,$$

$$|\mathcal{C}_A| \leq \Big(1 + \frac{8\sqrt{d}B^2}{\lambda\varepsilon^2}\Big)^{d^2}.$$

*By Equation D.7, for any $V_1 \in \mathcal{V}$, there are $w_2 \in \mathcal{C}_w$ and $A_2 \in \mathcal{C}_A$ such that $V_2$ parametrized by $(w_2, A_2)$ satisfies $\text{dist}(V_1, V_2) \leq \varepsilon$. Therefore, we have $\mathcal{N}(\varepsilon) \leq |\mathcal{C}_w| \cdot |\mathcal{C}_A|$. Then, we can obtain*

$$\log\mathcal{N}_h(\varepsilon) \leq \log|\mathcal{C}_w| + \log|\mathcal{C}_A| \leq d\log\Big(1 + \frac{4L}{\varepsilon}\Big) + d^2\log\Big(1 + \frac{8\sqrt{d}B^2}{\lambda\varepsilon^2}\Big).$$

*This completes the proof of Lemma D.7.*

D.4  PROOF OF THEOREM 5.2

In this section, we extend the proof to stochastic reward, where $r_h \sim \mathcal{R}(\cdot|s, a)$, to get a more general result. When $\mathcal{R}(\cdot|s, a) = 1$, we get exactly the proof of Theorem 5.2 with deterministic reward.

With Lemma B.13, we need to bound $\mathbb{E}_{s',r}[\widehat{V}_{h+1}(s, b-r)] - \widehat{\mathbb{E}}_{s',r}[\widehat{V}_{h+1}(s', b-r)]$, considering the definition of $\iota_h$. We have

$$\mathbb{E}_{s',r}\left[\widehat{V}_{h+1}(s, b-r)\right] - \widehat{\mathbb{E}}_{s',r}\left[\widehat{V}_{h+1}(s', b-r)\right]$$

$$=\phi(s,a)^\top w_h(b) - \phi(s,a)^\top \Lambda_h^{-1}\Big(\sum_{k=1}^K \phi(s_h^k, a_h^k)\widehat{V}_{h+1}(s_{h+1}^k, b - r_h^k)\Big)$$

$$=\phi(s,a)^\top w_h(b) - \phi(s,a)^\top \Lambda_h^{-1}\Big(\sum_{k=1}^K \phi(s_h^k, a_h^k)\mathbb{E}_{s',r}[\widehat{V}_{h+1}(s_{h+1}^k, b - r_h^k)]\Big)$$

$$- \phi(s,a)^\top \Lambda_h^{-1}\Big[\sum_{k=1}^K \phi(s_h^k, a_h^k)\Big(\widehat{V}_{h+1}(s_{h+1}^k, b - r_h^k)$$

$$- \mathbb{E}_{s',r}\big[\widehat{V}_{h+1}(s_{h+1}^k, b - r_h^k)\big]\Big)\Big].$$

Then, we can get the following inequality,

$$\mathbb{E}_{s',r}\left[\widehat{V}_{h+1}(s, b-r)\right] - \widehat{\mathbb{E}}_{s',r}\left[\widehat{V}_{h+1}(s', b-r)\right]$$

$$\leq\Big|\phi(s,a)^\top w_h(b) - \phi(s,a)^\top \Lambda_h^{-1}\Big[\sum_{k=1}^K \phi(s_h^k, a_h^k)\mathbb{E}_{s',r}\big[\widehat{V}_{h+1}(s_{h+1}^k, b - r_h^k)\big]\Big]\Big|$$

$$+ \Big|\phi(s,a)^\top \Lambda_h^{-1}\Big[\sum_{k=1}^K \phi(s_h^k, a_h^k)\Big(\widehat{V}_{h+1}(s_{h+1}^k, b - r_h^k)$$

$$- \mathbb{E}_{s',r}\big[\widehat{V}_{h+1}(s_{h+1}^k, b - r_h^k)\big]\Big)\Big]\Big|. \tag{17}$$

For the first term, $|\phi(s,a)^\top w_h(b) - \phi(s,a)^\top \Lambda_h^{-1}[\sum_{k=1}^K \phi(s_h^k, a_h^k)\mathbb{E}_{s',r}[\widehat{V}_{h+1}(s_{h+1}^k, b - r_h^k)]]|$, we have

$$\Big|\phi(s,a)^\top w_h(b) - \phi(s,a)^\top \Lambda_h^{-1}\Big[\sum_{k=1}^K \phi(s_h^k, a_h^k)\mathbb{E}_{s',r}\big[\widehat{V}_{h+1}(s_{h+1}^k, b - r_h^k)\big]\Big]\Big|$$

$$=\Big|\phi(s,a)^\top w_h(b) - \phi(s,a)^\top \Lambda_h^{-1}\Big[\sum_{k=1}^K \phi(s_h^k, a_h^k)\phi(s_h^k, a_h^k)^\top w_h(b)\Big]\Big|$$

$$=\Big|\phi(s,a)^\top w_h(b) - \phi(s,a)^\top \Lambda_h^{-1}\big(\Lambda_h - \lambda I\big)w_h(b)\Big|$$

$$=\lambda\Big|\phi(s,a)^\top \Lambda_h^{-1} w_h(b)\Big|.$$

Due to the Cauchy-Schwarz inequality, there is

$$\Big|\phi(s,a)^\top w_h(b) - \phi(s,a)^\top \Lambda_h^{-1}\Big[\sum_{k=1}^K \phi(s_h^k, a_h^k)\mathbb{E}_{s',r}\big[\widehat{V}_{h+1}(s_{h+1}^k, b - r_h^k)\big]\Big]\Big|$$

$$\leq\lambda\Big\|\phi(s,a)^\top\Big\|_{\Lambda_h^{-1}}\big\|w_h(b)\big\|_{\Lambda_h^{-1}}$$

$$=\lambda\sqrt{w_h(b)^\top \Lambda_h^{-1} w_h(b)}\sqrt{\phi(s,a)^\top \Lambda_h^{-1}\phi(s,a)}$$

$$\leq\lambda\Big\|\Lambda_h^{-1}\Big\|_2^{\frac{1}{2}}\big\|w_h\big\|\sqrt{\phi(s,a)^\top \Lambda_h^{-1}\phi(s,a)}$$

$$\leq\lambda \cdot \lambda^{-\frac{1}{2}}u(H - h - b)\sqrt{d}\sqrt{\phi(s,a)^\top \Lambda_h^{-1}\phi(s,a)}$$

$$=u(H - h - b)\sqrt{d\lambda}\sqrt{\phi(s,a)^\top \Lambda_h^{-1}\phi(s,a)},$$

where the last inequality is based on Lemma D.5. Then for any function $V : \mathcal{S} \times [0, H] \to [0, V_{\max}]$, we set

$$\epsilon_h^k(V) = V(s_{h+1}^k, b - r_h^k) - \mathbb{E}_{s',r}\big[V(s_{h+1}^k, b - r_h^k)\big].$$

Therefore, for the second term, $\left|\phi(s,a)^\top \Lambda_h^{-1}\left(\sum_{k=1}^K \phi(s_h^k, a_h^k)(\widehat{V}_{h+1}(s_{h+1}^k, b - r_h^k) - \mathbb{E}_{s',r}[\widehat{V}_{h+1}(s_{h+1}^k, b - r_h^k)])\right)\right|$, by the Cauchy-Schwarz inequality, we have

$$
\begin{aligned}
&\left|\phi(s,a)^\top \Lambda_h^{-1}\Big[\sum_{k=1}^K \phi(s_h^k, a_h^k)\Big(\widehat{V}_{h+1}(s_{h+1}^k, b - r_h^k)\right.\\
&\left.\quad - \mathbb{E}_{s',r}\big[\widehat{V}_{h+1}(s_{h+1}^k, b - r_h^k)\big]\Big)\Big]\right|\\
&=\left|\phi(s,a)^\top \Lambda_h^{-1}\Big[\sum_{k=1}^K \phi(s_h^k, a_h^k)\epsilon_h^k(\widehat{V}_{h+1})\Big]\right|\\
&\leq \Big\|\sum_{k=1}^K \phi(s_h^k, a_h^k)\epsilon_h^k(\widehat{V}_{h+1})\Big\|_{\Lambda_h^{-1}} \cdot \sqrt{\phi(s,a)^\top \Lambda_h^{-1}\phi(s,a)}.
\end{aligned}
\tag{18}
$$

The rest of the problem is to upper bound $\big\|\sum_{k=1}^K \phi(s_h^k, a_h^k)\epsilon_h^k(\widehat{V}_{h+1})\big\|_{\Lambda_h^{-1}}$. Obviously, by Definition D.2, it holds that $\widehat{V}_{h+1} \in \mathcal{V}$. Set $\mathcal{N}_{h+1}(\varepsilon)$ is an $\varepsilon-$ cover of $V(\cdot, b)$, there is a function $V'_{h+1} \in \mathcal{N}_{h+1}(\varepsilon)$ such that

$$
\sup_{s \in \mathcal{S}} \big|\widehat{V}_{h+1}(s, b) - V'_{h+1}(s, b)\big| \leq \varepsilon.
$$

Hence, we can obtain

$$
\begin{aligned}
&\left|\mathbb{E}_{s',r}\big[\widehat{V}_{h+1}(s, b)\big|s_h, a_h\big] - \mathbb{E}_{s',r}\big[V'_{h+1}(s, b)\big|s_h, a_h\big]\right|\\
&=\left|\mathbb{E}_{s',r}\big[\widehat{V}_{h+1}(s, b) - V'_{h+1}(s, b)\big|s_h, a_h\big]\right|\\
&\leq \varepsilon.
\end{aligned}
$$

Then, by the triangle inequality, we have

$$
\left|\Big(\widehat{V}_{h+1}(s', b) - \mathbb{E}_{s',r}\big[\widehat{V}_{h+1}(s', b)\big]\Big) - \Big(V'_{h+1}(s', b) - \mathbb{E}_{s',r}\big[V'_{h+1}(s', b)\big]\Big)\right| \leq 2\varepsilon.
$$

Thus, we get

$$
\left|\epsilon_h^k(\widehat{V}) - \epsilon_h^k(V')\right| \leq 2\varepsilon.
$$

Due to $\|a + b\|_\Lambda^2 \leq 2\|a\|_\Lambda^2 + 2\|b\|_\Lambda^2$, we have

$$
\begin{aligned}
&\Big\|\sum_{k=1}^K \phi(s_h^k, a_h^k)\epsilon_h^k(\widehat{V}_{h+1})\Big\|_{\Lambda_h^{-1}}^2\\
&\leq 2\Big\|\sum_{k=1}^K \phi(s_h^k, a_h^k)\epsilon_h^k(V'_{h+1})\Big\|_{\Lambda_h^{-1}}^2\\
&\quad + 2\Big\|\sum_{k=1}^K \phi(s_h^k, a_h^k)\big[\epsilon_h^k(\widehat{V}_{h+1}) - \epsilon_h^k(V'_{h+1})\big]\Big\|_{\Lambda_h^{-1}}^2\\
&\leq 2 \sup_{V \in \mathcal{N}_{h+1}(\varepsilon)} \Big\|\sum_{k=1}^K \phi(s_h^k, a_h^k)\epsilon_h^k(V)\Big\|_{\Lambda_h^{-1}}^2 + \frac{8\varepsilon^2 K^2}{\lambda}.
\end{aligned}
\tag{19}
$$

Here we have an upper bound that is not related to the dataset $\mathcal{D}$. Then applying Lemma D.6 and the union bound, we have

$$
\begin{aligned}
&\mathbb{P}_{\mathcal{D}}\Big(\sup_{V \in \mathcal{N}_{h+1}(\varepsilon)} \Big\|\sum_{K=1}^K \phi(s_h^k, a_h^k)\epsilon_h^k(V)\Big\|_{\Lambda_h^{-1}}^2\\
&\quad > [u(H - h - b_h)]^2\Big[2\log\frac{1}{\Delta} + d\log\big(1 + \frac{K}{\lambda}\big)\Big]\Big)\\
&\leq \Delta\big|\mathcal{N}_{h+1}(\varepsilon)\big|.
\end{aligned}
$$

Set $\Delta = \frac{\delta}{H|\mathcal{N}_{h+1}(\varepsilon)|}$, where $\delta \in (0,1)$. For any $h$, with probability $1 - \frac{\delta}{H}$, there is

$$\sup_{V \in \mathcal{N}_{h+1}(\varepsilon)} \Big\| \sum_{K=1}^{K} \phi(s_h^k, a_h^k)\epsilon_h^k(V) \Big\|_{\Lambda_h^{-1}}^2$$
$$\leq [u(H - h - b_h)]^2 \Big( 2\log \frac{H|\mathcal{N}_{h+1}(\varepsilon)|}{\delta} + d\log\big(1 + \frac{K}{\lambda}\big)\Big)$$

Then, with Equation 19, with probability at least $1 - \delta$, the following inequality holds

$$\Big\| \sum_{K=1}^{K} \phi(s_h^k, a_h^k)\epsilon_h^k(\widehat{V}_{h+1}) \Big\|_{\Lambda_h^{-1}}^2$$
$$\leq 2[u(H - h - b_h)]^2 \Big( 2\log \frac{H|\mathcal{N}_{h+1}(\varepsilon)|}{\delta} + d\log\big(1 + \frac{K}{\lambda}\big)\Big) + \frac{8\varepsilon^2 K^2}{\lambda}, \forall h \in [H].$$

Setting $\varepsilon = \frac{dH}{K}$ and $\lambda = 1$, $L = u(H - h - b_h)\sqrt{\frac{dK}{\lambda}}$, by Lemma D.7, we have

$$\log \mathcal{N}_h(\varepsilon) \leq d\log\Big(1 + \frac{4L}{\varepsilon}\Big) + d^2\log\Big(1 + \frac{8\sqrt{d}B^2}{\varepsilon^2}\Big)$$
$$\leq d\log\Big(1 + 4u(H - h - b_h)d^{-\frac{1}{2}}K^{\frac{3}{2}}H^{-1}\Big) + d^2\log\Big(1 + 8B^2 d^{-\frac{3}{2}}K^2 H^{-2}\Big).$$

Then we set $B = 2\beta$, $\beta = cd \cdot u(H - h - b_h)\sqrt{\zeta}$, and $\zeta = \log(2dHK\delta^{-1})$, where $c > 0$ is a constant. Notice that $u(H - h - b_h) \leq u(H) \leq H$, due to the concavity of utility function $u$ along with $1 \in \partial u(0)$. Therefore, we have

$$\log \mathcal{N}_h(\varepsilon) \leq d\log\big(1 + 4d^{-\frac{1}{2}}K^{\frac{3}{2}}\big) + d^2\log\big(1 + 32c^2 d^{\frac{1}{2}}K^2\zeta\big)$$
$$\leq 2d^2\log\big(1 + 32c^2 d^{\frac{1}{2}}K^2\zeta\big) \tag{20}$$
$$\leq 2d^2\log\big(64c^2 d^{\frac{1}{2}}K^2\zeta\big).$$

Then with the fact that $\log \zeta \leq \zeta$, $\log(1+K) \leq \log(2K) \leq \zeta$, and Equation 20, we have

$$\Big\| \sum_{K=1}^{K} \phi(s_h^k, a_h^k)\epsilon_h^k(\widehat{V}_{h+1}) \Big\|_{\Lambda_h^{-1}}^2$$
$$\leq 2[u(H - h - b_h)]^2 \Big( 2\log(H\delta^{-1}) + 4d^2\log(64c^2 d^{\frac{1}{2}}K^2\zeta) + d\log(1 + K) + 4d^2\Big)$$
$$\leq 2[u(H - h - b_h)]^2 \Big( 2\log(H\delta^{-1}) + 4d^2\log(64c^2) + 4d^2\zeta + 2d^2\log(dK^4) + d\zeta + 4d^2\Big)$$
$$\leq 2[u(H - h - b_h)]^2 \Big( 2d^2\log(dHK^4\delta^{-1}) + 4d^2\log(64c^2) + 9d^2\zeta\Big)$$
$$= 2[u(H - h - b_h)]^2 \Big( 2d^2\log(dHK\delta^{-1}) + 6d^2\log K + 4d^2\log(64c^2) + 9d^2\zeta\Big)$$
$$\leq d^2[u(H - h - b_h)]^2\zeta\Big( 8\log(64c^2) + 34\Big).$$

By setting $8\log(64c^2) + 34 \leq \frac{c^2}{4}$, it holds that

$$\Big\| \sum_{K=1}^{K} \phi(s_h^k, a_h^k)\epsilon_h^k(\widehat{V}_{h+1}) \Big\|_{\Lambda_h^{-1}} \leq \frac{1}{2}cd \cdot u(H - h - b_h)\sqrt{\zeta} = \frac{\beta}{2}. \tag{21}$$

Therefore, based on Equation 17, Equation 18, Equation 21, we have

$$\Big| \mathbb{E}_{s',r}\big[\widehat{V}_{h+1}(s, b - r)\big] - \widehat{\mathbb{E}}_{s',r}\big[\widehat{V}_{h+1}(s', b - r)\big] \Big|$$
$$\leq \Big( u(H - h - b_h)\sqrt{d} + \frac{1}{2}cd \cdot u(H - h - b_h)\sqrt{\zeta}\Big)\sqrt{\phi(s,a)^\top \Lambda_h^{-1}\phi(s,a)}$$
$$\leq \beta\sqrt{\phi(s,a)^\top \Lambda_h^{-1}\phi(s,a)}.$$

Then, with Lemma B.7, we have

$$\mathrm{SubOpt_S}(\widehat{\pi})$$

$$\leq \sum_{h=1}^{H} \left\{ 2cd \cdot u(H - h - b_h) \sqrt{\log \frac{2dHK}{\delta}} \cdot \mathbb{E}_{\pi^\cdot} \left[ \sqrt{\phi(s_h, a_h)^\top \Lambda_h^{-1} \phi(s_h, a_h)} \Big| s_1, b_1^* \right] \right\}$$

$$\leq \sum_{h=1}^{H} \left\{ 2cd \cdot u(H - h) \sqrt{\log \frac{2dHK}{\delta}} \cdot \mathbb{E}_{\pi^\cdot} \left[ \sqrt{\phi(s_h, a_h)^\top \Lambda_h^{-1} \phi(s_h, a_h)} \Big| s_1, b_1^* \right] \right\},$$

where the last inequality holds because $b_h > 0$. Here we finish the proof of Theorem 5.2.

## E    OTHER IMPORTANT LEMMAS

**Lemma E.1 (Concentration of Self-Normalized Processes (Abbasi-Yadkori et al., 2011))** .

*Let $\{\mathcal{F}_t\}_{t=1}^{\infty}$ be a filtration and $\{\epsilon_t\}_{t=1}^{\infty}$ be an $\mathbb{R}$-valued stochastic process such that $\epsilon_t$ is $\mathcal{F}_t$-measurable for all $t \geq 1$. Moreover, suppose that conditioning on $\mathcal{F}_{t-1}$, $\epsilon_t$ is a zero-mean and $\sigma$-sub-Gaussian random variable for all $t \geq 1$, that is,*

$$\mathbb{E}[\epsilon_t | \mathcal{F}_{t-1}] = 0, \qquad \mathbb{E}[\exp(\lambda \epsilon_t) | \mathcal{F}_{t-1}] \leq \exp(\lambda^2 \sigma^2 / 2), \quad \forall \lambda \in \mathbb{R}.$$

*Meanwhile, let $\{\phi_t\}_{t=1}^{\infty}$ be an $\mathbb{R}^d$-valued stochastic process such that $\phi_t$ is $\mathcal{F}_{t-1}$-measurable for all $t \geq 1$. Also, let $M_0 \in \mathbb{R}^{d \times d}$ be a deterministic positive-definite matrix and*

$$M_t = M_0 + \sum_{s=1}^{t} \phi_s \phi_s^\top$$

*for all $t \geq 1$. For all $\Delta > 0$, it holds that*

$$\left\| \sum_{s=1}^{t} \phi_s \epsilon_s \right\|_{M_t^{-1}}^2 \leq 2\sigma^2 \cdot \log \left( \frac{\det(M_t)^{1/2} \cdot \det(M_0)^{-1/2}}{\Delta} \right)$$

*for all $t \geq 1$ with probability at least $1 - \Delta$.*

**Lemma E.2 (Covering Number of Euclidean Ball (Jin et al., 2020))** *For any $\varepsilon \geq 0$, the $\varepsilon$-covering number of the Euclidean ball in $\mathbb{R}$ with radius $R \geq 0$ can be upper bounded by $(1 + \frac{2R}{\varepsilon})^d$.*

## F    NUMERICAL SIMULATION

To verify the algorithms and theoretical results we proposed, we operate the numerical simulation under a specially designed MDP with $\mathcal{S} = \{s_1, s_2, s_3\}$ and $\mathcal{A} = \{a_1, a_2\}$. $s_1$ is set to be the initial state of every episode. The structure of the MDP is shown in Figure 3.

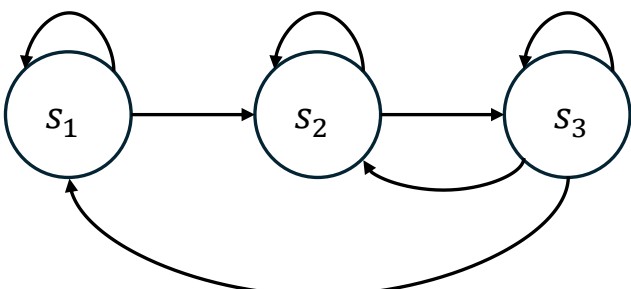

Figure 3: MDP for numerical simulation.

Starting from $s_1$, the agent can transfer to $s_2$ and $s_3$, consequently. At $s_3$, the agent can return to either $s_1$ or $s_2$ with different probabilities according to the action the agent takes. Besides, to add

randomness to the process, at any state the agent have a chance to "stay". The detailed transition and reward function is

$$
\begin{aligned}
&\mathbb{P}(s_1|s_1,a_1) = 0.1, \quad \mathbb{P}(s_2|s_1,a_1) = 0.9, \quad \mathbb{P}(s_1|s_1,a_2) = 0.9, \quad \mathbb{P}(s_2|s_1,a_2) = 0.1 \\
&\mathbb{P}(s_2|s_2,a_1) = 0.1, \quad \mathbb{P}(s_3|s_2,a_1) = 0.9, \quad \mathbb{P}(s_2|s_2,a_2) = 0.9, \quad \mathbb{P}(s_3|s_2,a_2) = 0.1 \\
&\mathbb{P}(s_1|s_3,a_1) = 0.1, \quad \mathbb{P}(s_2|s_3,a_1) = 0.1, \quad \mathbb{P}(s_3|s_3,a_1) = 0.8 \\
&\mathbb{P}(s_1|s_3,a_2) = 0.4, \quad \mathbb{P}(s_2|s_3,a_2) = 0.4, \quad \mathbb{P}(s_3|s_3,a_2) = 0.2
\end{aligned}
$$

and

$$
\begin{aligned}
r(s_1,a) &= 0, \quad \forall a \in \mathcal{A} \\
r(s_2,a_1) &= 0, \quad r(s_2,a_1) = 0.5 \\
r(s_3,a_1) &= 0, \quad r(s_3,a_1) = 1.
\end{aligned}
$$

The idea of constructing this MDP basically follows the idea of making a "dilemma", where the good action with a larger reward has a larger probability of leading the agent to a bad state. By this construction, considering the risk is important. We evaluate the CVar scenario with $\alpha = 0.5$. The result is shown in Figure 4.

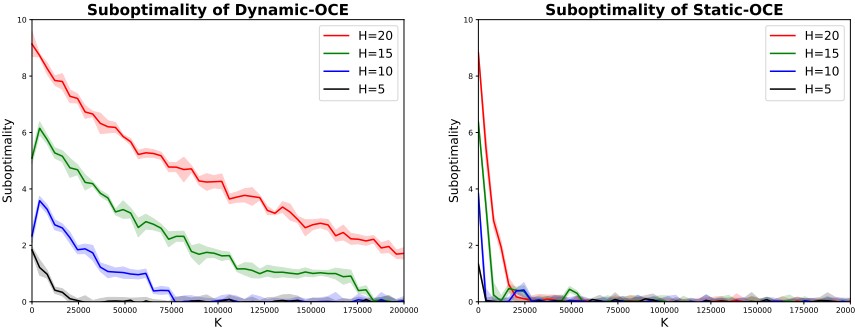

Figure 4: The suboptimality of the learned policy from Algorithm 1 and Algorithm 2. The mean results are plotted as solid lines. The error bar area corresponds to the 90% confidence interval.

By operating the simulation with $H = 20, 15, 10, 5$, we can conclude that the history-dependent policy learned by Algorithm 2 have lower suboptimality with the same $H$ and $K$.

# G    STATEMENT ON THE USE OF LARGE LANGUAGE MODELS

In the paper writing stage, large language models (LLMs), specifically OpenAI's ChatGPT, were employed to assist with tasks such as language polishing and grammar checking. GitHub Copilot was occasionally used for code completion and checking when writing test code. The models were not used to generate scientific content, proofs, research ideas, or code frameworks. All technical contributions, theoretical derivations, algorithmic developments, and algorithm implementations are the sole work of the authors. We have carefully reviewed and verified all text suggested by the LLMs to ensure accuracy and compliance with academic standards.

