# OpenReview forum: "General Risk Measure meets Offline RL: Provably Efficient Risk-Sensitive Offline RL via Optimized Certainty Equivalent"
_ICLR.cc/2026/Conference — ICLR 2026 Conference Withdrawn Submission_

### Official Review · Reviewer_MsaU · 2025-10-29

**Soundness:** 3
**Presentation:** 3
**Contribution:** 2
**Rating:** 4
**Confidence:** 3

**Summary:**

This paper studies the provably efficient risk-sensitive RL under the offline setting with a general risk measure, the optimized certainty equivalent (OCE), which captures various risk measures studied in prior risk-sensitive RL works, such as value-at-risk, entropic risk, and mean-variance. The authors (i) introduce the first offline OCERL frameworks and propose corresponding pessimistic value iteration algorithms (OCE-PVI) for both dynamic and static risk measures; (ii) establish suboptimality bounds for the algorithms, which can reduce to known results for risk-sensitive RL as well as risk-neutral RL with appropriate utility functions; (iii) derive the first information-theoretic lower bound of the sample complexity of offline risk-sensitive RL, matching the upper bounds and certifying optimality of the proposed algorithms; and (iv) propose the first provably efficient risk-sensitive RL with linear function approximation for both dynamic and static risk measures, together with rigorous suboptimality bounds, yielding a scalable and model-free approach.

**Strengths:**

1. This paper is well-written and easy to follow.
2. This paper is well executed and provides multiple results, including offline RL with dynamic and static OCE criteria in the tabular and linear function approximation settings.

**Weaknesses:**

1. My major concern is that the technical novelty of this paper is limited. While this paper introduces the first algorithms and theoretical results for offline RL with OCE criteria in the tabular and linear function approximation settings, given that RL with the dynamic and static OCE criteria, offline RL and linear MDPs are well studied in the literature, the algorithm design and theoretical analysis in this paper seem to be a combination of existing techniques in RL with the OCE criteria, offline RL (i.e., the pessimism idea) and linear MDPs (i.e., least squares value iteration).
2. There is no experiment provided in this paper, which limits the practicability of the proposed algorithms.
3. In algorithm 2 for the static OCE criterion, under the augmented state space, it should be enough to consider deterministic policies? Why does Line 9 in Algorithm 2 use $argmax_{\pi_h}$, instead of $argmax_{a \in \mathcal{A}}$?
4. Minor comment: The authors should enlarge the font size in the algorithm pseudo-codes to keep it the same as that of the main text.

**Questions:**

Please see the weaknesses above.

---

### Official Review · Reviewer_D32H · 2025-11-01

**Soundness:** 3
**Presentation:** 3
**Contribution:** 2
**Rating:** 4
**Confidence:** 4

**Summary:**

This paper proposes a unified algorithm framework based on the general risk measure "Optimality Criteria Equivalence" (OCE) for the risk-sensitive offline reinforcement learning problem. The authors have designed the first provably effective offline algorithms (DOCE-PVI/SOCE-PVI) for both dynamic and static OCE formulas, and established corresponding suboptimality bounds and the first information-theoretic lower bounds for offline risk-sensitive RL. In addition, this work further proposes the first provably effective risk-sensitive RL algorithms using linear function approximation under the OCE framework (DOCE-PLSVI and SOCE-PLSVI).

**Strengths:**

The paper successfully introduces the OCE general risk measurement framework into the offline RL setting. This addresses a key gap in existing work that is either limited to specific risk measures (such as CVaR or entropy risk) or only focuses on online settings.

The paper clearly distinguishes between dynamic OCE and static OCE in two different settings. For the challenge of non-Markov strategies in static OCE, the authors adopt the augmented MDP (Augmented MDP) technique, which is methodologically rigorous.

For Linear Function Approximation setting, this paper considers the ridge regression, separately estimating the reward function and the expected term in the OCE.

The results recover classical offline RL bounds (e.g., Jin et al., 2021) as special cases and align with known online risk-sensitive RL lower bounds (e.g., Xu et al., 2023; Chen et al., 2023).

**Weaknesses:**

My major concerns is that the technical novelty of this paper is not clear. The theoretical machinery—pessimistic value iteration, concentration-based bonuses, and Bellman operator analysis—directly parallels those in existing risk-sensitive RL (Fei et al., 2020; Xu et al., 2023; Chen et al., 2023) and OCE-based online RL (Wang et al., 2024). It is not surprising that we can achieve efficient offline reinforcement learning with OCE. For me, the proofs in this work are merely straightforward applications of existing algorithms, which limits the technical novelty.  Therefore, I suggest the authors to incorporate more discussion to highlight the technical contribution of this paper.

---

Wang, Kaiwen, et al. "A reductions approach to risk-sensitive reinforcement learning with optimized certainty equivalents.", ICML 2025.

**Questions:**

see weakness part.

---

### Official Review · Reviewer_rJqu · 2025-11-01

**Soundness:** 1
**Presentation:** 3
**Contribution:** 2
**Rating:** 2
**Confidence:** 5

**Summary:**

This paper studies risk-sensitive offline reinforcement learning under a general risk measure known as the Optimized Certainty Equivalent (OCE). The OCE framework was originally introduced by Ben-Tal and Teboulle (2007) and it has since been shown to unify several classical risk measures, including Conditional Value-at-Risk (CVaR), entropic risk, and mean–variance formulations.

The authors propose and analyze two algorithms: Dynamic-OCE RL and Static-OCE RL. The dynamic formulation has appeared in earlier works and applies to certain classes of recursive risk measures, while the static formulation is meant to generalize to measures such as CVaR and entropic risk.

Although the paper presents a theoretical analysis for both algorithms, the results closely mirror those established for standard episodic, risk-neutral offline RL with cumulative rewards, with minimal adaptation to the OCE framework. In essence, the work represents a direct transposition of existing episodic RL analyses into the OCE setting, without addressing the core technical challenges that make OCE-based formulations nontrivial: specifically, the necessity of state augmentation and discretization of the budget variable. Consequently, the theoretical development appears to rest on conceptually incorrect assumptions, rendering the analysis fundamentally flawed and mathematically unsound.

**Strengths:**

The paper addresses a problem formulation of clear contemporary interest, given the growing attention to risk-sensitive decision-making in recent years. It represents a notable attempt to unify developments in risk-neutral RL with those in risk-sensitive RL, by employing the OCE as a common analytical framework. The overall methodological structure follows a mostly sound and well-established approach, and the presentation of the algorithms is organized.

**Weaknesses:**

Incomplete literature coverage.
The paper overlooks several recent and relevant works, particularly those presented at major conferences, that have focused on the theoretical and algorithmic understanding of the OCE framework in reinforcement learning.

Superficial treatment of discretization.
The authors largely brush aside the central difficulty associated with discretizing the auxiliary state variable. While finer discretization may reduce approximation error, discretization fundamentally alters the underlying MDP structure, making both the theoretical analysis and algorithmic updates more complex. In particular, the proposed update rule is internally inconsistent: the value updates at each iteration may not correspond to points in the discretized grid, thereby invalidating the recursion as written.

Incorrect handling of state augmentation and scaling.
Although the authors mention state augmentation in the static-OCE formulation, their theoretical bounds entirely ignore its implications. In the static case, the effective state space is the augmented space whose cardinality explicitly depends on the discretization resolution. Consequently, the bounds presented in the theorems should scale with the augmented state dimension. Because this dependence is omitted, the claimed results are mathematically inconsistent: for instance, the suboptimality bounds for static OCE cannot recover known results for special cases such as the entropic risk measure, contrary to the authors’ claims.

**Questions:**

1- To achieve a vanishing approximation error, the discretization granularity of the auxiliary variable must scale with the number of trajectories or episodes. How does this dependence influence your final suboptimality bound?

2- In the static-OCE formulation, the effective state space is the augmented space that includes both the original state and the budget variable. Yet, your theoretical bounds depend solely on the size of the original state space. Could you clarify why the augmented dimension does not appear in the final bound?

3- Under what precise conditions is the dynamic OCE formulation applicable? Since many common risk measures (e.g., CVaR) are inherently non-recursive and therefore incompatible with dynamic OCE, could you clarify the motivation for studying this case and its practical relevance within the broader risk-sensitive RL literature?

---

### Official Review · Reviewer_orJR · 2025-11-01

**Soundness:** 3
**Presentation:** 3
**Contribution:** 2
**Rating:** 4
**Confidence:** 3

**Summary:**

This paper studies risk-sensitive offline reinforcement learning under the general Optimized Certainty Equivalent (OCE) framework, which encompasses CVaR, entropic, and mean-variance risk measures. The authors design two pessimistic value iteration algorithms for dynamic and static OCEs, derive the first provable sample-efficiency guarantees in offline OCE-RL, and extend the results to linear function approximation. Both upper and lower bounds are provided, showing near-optimal dependence on horizon and dataset size.

**Strengths:**

- The paper is well written and the motivation is very clear.
- Extend the OCE to offline setup to fill the gap.

**Weaknesses:**

- The main technical ideas combine well-known pessimism principles (Jin et al. 2021) with OCE-based risk modeling. The extension is valuable but not a radical theoretical leap.
- Though the work is most theoretical, the evaluation part should consider more benchmark and more complex setup to clearly demonstrate its practicality and efficiency.

**Questions:**

1. How tight are the lower bounds relative to prior results in CVaR or entropic RL?
2. Does the proposed method remain stable when OCE’s generator function is non-convex?

---

### Note · Authors · 2025-11-14

I have read and agree with the venue's withdrawal policy on behalf of myself and my co-authors.